# Using an Intelligent Control Method for Electric Vehicle Charging in Microgrids

**Samaneh Rastgoo, Zahra Mahdavi, Morteza Azimi Nasab, Mohammad Zand and Sanjeevikumar Padmanaban ***

Department of Electrical Engineering, IT and Cybernetic, University of South-Eastern Norway, Kjølnes Ring 56, 3918 Porsgrunn, Norway
* Correspondence: sanjeevi_12@yahoo.co.in

**Abstract:** Recently, electric vehicles (EVs) that use energy storage have attracted much attention due to their many advantages, such as environmental compatibility and lower operating costs compared to conventional vehicles (which use fossil fuels). In a microgrid, an EV that works through the energy stored in its battery can be used as a load or energy source; therefore, the optimal utilization of EV clusters in power systems has been intensively studied. This paper aims to present an application of an intelligent control method to a bidirectional DC fast charging station with a new control structure to solve the problems of voltage drops and rises. In this switching strategy, the power converter is modeled as a DC fast charging station, which controls the fast charging of vehicles with a new constant current or reduced constant current method and considers the microgrid voltage stability. The proposed method is not complicated because simple direct voltage control realizes the reactive power compensation, which can provide sufficient injected reactive power to the network. As a result, the test is presented on a fast charging system of electrical outlets with a proposed two-way reactive power compensation control strategy, in which AC/DC converters are used to exchange two-way reactive power to maintain the DC link voltage as well as the network bus voltage in the range of the basis. This charging strategy is carried out through the simulation of fast charge control, DC link voltage control, and reactive power compensation control to adjust the voltage and modify the power factor in the MATLAB software environment and is then verified. Finally, the results indicate that the proposed method can charge with high safety without increasing the battery's maximum voltage. It can also significantly reduce the charging time compared to the common CV mode.

**Keywords:** intelligent; optimization; car parking; battery

## 1. Introduction

Road traffic is one of the main causes of greenhouse gas emissions. Along with rising fuel prices and the issue of high energy efficiency, electric cars are predicted to be more widespread in the next few decades. Electric cars are more desirable devices to meet the needs of long trips. An electric car uses batteries and a combustion engine to minimize fuel consumption. A plug-in hybrid electric vehicle is charged when plugged into a home charger or a public charging station. However, this causes challenges in the electrical microgrid system. The high penetration of hybrid electric vehicles adds to the current peak load and creates a new peak load. This can cause voltage fluctuations and a transformer overload. Voltage deviations can cause damage to electrical appliances, while a persistent overload can cause a transformer to overheat, which can cause a blackout. Fortunately, the development of advanced measurement systems and communication systems enables us to develop better algorithms to overcome these problems. Therefore, the timing and speed of charging hybrid electric vehicles can be controlled to reduce the maximum load, called load demand management. In addition, numerous studies have been conducted in

various fields, such as microgrids, energy storage systems, renewable energy, and electric vehicles (EV), to exploit power systems in a distributed manner. Among these, EVs have attracted the most attention because, if their expansion continues, they are likely to become the most effective solution available in the future. In addition, since an EV can be considered a mobile battery in power systems, power system operators such as microgrids should consider the accuracy of EV uncertainties. As a result, EVs are studied in terms of their uncertainty patterns. In particular, issues like how EVs work and affect distribution systems, such as microgrids and smart grids, are studied.

*Literature Review*

In this section, studies conducted in the fields of load demand management and plug-in hybrid electric vehicles are presented. Source [1–3] presents a hierarchical control algorithm for understanding the trade-off and cooperation between plug-in hybrid electric vehicle charging and wind power. The three-level controller proposed in this resource uses plug-in hybrid electric vehicles to compensate for wind power fluctuations and, thus, to indirectly adjust grid frequency. This connection between an electric car and wind leads to preserving environmental resources and using clean energy sources. Reference [4] deals with the problem of bridging the gap by controlling a large population of plug-in hybrid electric vehicles. It also presents a decentralized algorithm, but it only proves the optimal state in the homogeneous state, where all hybrid electric vehicles have the same departure time, energy requirement, and maximum charging power. In reference [5], a control signal from a supplier company is modified to drive and update profiles of plug-in hybrid electric vehicles. This algorithm converges on optimal charging profiles in homogeneous and heterogeneous cases. However, if the communication between the company and the plug-in hybrid electric vehicle is asynchronous, the algorithm's performance can be affected.

In reference [6], increasing or decreasing the maximum load is used to reach the desired load waveform, considering the consumer's preferences and the load's characteristics. A multi-agent system solution was accepted in [7], which had the features of adaptability and high scalability. Paper [8] thoroughly examines the issues surrounding plug-in hybrid electric vehicles.

In the following, we describe article [9], in which the issue of load response in intelligent systems in the presence of electric vehicles is discussed. We focus on the impact of hybrid electric vehicle charging on low-voltage transformers (LVTs). The goal is to smooth out the load curve of each LVT and, at the same time, meet each customer's requirements for plug-in hybrid electric vehicles that must be charged at the required level at a time. We introduce the dynamic model of plug-in hybrid electric vehicles. Finally, the DSM problem for plug-in hybrid electric vehicles is planned as an optimization problem.

The latest articles in the present paper's field include the research stated in the references [10,11]. A methodology for scheduling EVs and PHEVs' charging and discharging processes was proposed in reference [12]. EVs' production, consumption, mobility and market price uncertainties are considered. These articles balance the system based on market prices. Market prices are used as a reference for EV charging and discharging schedules to achieve load leveling. In fact, with this methodology, it is possible to obtain a good, although not optimal, solution. The rules used to model the constraints of the problem are only considered for one period (15 min) in each set of rules; therefore, the car charging process considers the needs of travel and not the system's needs. In addition, the technical limitations of the network are not considered.

The valley removal and peaking methodology described in reference [13] take V2G into account. This system considers two levels of control, i.e., the control center of smart networks and the V2G control center. The smart grid control center sends control signals to the V2G control center based on an "optimal energy distribution", The V2G control center controls the V2G charging and discharging process. The predicted load and hourly usage goals the user is exposed to are considered. The objective function minimizes the differences between the target and the actual power demand.

Article [14] investigates the impact of V2G on system operation costs and the power demand curve for a distribution network, focusing on distributed energy sources. This effect has been studied from several perspectives, including minimizing operation costs and minimizing the difference between the minimum and maximum demand during a 24 h period. A multi-objective problem is presented to minimize operation costs and the power demand curve. In the proposed methodology, the consequences of the number of different EVs in the distribution network are analyzed considering the vehicle's characteristics and application characteristics under the report of the US Transportation Agency.

Optimizing the power demand curve increases the minimum load consumption or decreases the maximum load consumption, which leads to a power demand curve close to a rectangular shape. The electric power demand curve is evaluated by analyzing the value of the load factor. The load factor related to the total energy consumption is obtained from the peak consumption within a certain time horizon.

Article [15] investigated the integration of an electric vehicle connected to PEV electricity with an EMS energy management system, serving as the first step in understanding its potential role in the energy resources of local semiautonomous groups and microgrids. In their past research, many authors discussed new power electronics technologies accompanying various DERC energy sources, especially variable-frequency AC/DC sources, solar panels, batteries, and asynchronous generators such as microturbines. High-speed switching disconnects and reconnects to the integrated network. These power electronics could form a microgrid that operates semi-autonomously from the traditional centralized power line. Additionally, this paper analyzes the integration of PEV in a building energy management system (EMS) and the difference between vehicle-to-microgrid (V2G) and microgrid-to-vehicle frameworks. These relationships are modeled using the DER_CAM energy resource distribution criteria approval model, which provides optimal combinations of equipment required by the microgrid with the lowest cost, carbon footprint, or other criteria. The DER_CAM model is an optimization tool that minimizes annual energy costs for microgrids. For an office building using a PEV connection under the business model, where the distributed threshold values for the maximum payment are obtained, it is concluded that the economic effects have their limitations. However, this paper shows that some economic benefits are created due to the avoidance of charging demand and TOU rates. The strategy adopted by the office building to reduce costs is to use PEU batteries in the afternoon hours. The results obtained would vary depending on the case. Of course, the results of the $CO_2$ removal are not presented in this article.

In the paper [16], a modified particle swarm optimization (PSO) technique is presented and demonstrated to solve the problem of managing highly influential energy resources from distributed generation and pluggable (V2G) electric devices. The purpose of reducing and minimizing operation costs, i.e., energy costs, especially the issue of managing these resources, is to obtain a smart network. The reforms used for the PSO technique aim to improve and promote its competence and suitability to solve the mentioned problems. This present article is an evolution of traditional particle swarm optimization called ASMPSO, which is used for the problem of energy resource management in smart grids. It considers and addresses realistic grids using energy sources, such as distributed generation (DG), based on renewable energy sources and plug-in electric vehicles (EVs) (V2G). An accurate AC load spread and grid physical constraints check the practicality and feasibility of a reliable solution.

Additionally, execution time is critical for up-to-date scheduling due to the numerous resources involved and the need to simulate various operational scenarios. However, they should be properly adapted to the characteristics of the problem. The main advantage of this technique is better constraints with a simpler mechanism to adjust the speed constraints intelligently and dynamic parameterization to create a more accurate solution to improve the fitness of the problem.

Charging and discharging electric vehicles in distribution networks has always been one of the leading challenges in operating these vehicles. Due to the movement of cars in

a network, the charging time and amount of charging always vary. Therefore, when many electric cars enter a distribution network, the issue of operating these cars becomes important, and serious planning can be necessary. When locating the construction of charging and discharging stations, considering their effects on a network and the problem of voltage deviation and losses in the distribution network should be considered. The further the charging stations are from the main input distribution point, the higher the losses in the network. Network voltage drops become more serious when the main input distribution substations are outside urban areas. Therefore, locating and estimating the size of charging and discharging stations for electric vehicles can be realized with a reduction in losses as a goal. In reference [17], the network losses related to the operation of electric vehicles were determined by carrying out a load distribution. With geographic information, the distance between the main entrance distribution post and the location of the electric vehicle charging station was calculated. The location of charging stations in distribution systems can be determined using the reduction in network losses and voltage deviation on all buses. The location and capacity problem can be solved by minimizing the total power loss and voltage drop. In reference [18], the problem's constraints are the losses, voltage, and power in the buses and the number of cars in the stations.

Charging electric vehicles at charging stations can be expensive [19–21]. The total cost includes the loss cost, operation cost, maintenance cost, and network loss cost. The investment cost includes charging equipment, feeders, and transformers. Operating costs include charging, active power filter, reactive power compensation, and human resource costs. Operation costs include maintenance costs for transformers, charging equipment, and other equipment in the station. The cost of network losses depends on the amount of power loss and energy price. The limitations of the problem are the number of charging stations, the voltage and current limits of the buses, and the size of the stations' capacity.

One of the most significant challenges to using electric vehicles is the lengthy charging and limited use time (due to battery discharge). According to the road transportation network, the correct location of electric vehicle charging stations can lead to longer driving distances and more efficient electric vehicles. Limitations in the battery capacity of electric vehicles can impact the location of charging stations [22]. Therefore, the location of charging stations should be selected in such a way as to guarantee the provision of charging services in Zarib, given the high penetration of electric vehicles. This issue can be addressed by building charging stations where electric vehicles have easier access [23]. In addition, the capacity of the charging stations should be proportional to the number of cars so that there is no traffic and the delay for the owners of electric cars is as minimal as possible. Several solutions have been presented in the literature to resolve this problem, defined as the location of charging stations [24]. Some plans are based on meeting the demand for electric vehicles. In these designs, the amount of power requested by cars is initially estimated. Charging stations are seized, and some plans to increase the number of covered roads [25–28]. In some designs, charging systems with numerous stations and low capacities have been considered [29–32].

We can achieve a better working point when the electricity distribution and road transportation networks are seen together. Here, the optimal placement can be determined by considering the issues related to the electrical network and the issues related to road transportation. Here, the charging stations should be built in places with high access on roads, and the location of the stations in the power grid should cause the minimum amount of voltage drops and losses [33]. Because the volume of the constraints is large, it is possible to use constraint weighting and consider them as part of the objective function to reduce the number of constraints. For example, in reference [34], meta-invention methods were used to solve such a problem. In references [35,36], evolutionary algorithms were used to solve the problem in a multi-constraint mode. In reference [37], the accessibility of charging stations and the cost of each trip were considered as parameters of the problem.

Some authorities have used plans in which scattered production sources and new energies are used, and the optimal location of charging stations is selected according to the existence of these sources in the network. For example, sources [38–41] used distributed generation sources and new energy in charging stations. Reference [42] discussed the advantages of using solar systems in EV charging stations. References [43,44] examined the effect of production resources on choosing the size of the charging station. In reference [45], solar panels were installed on the roof of a charging station, and the seasonal radiation changes were also investigated. In reference [46], an energy storage system was considered in a charging station, and its effect on the location of the charging station was investigated. In references [47], the design of electric vehicle charging stations inside a microgrid was investigated. Reference [48] mentions the design of an electric vehicle charging system inside a microgrid. In reference [49], production planning for distributed production units in a microgrid and its effect on the performance of electric vehicle charging stations were investigated. Reference [50] discusses controlling an energy storage system in a microgrid with an electric vehicle charging station. In reference [51], the experimental results obtained from creating an electric vehicle charging station inside a microgrid were stated. Reference [52] discusses controlling interactions between V2G technology and microgrids. Reference [53] discusses the charging and discharging of electric vehicles to minimize microgrid load fluctuations. In reference [54], the establishment of charging stations in distribution systems with a high penetration of solar panels was investigated. In reference [55], the capacity of scattered production sources, along with locating electric vehicle charging stations, was located and determined.

In general, the purpose of this research can be summarized as follows:

(A) The proposed method checks the time loop and charging level according to the objectives to reach the desired answer in the objective functions according to the predetermined scenarios.

(B) The proposed method can perform intelligent charging with high safety without increasing the battery's maximum voltage.

(C) The proposed method designs the charging station with simultaneous consideration of the three goals: maximizing the charging demand every hour of the day and night, improving the network load profile, and minimizing the operation costs.

For this purpose, in the optimal charging station structure, a storage system was considered to store electric energy during peak blackout hours, and this energy was used to charge electric vehicles during peak consumption hours, reducing the purchase cost. Electric energy also improves the load profile of networks. The most important issue in this was determining the optimal capacity for the desired storage system, which required knowing the amount of electric vehicle charging demand at different times of the day and night. Therefore, in order to achieve the final research goals, the framework of the article was formed based on four models, which were:

An electric vehicle charging demand model
A model for the main components of the charging station
A model of the effect of the charging station performance on the power network
An optimal supply model for electric vehicle charging.

Next, issues related to charging stations were reviewed in the second part, and in the third part, the charging station models and suggested charging stations were reviewed. The simulations and comparisons between them and the results were given in the fourth and fifth parts.

## 2. DC Bidirectional Fast Charging Station

Electric vehicle chargers can be installed on-board and off-board the vehicles. On-board chargers usually have a small size and limited power and are used for slow charging. The off-board charger is created especially for fast charging. Fast charging networks play an important role in promoting electric cars, as they offer fast charging that can

reduce drivers' worries. This project was a 50 kW off-board fast charging station with a voltage range of 50 to 600 Vdc and a permissible current of 125 A. A common DC fast charging station configuration consists of AC/DC and DC/DC power converters. The uni-directional DC fast charging station rectifies the three-phase AC input to the DC output. The DC/DC converter is used to shift the DC output to a suitable level, which is suitable for charging electric vehicle batteries. In addition, the two-way fast charging station can be controlled with power converter modeling. The key control structure was to review the power transferred between the DC fast charging station and the power grid. Figure 1 shows a diagram of an electric vehicle's (EV) system connection to the grid. The EV system consists of a comprehensive AC/DC voltage source converter and an independent DC/DC voltage source converter for each EV. DC/DC converters are connected in parallel to the common DC buses, which are regulated with the comprehensive AC/DC converter system.

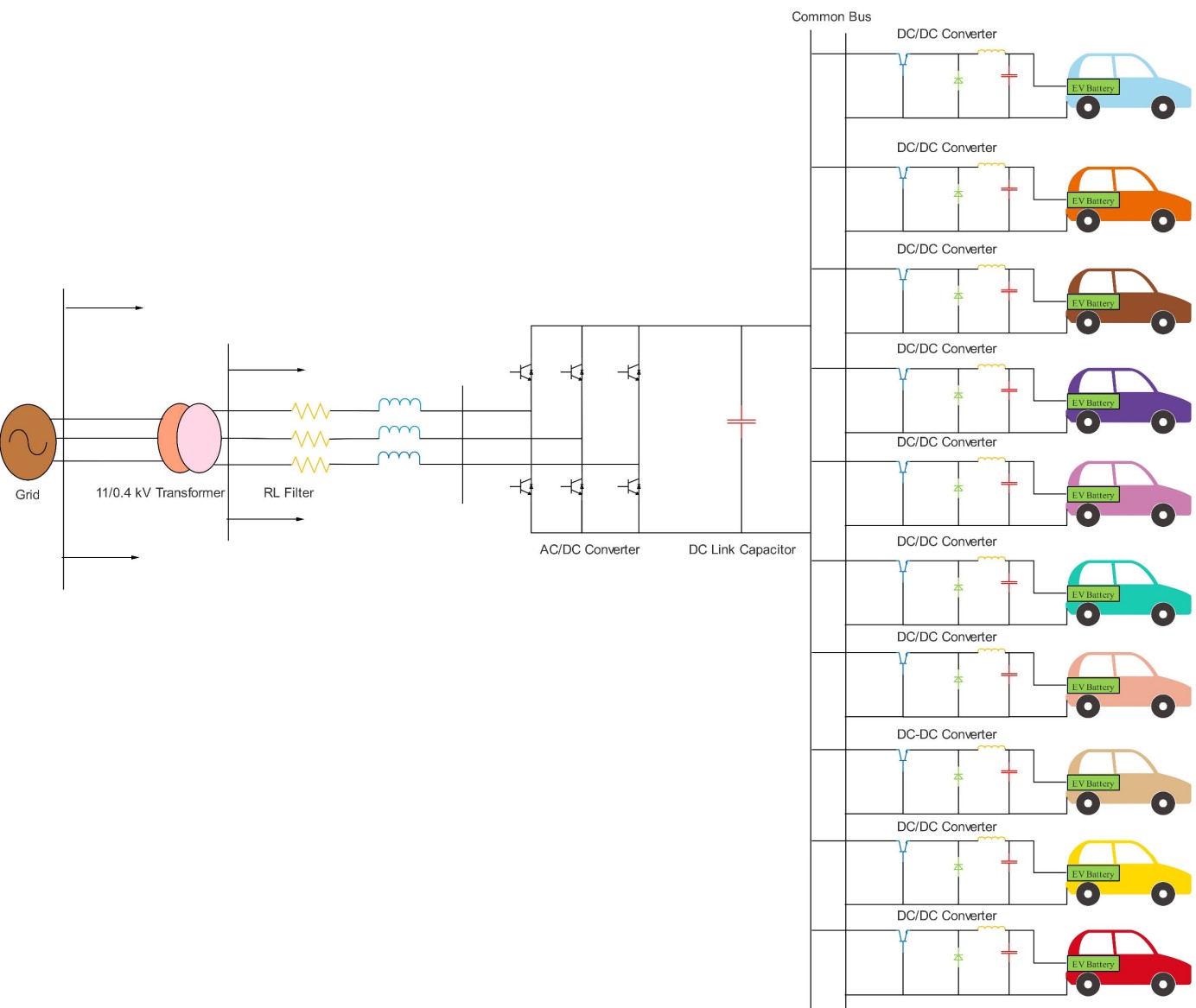

**Figure 1.** Configuration of a grid-connected EV system.

An RL filter was placed between the EV system and the low-voltage distribution network to filter the harmonics generated by the fast switching operation of the power converters. In this project, the fast charging system was capable of a bidirectional Q

transmission with the help of a powerful control system. In AC/DC converters and DC/DC converters, the control structure of the fast charging station or the two-way DC fast charging station can play a role. Using the comprehensive AC/DC converter control to inject reactive power into a network, it is possible to control the voltage, correct the power factor, and keep the DC bus voltage at a constant value. Moreover, EVs can be charged and discharged by controlling the DC/DC converters.

### 2.1. Comprehensive AC/DC Converter Control

The comprehensive AC/DC converter compensated for the reactive power and adjusted the distribution network voltage during EV charging.

A graphical description of the reactive power exchanged between the grid, and the EV system is shown in Figure 2. Reactive power flowed based on the voltage difference between the grid bus and the voltage bus of the EV converter. Positive reactive power indicated that reactive power flowed from the EV converter to the distribution network. If the voltage E was greater than the network bus voltage range (V), the reactive power was transferred from the EV to the network. In Figure 2, when the voltage E was lower than the network voltage range, the reactive power was transferred from the network to the EV.

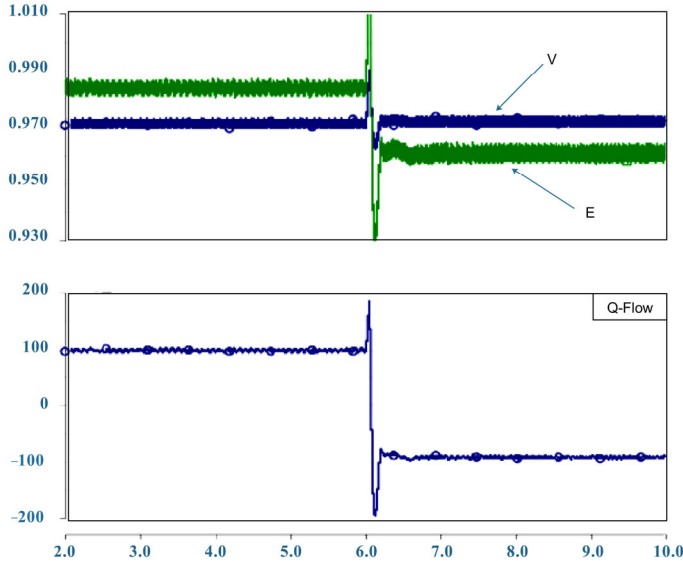

**Figure 2.** Reactive power generation is based on the voltage difference between the EV system and the grid.

Figure 3 shows a block diagram of the comprehensive AC/DC converter. The power converter injected Qs into the network if the output voltage range of the converter was greater than the network voltage range. Similarly, the active power (Ps) would be transferred from the converter to the grid if the converter voltage angle was greater than the grid voltage angle.

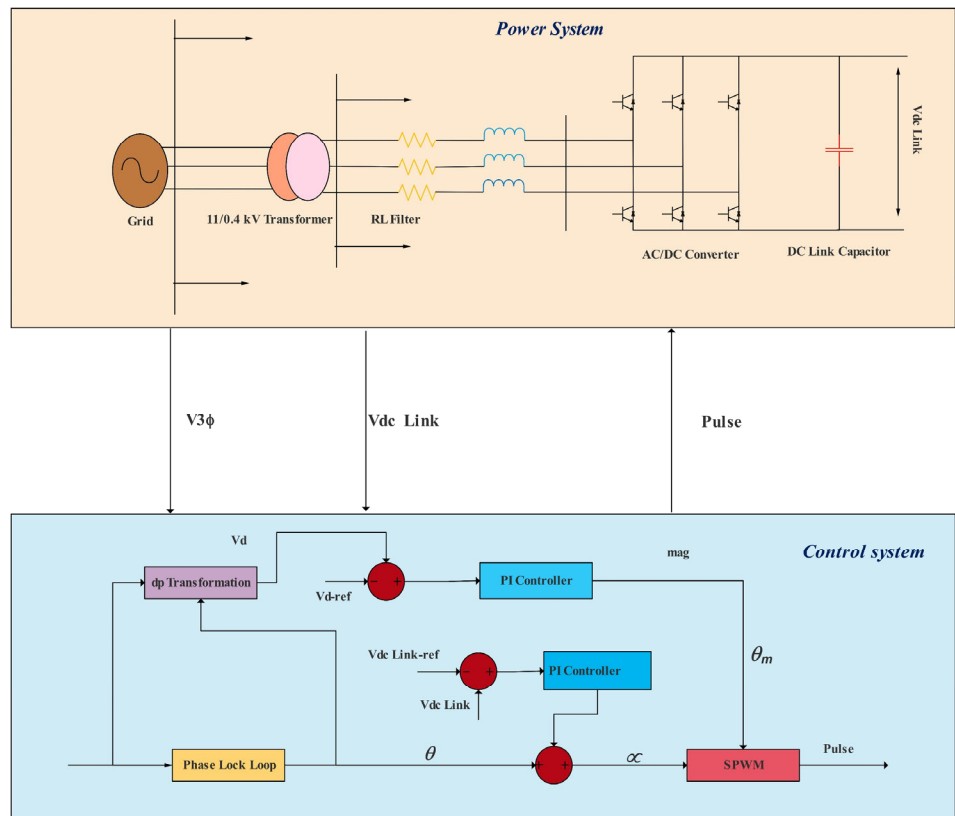

**Figure 3.** Block diagram of comprehensive AC/DC converter control.

The phase angle was the transition angle between the converter and network voltage (v3). Vd is the direct voltage, and vq is the quadrature voltage of a phase of the network voltage. Vd was compared with a reference voltage. There were two types of modes in this project. In the first case, the reference voltage was set at 0.96 p.u., representing the steady-state voltage. In the second case, the reference voltage was the allowed network voltage set to 1 p.u. Then, the error signal was sent to the PI controller, and the output signal (mag) amplitude was obtained. This output signal (mag) was the amplitude of the modulation signal. In the same way, the DC voltage of the converter was compared with the DC voltage value of 800 volts. The Pi controller received the error signal and determined the output signal angle (m).

The sum of the phase angle and the angle of the output signal (m) determined the angle of the modulation signal.

The overall modulation signal was compared with a triangular waveform (SPWM, sinusoidal pulse width modulation) to obtain switching pulses. These pulses were sent to the AC/DC converter for optimal control performance. The AC/DC converter's control structure allowed for a sufficient reactive power transfer to adjust the voltage of the network bus, giving a uniform and permissible voltage in two states; it also improved the power factor of the network. In addition, the controller kept the DC common bus at a constant value of 800 volts.

### 2.2. DC/DC Converter Control

The DC/DC converter was used to develop a fast-charging strategy control for electric vehicle battery charging. There are many ways to control a battery's charging, including constant current charging, voltage charging, power charging [52], pulse charging, and slow charging [30]. The combined method of constant voltage and the constant current was used in charging the lithium batteries of electric vehicles (lithium-ion). The function of the constant current/constant voltage (CC/CV) charge control was performed in two stages. First, the battery was charged with a constant current at a high level. Most of the

batteries in this mode were charged to 80% of their capacity in 30 min. When the battery voltage reached more than the defined value, the CC mode changed to the CV mode. The battery was charged with a decreasing current while its voltage was kept constant. Although 20% of the battery was charged in the CV mode, the time spent in this mode was longer than in the CC mode [31]. In this project, a new charge control strategy for the DC/DC converters was proposed: a constant current/reduced constant current (CC/RCC). During the charging process, the battery voltage (Vbat) was continuously monitored, compared with the predefined value (*Vmax* = 293), and kept below this value. The default value was chosen because it was lower than the actual value (403 V), creating a safe margin for the battery.

Figure 4 shows the block diagram of the DC/DC converter control. In the CC mode, the output current of the DC/DC converter (Idc) was compared with the reference current value, which was approximately 100 amps. The error signal was sent to the PI controller to generate the output signal amplitude (mag_dc). Based on mag_dc, the SPWM modulation generated pulses sent to the DC/DC converter for optimal control performance. When the battery voltage reached more than the allowed value, the CC mode changed to the RCC mode.

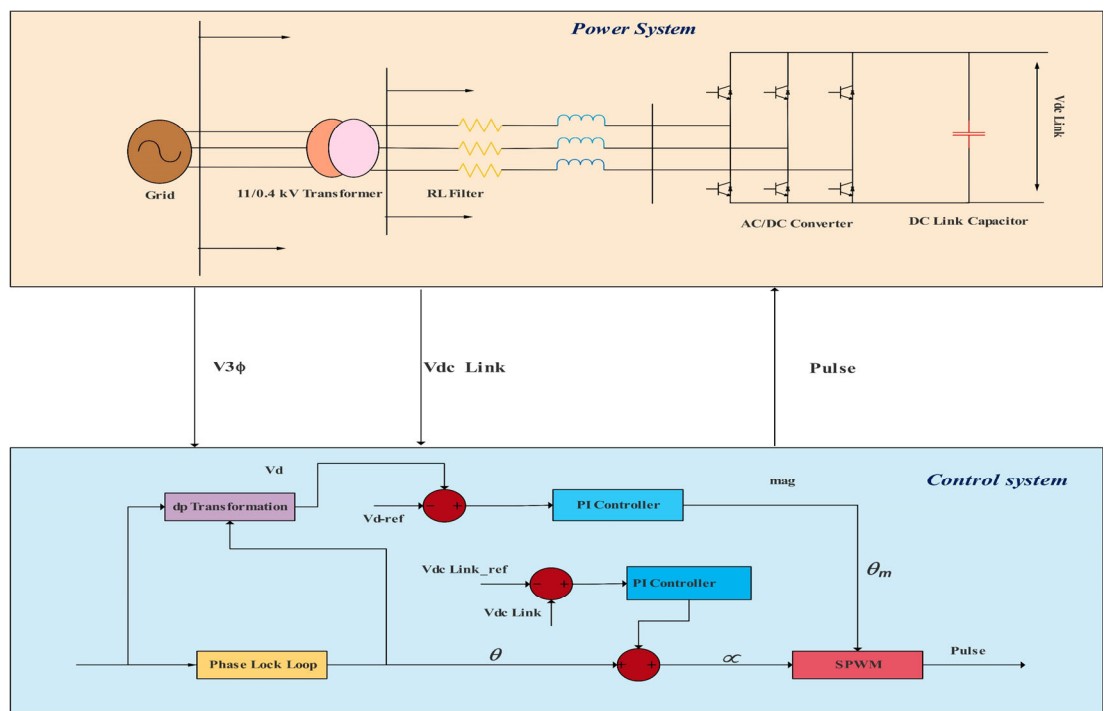

**Figure 4.** Block diagram of DC/DC converter control.

In the RCC mode, the reference DC current amplitude was kept constant and decreased whenever the battery voltage reached a predetermined value. The DC output current (Idc) of the DC/DC converter was measured and compared to the reduced DC reference current, and the result was sent to the PI controller to produce the output signal amplitude (Mag dc).

The SPWM generated the switching pulses and sent them to the DC/DC converter. Figure 5 shows the current and voltage waveforms of the proposed CC/RCC design. The battery voltage was monitored and kept below the defined value for safety reasons. In the CC mode, a high charging current was drawn from the power grid to charge the EV battery quickly, and the battery voltage increased rapidly. The CV mode is like the previous methods; most often, the battery state of charge (SOC) is charged in this mode. As shown in Figure 5, when the battery voltage reached a predetermined value, the CC mode changed to the RCC mode. As mentioned earlier, in the RCC mode, the charging current

changed from 100 to 90 amps, and this reduction in current continued to keep the battery voltage constant. This created more space for charging the battery.

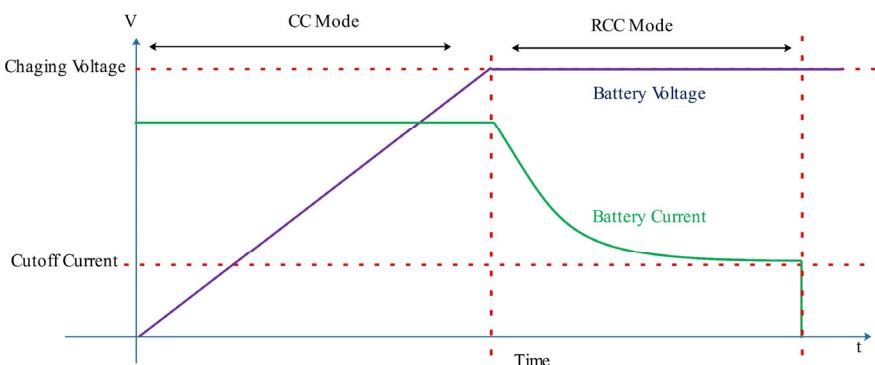

**Figure 5.** Current and voltage waveforms in the proposed CC/RCC are a charging scheme.

When the battery voltage reached more than the specified value, the battery current decreased again, and if it reached 90 to 80 amps, the battery voltage decreased slightly. This process was repeated until the remaining SOC of the battery was charged, and the current reached zero, which completed the charging process. The CC/RCC, charge control strategy could safely charge the battery without the battery voltage rising above the set maximum value, and it reduced the charging time more than the CV method. In addition, in this method, a common PI controller was used in the CC and RCC modes, whereas, in previous methods, two levels of closed-loop control were used, which required more PI controllers. EV loads could be modeled as a constant power load, current load, or impedance load. Constant power indicates that the EV load has the greatest effect on the network power; the effects of a constant current and then a constant impedance are shown in the article [32]. EVs based on constant power are a simple model, but they represent the worst-case scenario in load modeling. However, most of the time, they are likely to be intelligent and adaptable to network conditions. The constant impedance load drew different powers from the network according to the network conditions. When the bus voltage level was high, it could draw more power from the network, so it was better to use the constant bar impedance in this project. In order to model EVs as constant impedance, the two current references, the CC and RCC modes, were related to the voltage bus, which was linear.

### 2.3. Two-Way Power Transmission

Figure 6 shows the two buses, one and two, connected with impedance $Z\angle\gamma = R + jX$. Each bus had an ideal voltage source defined as $V\_1\angle\delta\_1$ and $V\_2\angle\delta\_2$. Additionally, Figure 6 illustrates the transfer of active power ($P$) and reactive power ($Q$) between the two buses, assuming that both buses could feed and fetch $P$ and $Q$.

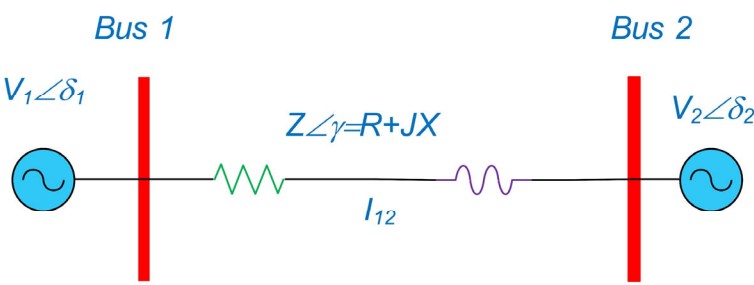

**Figure 6.** Overview of the connecting two buses.

If the current shown by $I_{12}$ was maintained in the circuit, *P* and *Q* could be seen from the following equations:

$$P_{12} = \frac{V_1^2}{Z}\cos\gamma - \frac{V_1 V_2}{Z}\cos(\gamma + \delta_1 - \delta_2) \tag{1}$$

$$Q_{12} = \frac{V_1^2}{Z}\sin\gamma - \frac{V_1 V_2}{Z}\sin(\gamma + \delta_1 - \delta_2) \tag{2}$$

Equations (1) and (2) could be simplified by using the assumption $X \gg R$.

$$P_{12} = \frac{V_1 V_2}{X}\sin(\delta_1 - \delta_2) \tag{3}$$

$$Q_{12} = \frac{V_1}{X}[V_1 - V_2\cos(\delta_1 - \delta_2) \tag{4}$$

In Equations (3) and (4), the voltage amplitudes ($V_1$ and $V_2$) and voltage angles ($\delta_1$ and $\delta_2$) could be noticed. Changes in the voltage angle mostly affected the active power, while changes in the voltage range affected the reactive power. Table 1 shows the power transfer conditions between the two buses.

**Table 1.** Power transfer conditions.

| Condition | Power Transfer | |
|---|---|---|
| $\delta_1 > \delta_2$ | Active power of bass | Transfer from two to one |
| $\delta_1 < \delta_2$ | Active power of bass | Transfer from one to two |
| $V_1 > V_2$ | Reactive power of bass | Transfer from two to one |
| $V_1 < V_2$ | Reactive power of bass | Transfer from one to two |

*2.4. EV Battery Modeling*

Accurate battery modeling was needed to study the power system connected to the EVs. The simplest model was the ideal voltage source connected to internal resistance. However, not showing the state of charge (SOC) was the drawback of this model. The electrochemical behavior of the battery was described as an equation to develop the presented mode in terms of the SOC, terminals voltage, open circuit voltage, internal resistance, and discharge current. In order to validate this, the obtained model was compared with the real battery's datasheet, and a 90% accuracy was confirmed. In this project, the battery was modeled based on [20], a controlled voltage source in series with internal resistance. The controlled voltage source was described through the lithium battery charging equation, which can be seen below:

$$V_{batt} = E_0 - R \cdot i - K\frac{Q}{it - 0.1Q}i^* - K\frac{Q}{Q - it} + Ae^{-B.u} \tag{5}$$

Where is the battery voltage, the constant voltage of the battery, the internal resistance, the battery current, the polarization constant in ohms, the battery capacity in ampere-hours, the actual battery charge in ampere-hours, the filtered current, the amplitude of the exponential part, and the time constant of the exponential part? Based on the above equation, when the SOC of the battery reached 90%, the voltage fluctuations of the battery became apparent. Because t = 0 started when the battery was fully charged at t = 8 s with t = 25 s, the lithium batteries had a high dynamic response time. The lithium-ion battery's minimum and maximum actual charging times were approximately 50 min, assuming that every second was equal to approximately 2 min. Therefore, the above formula could be rewritten as follows (Figure 7):

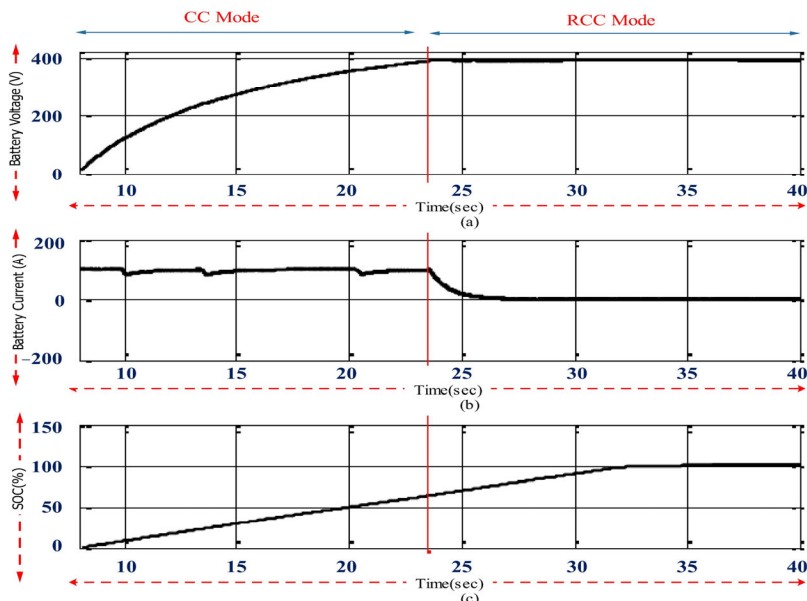

**Figure 7.** Battery charging waveforms in the charging method RCC/CC: (**a**) battery voltage; (**b**) battery current; (**c**) battery charging status with SOC.

$$V_{batt} = E_0 - R.i - K\frac{Q}{it - Q}i^* - K\frac{Q}{Q - it} + Ae^{-B.u} \tag{6}$$

Some parameters of the above equation could be deduced from the battery discharge. Three important data were needed to calculate the parameters of the above equation, which were the full charge voltage ($V_{full}$), the capacity and voltage at the end of the exponential area ($Q_{exp}, V_{exp}$), and the capacity and voltage at the end of the apparent area (nominal). According to the battery model and its discharge curve, these data are shown in Table 2.

**Table 2.** Extracted values of the discharge curve.

| Parameters | Lithium Battery 360 Volt and 66.2201 Amp/h |
|---|---|
| $V_{full}$ (V) | 403.2643 |
| $Q_{exp}$ (Ah) | 6.004 |
| $V_{exp}$ (V) | 384.864 |
| $Q_{nom}$ (Ah) | 58.32 |
| $V_{nom}$ (V) | 345.62 |

The values of battery parameters could be determined from the above equation. The internal resistance $R$ was assumed constant and equal to 0.05 Ohm. The equations for determining the values of $E_0$, $B$, $A$, and $K$ were obtained as follows:

$$A = V_{full} - V_{exp} = 384.2643 - 384.864 \tag{7}$$

$$B = \frac{3.03}{Q_{exp}} = \frac{3.03}{6.004} \cong 0.5543(Ah)^{-1} \tag{8}$$

$$E_0 = V_{full} - A + R.i = 403.2643 - 19.542 + (0.05343) \times (66.32444) = 388.65 \tag{9}$$

$$K = (E_0 - V_{nom} - R.i + Ae^{\frac{-3}{Q_{exp}}Q_{exp}})(\frac{(Q - Q_{nom})}{Q(Q_{nom} + i)}$$

$$= \left(387.324 - 345.675 - 0.0564 \times 66.435 + 19.65e^{\frac{-3\times 58}{Q_{exp}}Q_{exp}}\right) \tag{10}$$

$$\times \left(\frac{(66.34334 - 58.4345)}{66.34334(58.4543 + 66.34334)}\right) = 0.0383.656$$

Figure 8 shows the characteristics of a real battery with high accuracy.

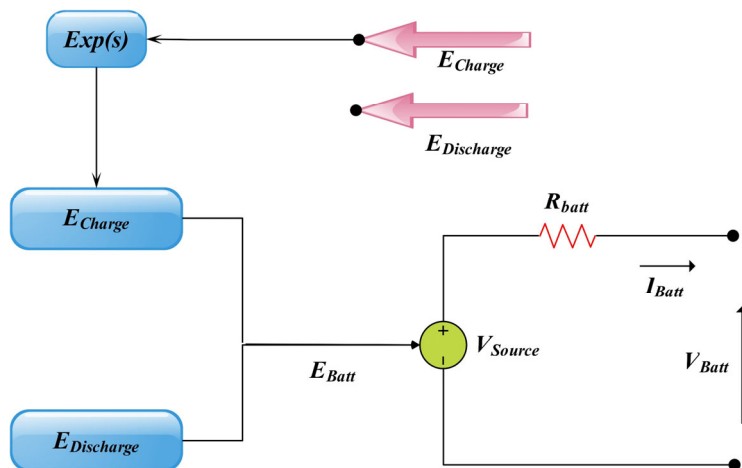

**Figure 8.** Lithium-ion battery simulator model.

Sine Pulse Width Modulation

In modulation, $V_{mB}, V_{mA}$, and $V_{mC}$ were the three-phase sinusoidal modulation waves, and $V_{Cr}$ was the triangular carrier waves.

The amplitude modulation index $m_a$ could control the main component of the output voltage range of the inverter.

$$m_a = \frac{V_m}{V_{Cr}} \tag{11}$$

where $V_{mo}$ and $V_{Cr}$ are the peak values of modulation and carrier waves, respectively. The amplitude modulation index, $m_a$, is adjusted by changing $V_m$ and keeping the amplitude $V_{Cr}$ constant. The frequency modulation index was also defined as follows:

$$m_f = \frac{f_{cr}}{f_m} \tag{12}$$

where $f_m$ and $f_{cr}$ are the modulation and carrier wave frequencies, respectively.

## 3. Proposed Method

### 3.1. Paralleling Synchronous Generators with the Network and with Each Other

According to Figure 9, generators can be used in two ways. The first way is to connect the generator to the load, feed it, and work independently from the network, which is used for small places and often as emergency power. Where high power is required, a parallel connection of several generators is used to produce electricity independent of the grid, or the generator is parallel with the grid to inject power.

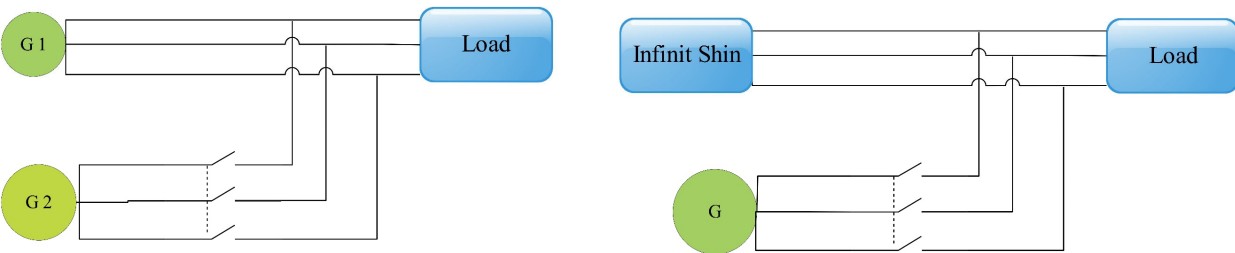

(**a**) Paralleling two independent generators.　　　(**b**) Paralleling the generator with the grid.

**Figure 9.** The proposed method for fast charging. (**a**) Paralleling two independent generators. (**b**) Paralleling the generator with the grid.

The proposed method for fast charging stations was a multistep constant current method, and the goal was to achieve charging in the shortest time, with the maximum voltage, while the current of each step was no less than the previous step. It should be noted that the fast-charging stations are placed at level three of charging, with a maximum output power of 240 kW and a maximum current of 400 A.

In this problem, five flow steps were used, which could be expressed as follows:

$$Minimize\ t_{charge}$$
$$Where\ I_j^i = [I_1^i, I_2^i, I_3^i, I_4^i] \qquad (13)$$
$$Subject\ to\ V_{batt} < V_{max}$$

$$I_j^i > I_k^i \quad if\ j < k\ \ j,k = 1,2,3\ ....5 \qquad (14)$$

Where is the charging time? What are the current levels in the step? What is the battery voltage during charging? What is the maximum allowed voltage of the battery while charging? The charging process chart is shown in Figure 10. According to the flowchart below, checked and identified the battery type and SOC in the first step of the flowchart block. In the second step, the proposed algorithm was executed. In the third step, the charging algorithm was executed. In the next step, the discharge status of the battery was checked, and the objective function calculated using the algorithm was processed in the next step. Finally, in the next step, if the convergence condition of the algorithm was met, the charging algorithm was changed, i.e., the charging time and level were changed, and it would return to the previous four steps according to the flowchart. Additionally, this loop was repeated until the condition of convergence was achieved.

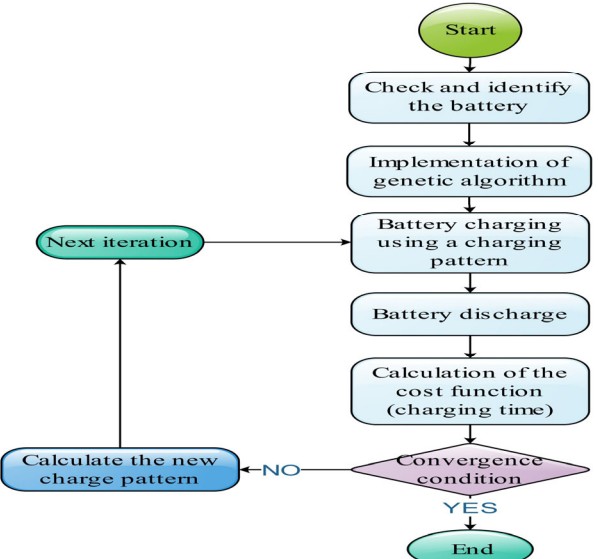

**Figure 10.** Proposed charge management flowchart based on a genetic algorithm.

In the multistep constant current method, due to the low current level in the last steps, the charging time increased so that a third of the charging time corresponded to the final 20% of the charge. In the current pulse method, to fully charge the battery, the width of the pulses should have been reduced to prevent overvoltage, and the rest period should have been increased, which would have reduced the charging speed. It should be noted that, in addition to the battery, modern electric cars have a supercapacitor, which can inject more current into the engine to increase acceleration. Supercapacitors have high power and low energy, which means they can inject much current but discharge quickly; they charge within a few seconds. This feature can be used in emergencies, and in the multi-bridge constant current method, the internal supercapacitor performs the final steps of charging while moving, thus, reducing the charging time. Figure 11 shows a comparison between different fast charging methods. The multi-bridge constant current method had a higher speed than the other two methods, so with an initial capacity of 10%, the battery was charged to more than 90% of its capacity in 15 min. It took approximately 21 and 23 min to charge up to 90% capacity using the pulse current and reflection methods. In the last two methods, because the battery voltage had reached its maximum value, the pulses' size and width had to be changed for full charging, which increased the charging time.

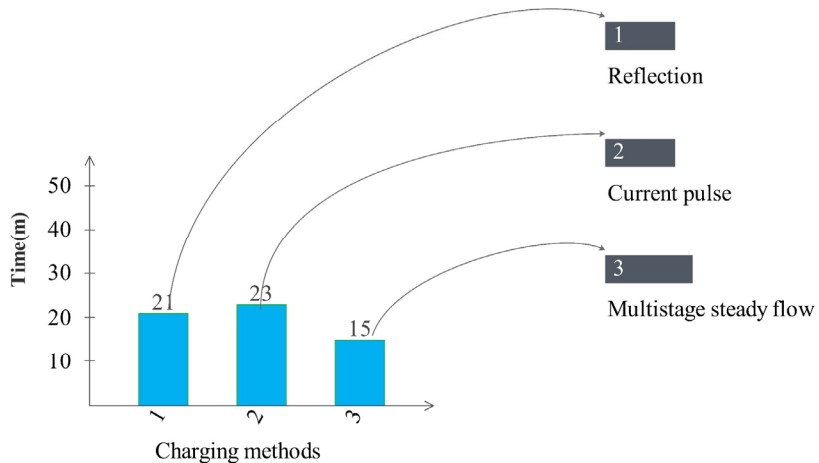

**Figure 11.** Comparison of charging time in fast charging methods.

### 3.2. Proposed System Modeling

In this section, a fast charging system for electric vehicles with a bidirectional reactive power compensation control strategy was presented, in which AC/DC converters were used for a two-way reactive power exchange to maintain the DC link voltage and the network bus voltage at their base values. A fast charging station for electric vehicle batteries was connected to six empty lithium-ion batteries at the farthest feeder. DC/DC converters were used to charge the EV systems with a certain voltage level and CC/RCC control strategy. Finally, a local diesel generator charged the batteries in two modes: connected to the grid (on-grid) and separated from the grid (off-grid). Here, with a diesel generator and without a controlled AC/DC converter, the DC link voltage was maintained at the base value; the microgrid was in island mode, and without a connection to the national power grid, it met the needs of the cars. All modes, including the controlled AC/DC converter, uncontrolled diode AC/DC converter, and diesel generator without a controlled AC/DC converter, were simulated in different modes with MATLAB software.

Figure 12 shows the single-line diagram of the studied network. This system was a radial network; the error level in the 132 kV bus was 11 kV, equaling 16.3 ka, showing the network's relatively strong (high short-circuit power). The maximum loading on both sides of the 132/33 kV transformers was equal to 28 Mw. The source voltage of 132 kV reached the average level of 33 kV using three-phase step-down transformers and was

converted to 11 kV. The coupling switches of the main 33 kV and 11 kV buses were normally closed to increase the network's reliability. They were usually set at higher voltages to compensate for the voltage drops in long cables. Therefore, the 33 and 11 kV buses in this network were 2.73% higher than the nominal voltage, i.e., 33.9 and 11.3 kV.

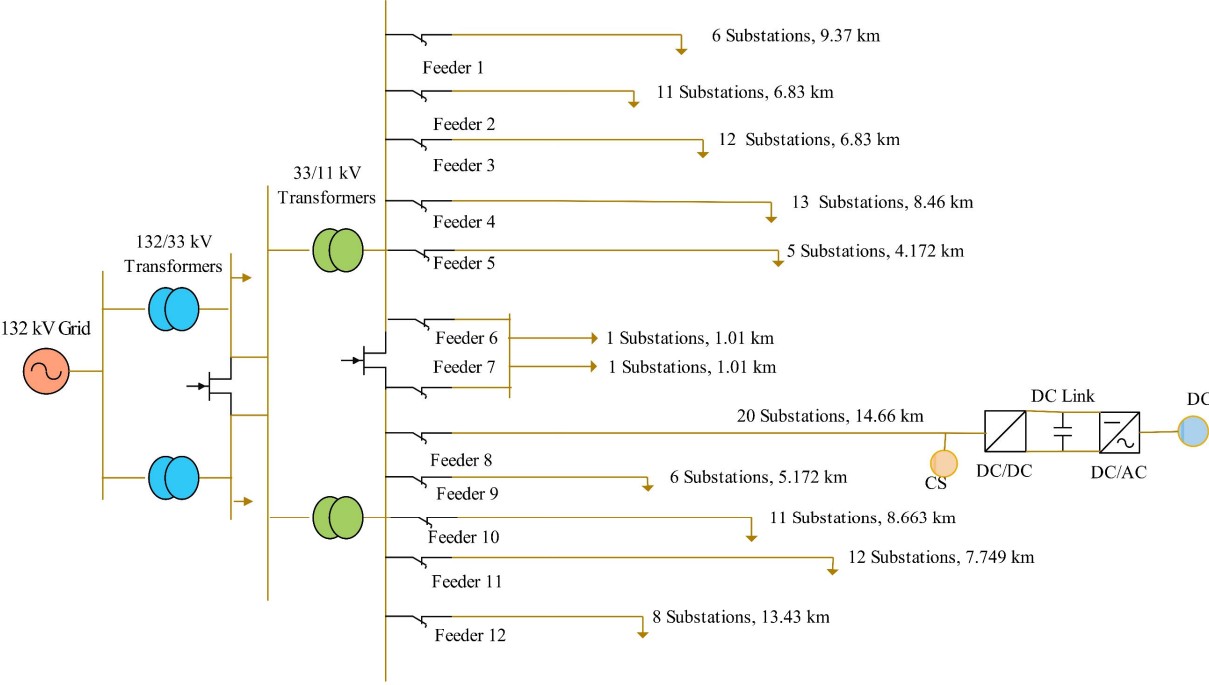

**Figure 12.** Single-line diagram of the studied network.

There were 12 feeders connected to the 11 kV distribution surface and 106 distribution posts connected to these 12 feeders. The number of posts and the length of each feeder's cables are shown in Figure 12. Each of the 106 distribution posts had a step-down transformer of 11.04 kV, which fed loads at a low level.

## 4. Simulation of Bidirectional Reactive Power Exchange in the Studied Network

In this article, an AC/DC converter was used to compensate for reactive power, which could adjust the voltage of the distribution network during the charging process of EVs. The control strategy of this converter was described in the previous section. The simulation shown in Figure 13a,b depicts the reactive power exchange between the distribution network and the EV system. The reactive power exchange was based on the voltage range difference between the grid bus voltage and the EV converter system voltage. The positive reactive power passed from the EV converter to the distribution network.

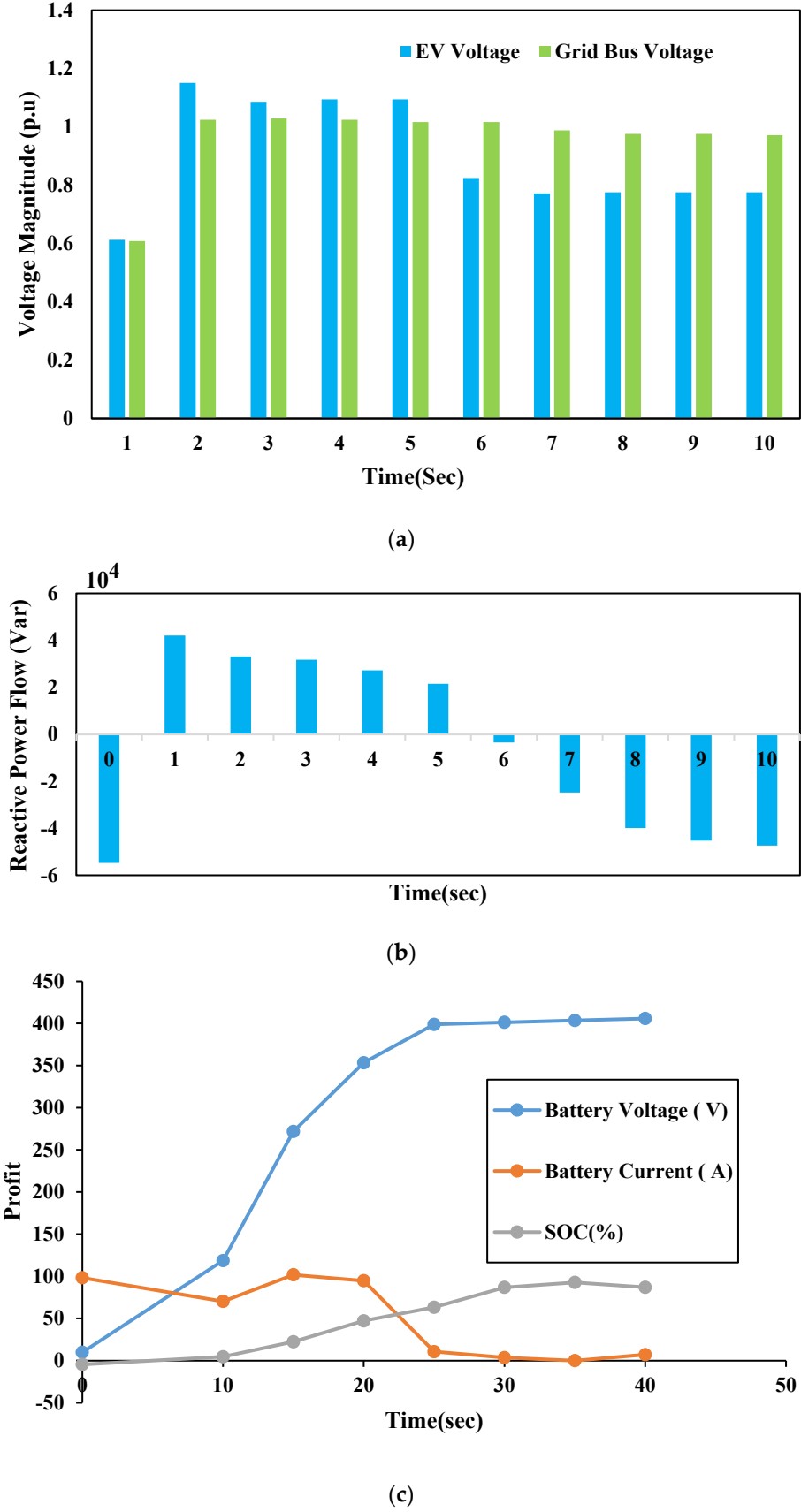

(**a**)

(**b**)

(**c**)

**Figure 13.** Reactive power exchange and battery charging (**a**) the voltage difference between the EV system and the grid. (**b**) Reactive power exchange in grid (**c**) Battery charging waveforms in the CC/RCC are charging methods.

When the output voltage range of the converter was greater than the network bus voltage range, the reactive power passed from the EV converter to the distribution network. Additionally, according to Figure 13c, when the output voltage range of the converter was lower than the network voltage range, then the reactive power flowed from the distribution network to the EV converter, in which case, the reactive power sign would be negative.

### 4.1. Analysis and Simulation of CC/RCC Charging Modes with DC/DC Converters

The DC/DC converter was used to implement the fast charging control strategy of the EV batteries. The constant current/constant voltage (CC/CV) charge control method was used based on two important processes for the quick charging lithium-ion batteries in EVs. At first, the battery was charged through a constant high current, and most batteries were charged in this short period (approximately 80% of the capacity in 30 min of charging). When the battery voltage reached a predetermined value, the CC state changed to the CV state. The battery was charged with a decreasing current until the charging voltage remained constant. Although the CV mode was used to charge the remaining 20% of the battery capacity, it took a long time—roughly three times as long as the CC mode.

This study used a control strategy for the DC/DC converters called the constant current/constant reverse current (CC/RCC) method. During the charging process, the battery voltage was continuously measured. Then, it was compared with a predetermined value, *Vmax* = 392 volts, and remained constant at this value. The reason for choosing a preset value compared to the actual maximum battery voltage of 403.2 volts was to create an additional safety margin. In the CC mode, the output DC current of the DC/DC converter, *Idc*, was compared with a reference DC current, which was set to approximately 1.5 C (100 A). When the battery voltage hit a specific level in the RCC mode, the DC reference current range shrank, and the measured output DC current of the DC/DC converter (*Idcc*) contrasted with the shrinking DC reference current.

The current and voltage waveforms with the CC/RCC charging approach obtained from the simulation are shown in Figure 13c. As can be seen, for safety reasons, the battery voltage was kept below 392 volts during the entire charging process. In the CC mode, for fast EV battery charging, a high charging current of approximately 100 amps was pulled from the distribution network, and the battery voltage increased rapidly during the charging process. According to Figure 13c, the CC mode changed to the RCC mode when the battery voltage reached a predetermined value. At the start of the RCC mode, the charging current was reduced from 100 to 90 amps, leading to a slight drop in the battery voltage. When the battery voltage reached the predetermined value again, the charging current was further reduced from 90 to 80 amps, in which case, the battery voltage dropped slightly. This situation continued until the end of charging when the charging current reached zero. The CC/RCC, charge control method could charge with high safety without increasing the maximum battery voltage. It could also significantly reduce the charging time compared to the common CV mode. It should be noted that the simulation time of this part started at t = 8 s due to the high dynamic response time of the battery, and it was fully charged at t = 32 s according to the state of the diagram or the SO. Of course, the maximum real-time for charging a lithium-ion battery is approximately 50 min, whereas each simulation second equals approximately 2 min in the real world.

### 4.2. Simulation and Analysis of Bidirectional Reactive Power Exchange

The study of the power system under simulation, whose single-line diagram was shown in Figure 14, basically investigated the effect of EV fast charging on voltage drops. The proposed EV system had an AC/DC voltage source converter and a DC/DC voltage source converter for each EV. All DC/DC converters were connected in parallel with a common DC bus, where the DC link voltage was regulated using the AC/DC converter. An RL filter was placed between the EV system and the low-voltage distribution network to remove harmonics. The depth of discharging a lithium-ion battery is approximately 70–

80%. Therefore, as this study was conducted for the worst conditions, the EVs were assumed to be fully discharged and be at the farthest post of the distribution network with low-voltage and maximum load conditions.

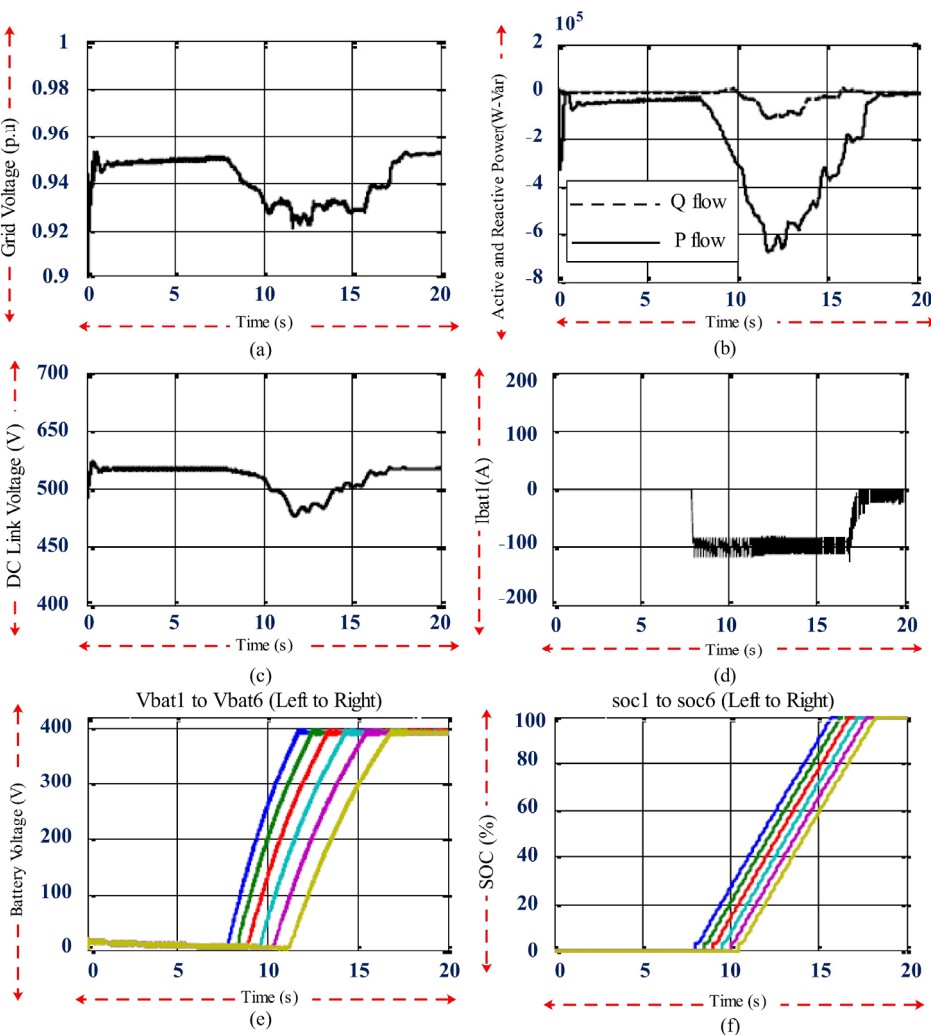

**Figure 14.** Fast charging results for EVs without reactive power compensation control. (**a**) Bus voltage; (**b**) exchange of active and reactive power; (**c**) link voltage; (**d**) battery current; (**e**) battery voltage in the SOC; (**f**) batteries.

The fast charging control, DC link voltage control, and reactive power compensation control to adjust the voltage and modify the power factor were used through the simulation in the MATLAB software environment. This software could not complete an EV charging process in more than one hour due to the limitation of simulation time; as a result, it was assumed that each second in the simulation was equal to two minutes in real-time.

### 4.2.1. Fast Charging of EVs without Reactive Power Compensation

This part investigated the effect of fast charging six fully discharged EV systems at the farthest post of the low-voltage distribution network during maximum loading and without reactive power compensation.

When the power system was in a steady state, the first EV was connected to the grid at t = 8 s, and the next ones were connected to the power grid at the same time interval and with a delay of 5.0 s. The second EV entered the network at t = 5.8 s and the third at t = 9 s, etc., following the same procedure. Figure 14 shows the full simulation results for

the fast EV charging without reactive power compensation. Figure 14a show the bus voltage of the farthest substation to which the studied EV system was connected. Before t = 8 s, the system was in a steady state, and the bus voltage was approximately 0.96 per unit, equivalent to 384 volts. Although the studied network had the highest error level, the main buses were set at voltage values of 33.9 and 11.3 kV to compensate for the voltage drop of the feeder cables.

There was a significant voltage drop on the longest feeder, with a length of 15 km in the farthest post. Accordingly, connecting EVs to the network resulted in a step voltage drop, which then gradually recovered. The voltage drop recovery was due to the EVs in the RCC mode receiving less power from the distribution network. The 400 V low-voltage distribution system had a voltage tolerance of +0.1% and −4%. When the sixth EV was connected to the network at time t = 10.5 s, the bus voltage at the farthest post dropped below 0.94 kV, which was lower than the minimum acceptable working voltage level. Therefore, fast EV charging would have a significant impact on the performance of the existing network. Connecting only six EVs to the grid would cause the voltage to move out of its safe range. Figure 14b shows the exchange of active (*P*) and reactive (*Q*) power, where positive power meant power passing from the EV system to the distribution network, and power with a negative sign meant power passing from the distribution network to the EV system.

Before t = 8 s, when the grid was in a steady state, no active or reactive power was exchanged between the EV system and the distribution grid. When the EVs were connected, each pulled approximately 33 kW of power from the grid. This obtained power was used for the fast charging of the EV batteries and was the main cause of the bus voltage drop shown in Figure 14a. Each EV would absorb the reactive power once successfully connected to the grid.

When the EVs were charged in the RCC mode or fully charged, they received little reactive power from the grid, whose recovery waveform was shown after t = 10.5 s.

The DC link voltage between the AC/DC converter and the DC/DC converters is shown in Figure 14c. In this design, the AC/DC converter was modeled as a full-wave diode bridge rectifier, which had no control over the DC link voltage, which caused the waveform of the DC link voltage not to be constant but to follow the waveform of the network voltage.

Figure 14d shows the charging current of the first EV. Before t = 8 s, there was no current, but in the CC mode, the current pulled from the network was 100 amps in the fast-charging mode. In the RCC mode, the pulled current gradually decreased and reached zero, and the charging process was completed. The waveform of the charging current related to other EVs was similar.

Figure 14e shows the battery voltage of all six EVs. At the beginning of the charging process, the battery voltage increased rapidly. Then, the battery voltage gradually increased during the charging process in the rated area. The battery voltage of all six EVs was kept under 392 volts with the CC/RCC charge control mode.

Figure 14f shows the battery SOC of all six EV drives. If the EVs were charged with 100 amps, the SOC of the batteries would reach 80% in the CC mode within 30 min. When the charging current decreased compared to the ideal charging current due to the lower grid voltage, the SOC of the batteries in the CC mode reached only 70% within 30 min, and the rest of the SOC of the battery increased in the RCC mode. The entire charging process of each EV took approximately 50 min.

4.2.2. Fast Charging of EVs with Reactive Power Compensation

The conditions and settings used, similar to the previous mode, were also used for this mode. Here, the effect of the fast charging of six fully discharged EVs connected to the farthest low-voltage distribution post with maximum loading was investigated. The reactive power compensation controller and the DC link voltage controller switched the controllable AC/DC converter to achieve a constant voltage of the bus and DC link. Two

scenarios were considered: in the first case, the bus voltage was set at the steady state value of 0.96 per unit; in the second case, the bus voltage was set at the nominal value of 1 per unit.

(A) Voltage adjustment for steady state voltage of 0.96 p.u

Figure 15a shows the bus voltage of the farthest post to which the EV system was connected with bidirectional control. Figure 15b shows the power exchange between the distribution network and the EV system. Before t = 8 s, the distribution system was steady, and the bus voltage was 0.96 per unit. No power was exchanged between the EV system and the distribution network in this case. At t = 8 s, the first EV was connected to the farthest distribution substation. This EV drew approximately 33 kilowatts of power from the grid to quickly charge its batteries. Figure 15b confirms that when the reactive power controller started working, the amount of reactive power was injected from the DC link capacitor of the AC/DC converter to the distribution post to set the bus voltage at the steady state voltage level. After connecting other EVs, more active power would be drawn from the grid after a fixed delay. As a result, the reactive power controller achieved the amount of reactive power injection required to adjust the network bus voltage. When the EVs were charged in the RCC mode, they received less real power; therefore, less reactive power was injected into the network. When the distribution substation bus voltage was maintained at its steady state value, 0.96 per unit, no violation would occur in the working voltage of the network. By establishing the switching of the power converter modules, the reactive power controller could solve the problem of the voltage drop caused by the EV's fast charging.

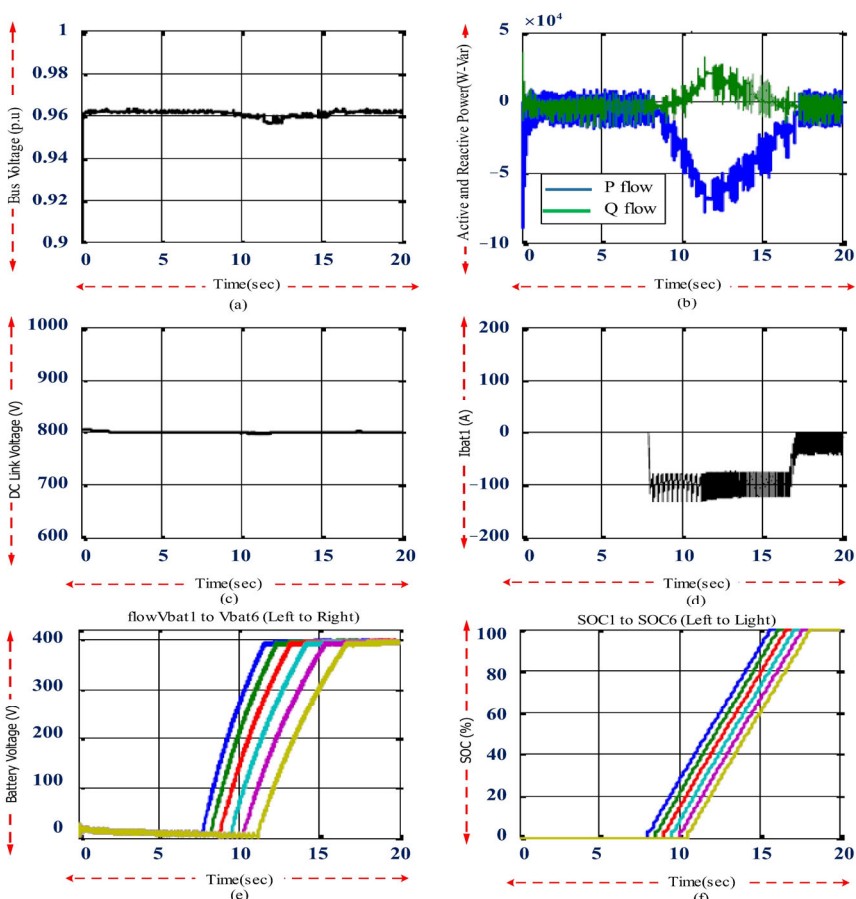

**Figure 15.** The results of fast charging of EVs with reactive power compensation control at bus voltage of 0.96 p.u. (**a**) Bass voltage; (**b**) exchange of active and reactive power; (**c**) link voltage; (**d**) battery current; (**e**) battery voltage; (**f**) SOC.

Figure 15c shows the DC link voltage between the AC/DC converter and the DC/DC converters. It can be seen that the DC link voltage was maintained at 800 volts.

At the moment of connecting each EV to the charge, a small fluctuation in the DC link voltage occurred. Therefore, the DC link voltage controller could reliably and quickly restore the DC link voltage to the desired value. The charging current of the first EV is shown in Figure 15d. This EV was charged at approximately 90 amps in the CC mode and had a much lower current level in the RCC mode. Other EVs had similar charging current waveforms.

Figure 16e shows the battery voltage of all six EVs. It can be seen that the voltage of all batteries was maintained below 392 volts. The effect of the CC/RCC charging method was well shown in this scenario.

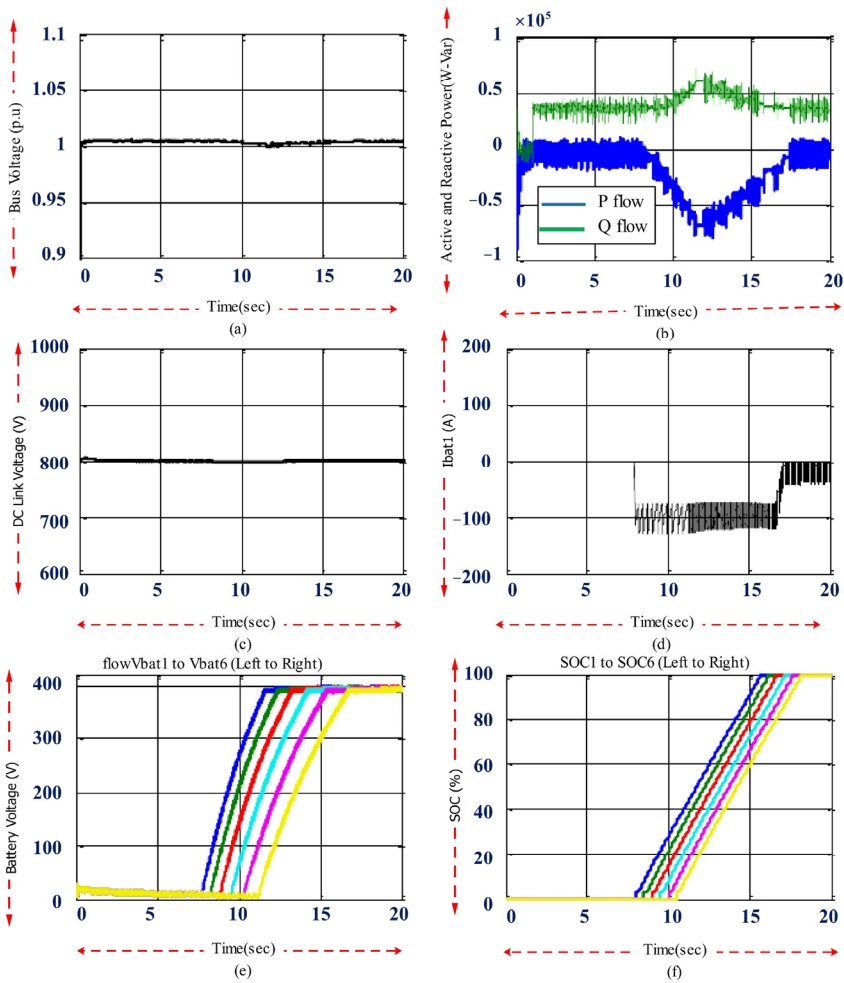

**Figure 16.** EV fast charging results with reactive power compensation control at a bus voltage of 0.96 p.u. (**a**) Bus voltage; (**b**) active and reactive power exchange; (**c**) DC link voltage; (**d**) battery current; (**e**) battery voltage; (**f**) SOC.

Figure 16f shows the SOC level of each EV. During charging, the fully discharged EVs received a charge from the grid, which increased the SOC level. The SOC of batteries in the CC mode could reach 75% within 30 min, which showed that the fast-charging process was faster than the case without reactive power control. This was because the application of voltage control allowed the bus voltage to be set at a steady state value of 0.96 per unit. The entire charging process for each EV took approximately 45 min.

(B)   Voltage adjustment for the nominal voltage of 1 unit

This part discussed the study of the proposed scenario in setting the nominal voltage of 1 p.u. Figure 16a, shows the network bus voltage at the farthest distribution post to which the proposed EV system was connected. The exchange of active and reactive power between the distribution network and the EV system is shown in Figure 16b. Before the time t = 1 s, this system worked in stable conditions, and the bus voltage was 0.96 per unit, equivalent to 384 volts, and no power was exchanged between the EV system and the distribution network. The reactive power controller t = 1 s started working. The controllable switching of the AC/DC converter of the proposed EV system caused approximately 40 kVAR of reactive power to be injected into the distribution substation through the DC link capacitor.

At the moment of connecting each EV to the charge, a small fluctuation in the DC link voltage occurred. Therefore, the DC link voltage controller could reliably and quickly restore the DC link voltage to the desired value.

The charging current of the first EV is shown in Figure 16d. This EV was charged at approximately 90 amps in the CC mode and had a much lower current level in the RCC mode. Other EVs had similar charging current waveforms.

Figure 16e shows the battery voltage of all six EVs. It can be seen that the voltage of all batteries was maintained below the preset value of 392 volts. The effect of the CC/RCC charging method was well shown in this design.

Figure 16f shows the SOC level of each EV. During charging, the fully discharged EVs received a charge from the network, which increased the SOC level. The SOC of batteries in the CC mode could reach 75% within 30 min, which showed that the fast-charging process was faster than in the case without reactive power control. This was because the application of voltage control allowed the bus voltage to be set at a steady state value of 0.96 per unit. The entire charging process for each EV took approximately 45 min.

(C)   Simulation and analysis of the proposed plan

In this part of the article, a proposed technique was presented to investigate the fast charging of electric vehicles (EV) in a microgrid with the help of distributed generation (DG), a diesel generator with a PID controller, and automatic voltage regulation. The specifications of the mentioned diesel generator are presented in Table 3. This diesel was connected to the DC link through a full-wave uncontrolled diode AC/DC converter and produced the necessary power to charge the EV batteries according to different situations. It could operate as an independent microgrid or be connected to the distribution network. Since unwanted errors and planned or unplanned outages could cause interruptions in the power transmission to EV fast charging systems, it was necessary to have an emergency energy source that could charge the EVs under optimal conditions. This study examined the ability to charge EVs using a diesel generator in three different modes. In the first case, the microgrid was isolated from the distribution network, and the batteries were fed only with the diesel generator. In the second case, it was possible to charge the EVs through the distribution network and the diesel generator simultaneously. In the third state, the microgrid was connected to the distribution network, and the diesel generator was in standby mode. As soon as the network could not provide a charge for the EVs, it came into the circuit.

**Table 3.** Specifications of diesel generator of the proposed design.

| Generator Specification | Values |
|---|---|
| Nominated Demand and Power Factor | 200 kVA, 0.85 lag |
| Voltage and Nominated Frequency | 440 V, 50 Hz |
| H (Inertia Constant) | 24 s |
| Number of Poles | 4 |
| $Xd, Xd', Xd''$ | 1.0305, 0.296, 0.252 (p.u) |
| $Xq, Xq'', Xl$ | 0.474, 0.243, 0.18 (p.u) |
| $Td', Td'', Tqo''$ | 1.01, 0.053, 0.1 (s) |

### 4.3. Autonomous Microgrid Simulation for Charging EV Batteries

In this part, the simulation and analysis of the state of the microgrid were discussed, where the only source of energy supply was a diesel generator with the specifications listed in Table 2, and the microgrid had no connection with the distribution network. The diesel generator was modeled as a PV bus with a constant voltage and power. The diesel generator was in the working mode under nominal voltage and frequency.

At time t = 8 s, the first EV entered the circuit, and the other EVs were placed in charging mode after 0.5 s. Figure 17 shows the simulation related to EV charging with an independent DG separated from the grid. Figure 17a shows the diesel generator voltage. After the t = 8 s, the generator's voltage decreased noticeably from the steady state value of 1 per unit. Due to the high inertia of the rotor, it remained at the same value of 0.95 per unit even after the charging was completed. The only disadvantage of this phenomenon was reducing the charging current and lengthening the charging time in the CC mode.

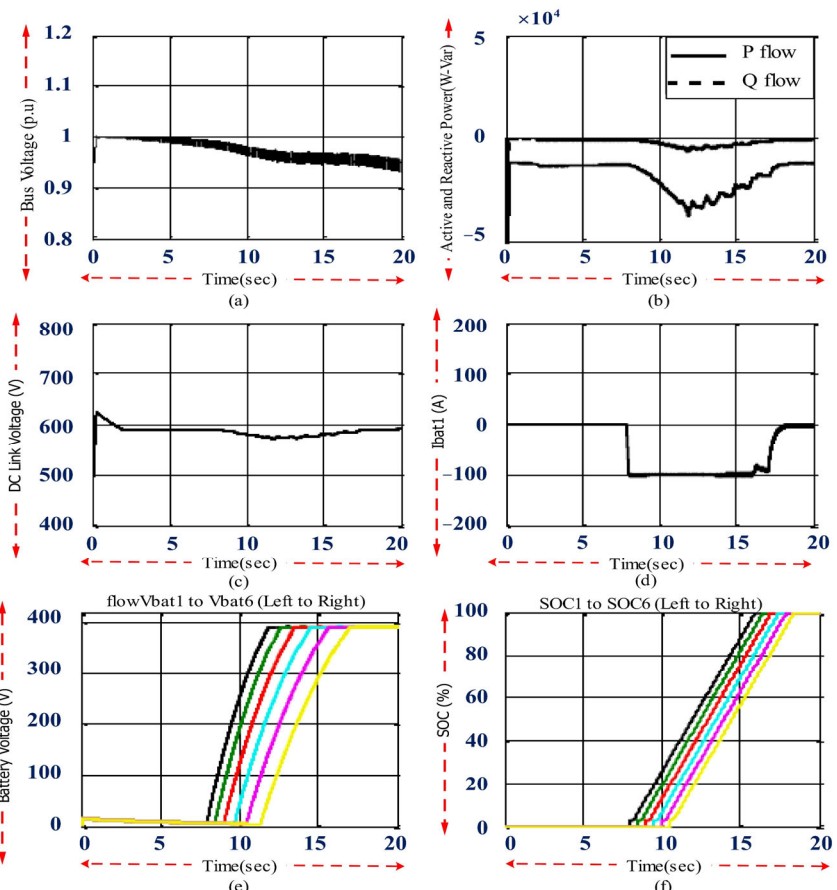

**Figure 17.** Fast charging results of EVs in microgrid and stand-alone and off-grid conditions (**a**) generator voltage; (**b**) active and reactive power exchange; (**c**) DC link voltage; (**d**) battery current; (**e**) battery voltage; (**f**) SOC.

Figure 17b shows the active and reactive power exchange. Before t = 8 s, the basis of the active power was not 0, but approximately 1 kW. This non-zero value was the existence of other necessary uses other than charging EVs, but the reactive power was based on zero. After t = 8 s, the EV charging was started, and some active and reactive power was injected into the EV system to charge the DC link.

Figure 17c shows the DC link voltage at contrasting times. As soon as EVs started fast charging, the DC link voltage dropped significantly since there was no control over this voltage. As a result, its value decreased with the increase in EV loading. The

maximum DC link drop compared to its reference voltage was approximately 3% in the worst case, which could be suitable under the current conditions.

Figure 17d–f shows the charging current, voltage, and charging status. Figure 17d shows the charging current of the first EV; according to the investigations carried out in the previous stages, the charging current in the CC mode was approximately 100 amps. Figure 17e shows the voltage of all six EVs, all of which remained below the predetermined value of 392 V. Finally, Figure 17f shows the state of charge (SOC) of the batteries, which were fully charged in approximately 50 min.

The behavior of the diesel generator is shown in Figure 18. Figure 18a shows the angular velocity of the rotor. Here, the speed of the generator at the moment t = 8 s was 1.1 per unit; after t = 8 s and until the end of the 10% charging time, its speed was increased to 1.2 per unit. The reason for not reducing the speed after charging the EVs was the high inertia of the diesel generator, which could not reduce the speed immediately after the change in its electromechanical torque. Figure 18b shows the DC excitation voltage of the synchronous generator, which had a value of approximately 1 per unit while charging the EVs. The changes in the diesel electromechanical torque are shown in Figure 18c. By connecting the first EV to the charge at t = 8 s, the electromechanical torque of the diesel is also increased; at the maximum load and t = 12 s, it reached approximately 0.4 per unit, and with the step reduction in the charging of the EVs, the torque value also decreased.

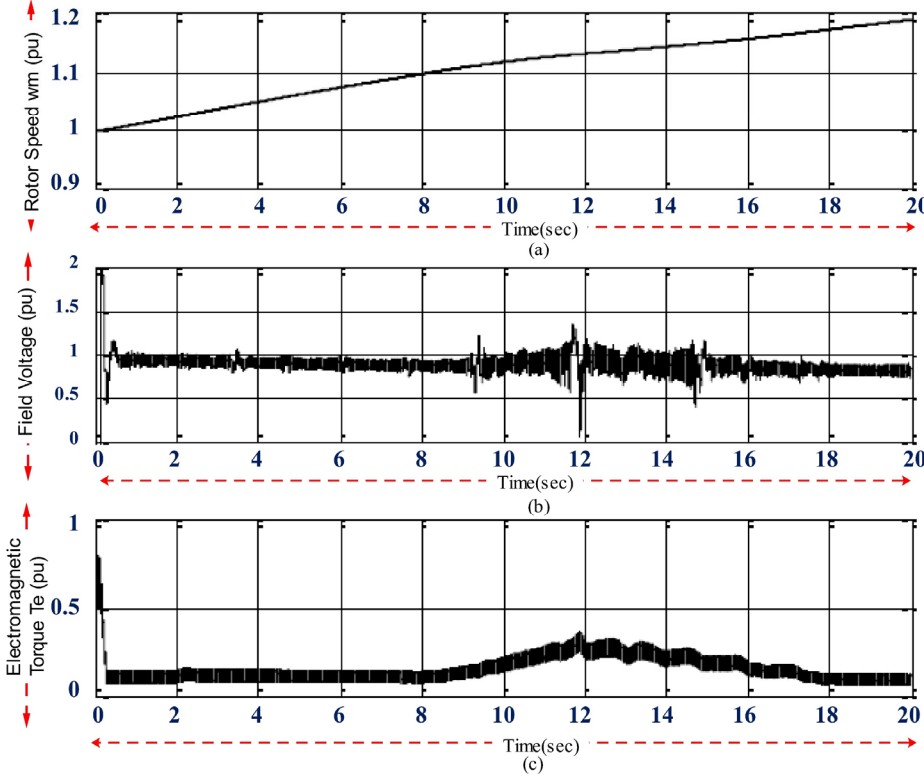

**Figure 18.** Diesel generator operation in a microgrid separated from the distribution network (off-grid); (**a**) angular speed; (**b**) stimulation voltage; (**c**) DC electromechanical torque.

### 4.4. Simulating the Microgrid Connected to a Distribution Network and a Diesel Generator

As seen in the previous section, with the increase in EV loading and especially when all six completely discharged EVs were connected to the distribution network simultaneously, much active power flowed from the network to the DC link. This would cause a noticeable drop in the network bus voltage when the reactive power compensation controller was not used. In this proposed method, the diesel generator proposed in the previous section was used to maintain the network bus voltage at its nominal value. In this

situation, the low-voltage distribution substation and the diesel generator simultaneously provided the power required to charge the EVs and maintain the DC link voltage. The connection of both sources to the DC link was through a full-wave diode bridge rectifier. Figure 19 shows the simulation of this case.

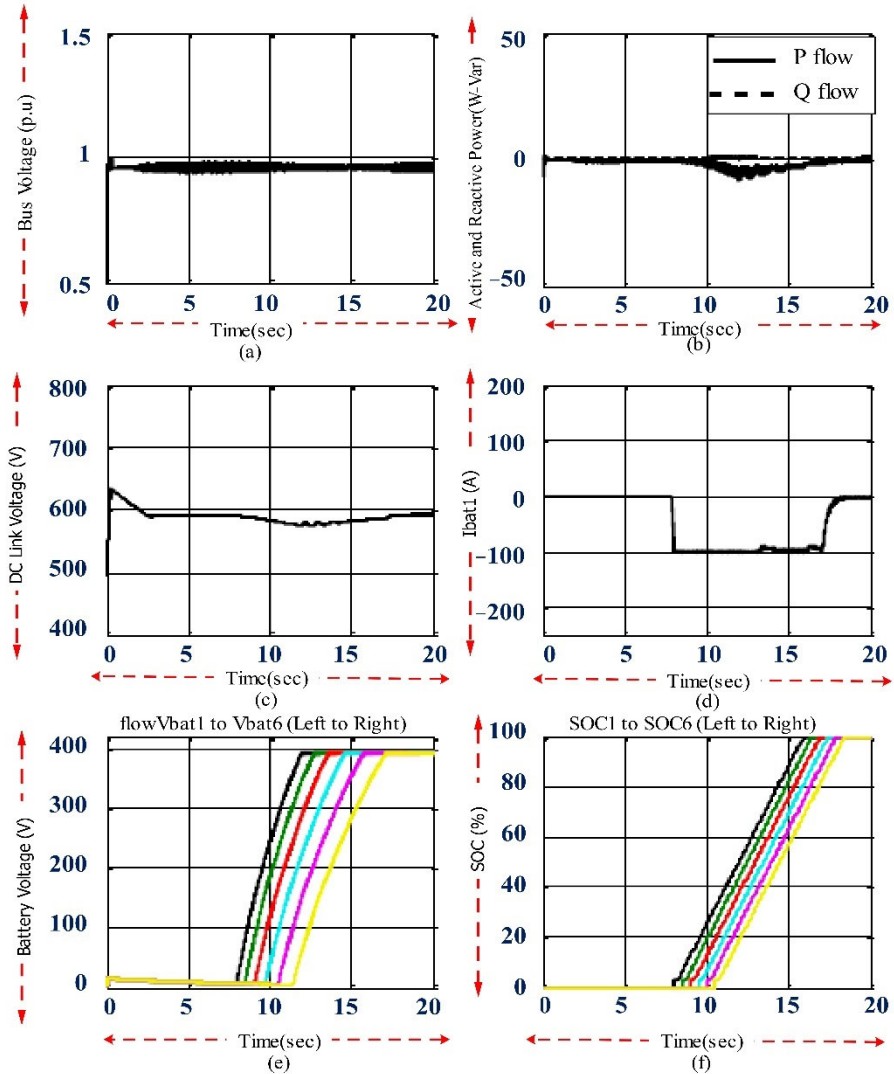

**Figure 19.** The results of fast charging of EVs connected to the grid and diesel (**a**) grid bus voltage; (**b**) active and reactive power exchange of the grid bus with the EV system; (**c**) DC link voltage; (**d**) battery current; (**e**) battery voltage; (**f**) SOC.

Figure 19a shows the network bus voltage, kept at 0.96 units. The reason was the insignificant participation of the distribution network in providing active power for EV fast charging. Figure 19b shows the active and reactive powers exchanged by the network bus and the EV system, whose values were very small and zeroed at the maximum load of EVs.

Figure 19c shows the voltage of the DC link. Despite the simultaneous participation of the distribution network and the diesel generator, a voltage drop of approximately 3% was observed in the maximum load of EVs. The reason for this was the small participation of the distribution network and the maximum participation of the diesel generator so that the bus voltage of the network remained constant within the permissible value.

Figures 19d–f show the charge current, battery voltage, and state of charge (SOC) of the batteries. Figure 19d shows the charge current of the first EV. According to the investigation carried out in the previous stage, its charge current in the CC mode was

approximately 100 amps. Figure 19e shows the voltage of all six EVs, all of which remained below the preset value of 392 V. Finally, Figure 19f shows the state of charge (SOC) of the batteries, fully charged within approximately 50 min. This part was very similar to the independent mode.

The behavior of the diesel generator is shown in Figure 20a shows the generator bus voltage, which displayed a decreasing trend during charging. The reason for this, as mentioned in the previous section, was the high inertia of the rotor. Figure 20a shows the participation of the diesel in providing active and reactive power. Unlike the grid bus, which did not play any role in supplying the EV system, the diesel generator could provide all the necessary power to the EVs. In the worst conditions, the diesel could deliver up to 50 kW of active power and approximately 5 kVAR of reactive power to the EV system.

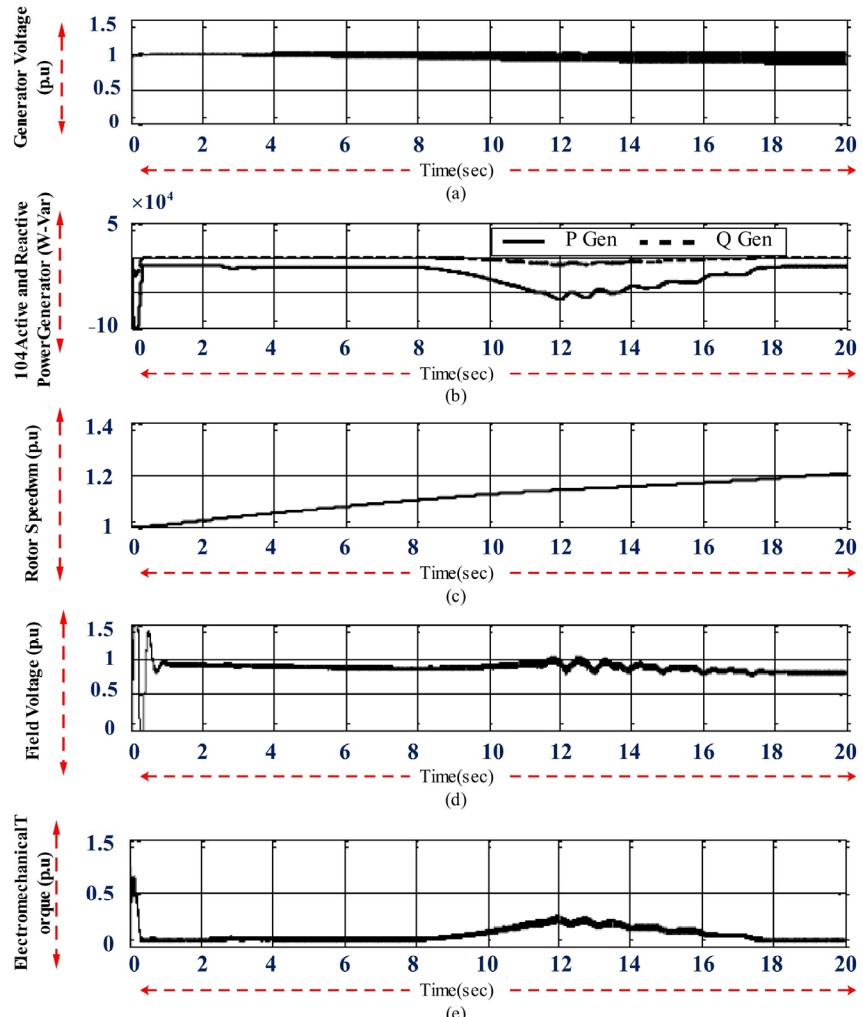

**Figure 20.** The behavior of the diesel generator in the microgrid. (**a**) Generator bus voltage; (**b**) active and reactive power exchange of the generator with the EV system; (**c**) angular speed of the generator; (**d**) excitation voltage; (**e**) electromechanical torque.

Figure 20c shows the angular velocity of the rotor. In this mode, similar to the independent mode, the generator speed at t = 8 s was 1.1 per unit; after t = 8 s until the end of the 10% charging time, its speed increased to 1.2 per unit. The reason for not reducing the speed after charging the EVs was the high inertia of the diesel generator, which could not reduce the speed immediately after the change in its electromechanical torque.

Figure 20d shows the DC excitation voltage of the synchronous generator, which had a value of approximately1 per unit during EV charging, and fluctuated a bit during maximum loading.

The changes in the diesel electromechanical torque were also shown in Figure 20e. By connecting the first EV at t = 8 s, the electromechanical torque of the diesel also increased and reached approximately 0.3 per unit at maximum load and t = 12 s. With the gradual reduction in EV charging, the torque also decreased.

The application of this method had high reliability due to the simultaneous supply of microgrid energy from two energy sources (the distribution network and diesel generator) compared to the independent method. In case of the failure of either source, the other one could supply the required voltage. The disadvantage of this system was that it could not be effective in loading maximum EVs in the mode of connection to the grid and without a diesel generator. Because it was in the farthest distribution post, results similar to those in the previous section were obtained, which was unsatisfactory.

*4.5. Simulation of Microgrid Connected to the Distribution Network and Diesel Generator Selectively*

This part discussed the simulation study of the microgrid connected to the distribution network and the diesel generator, discussed selectively and alternatively. Under these and normal conditions, the microgrid was only connected to the distribution network, and the diesel generator was in an unloaded and standby mode. In order to improve the bus voltage of the network and maintain it at the allowed value, an effective value detector took samples from the bus at every moment and compared them with a reference value, which was 0.94 per unit in this simulation.

At the maximum load of EVs and when the network bus voltage drops below the reference value according to the DC link voltage, the diesel generator immediately enters the circuit using a two-position switch, and the network bus is taken out of the circuit. Consequently, the network bus voltage would return to the nominal state, and the DC link voltage would be restored to the permitted value. After restoring the normal conditions, the generator was disconnected from the circuit, and the distribution network fed the consumers.

Figure 21 shows the simulation related to this mode for the network bus. In Figure 21a, it can be seen that prior to the maximum loading through the EVs, before t = 10 s, the network bus voltage was in its permitted interval. However, at t = 10 s, when the bus voltage dropped below the allowed reference value (0.94 per unit), immediately, after approximately 100 milliseconds, when the diesel generator entered, and the network bus left the circuit, the bus voltage was again maintained at its nominal value.

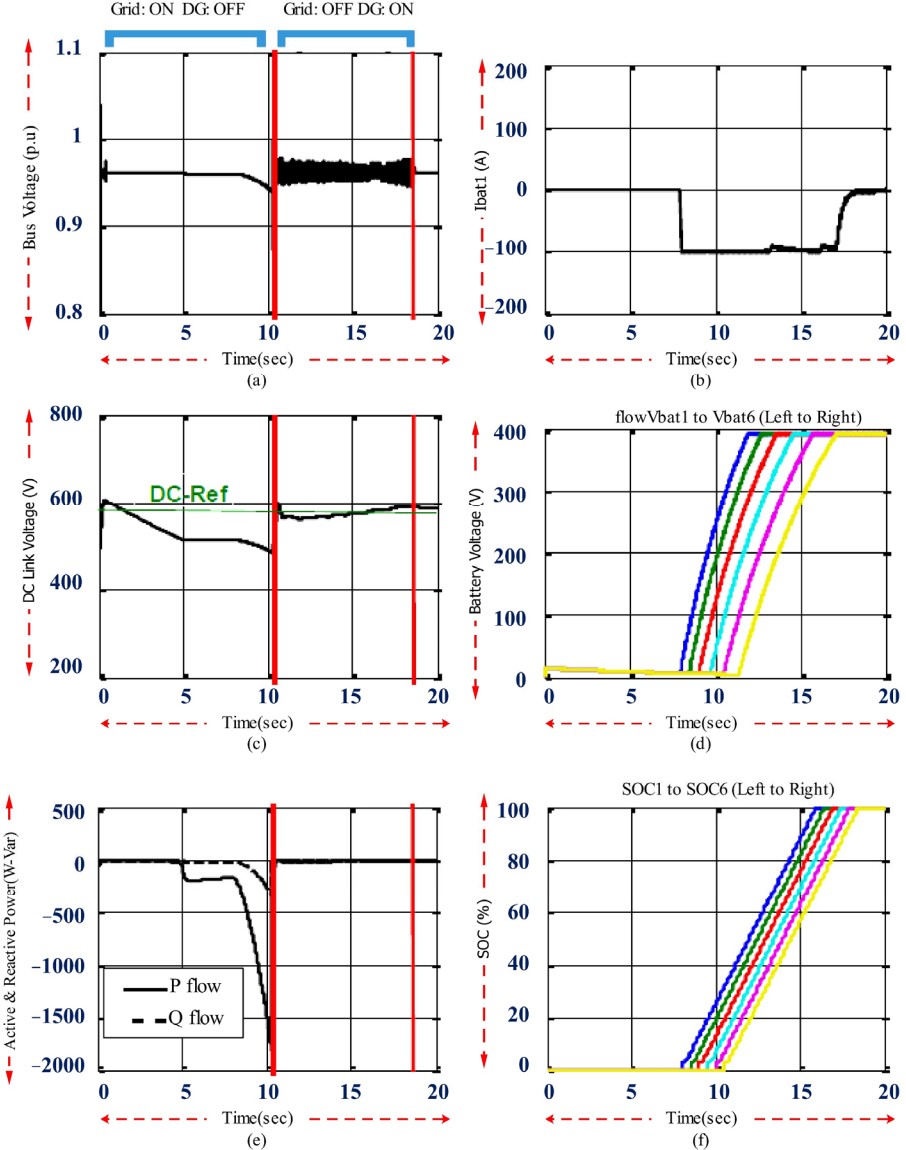

**Figure 21.** The results of fast charging of EVs in the microgrid in the state of connection to the grid and selected diesel (**a**) network bus voltage; (**b**) exchange of active and reactive power of the network bus with the EV system; (**c**) DC link voltage; (**d**) battery current; (**e**) voltage of batteries; (**f**) SOC.

Figure 21b is related to the DC link voltage. It can be seen that before t = 10 s, the DC link voltage range decreased rapidly, approximately 17% less than its reference value. After t = 10 s and connecting the diesel to the DC link voltage circuit, it was restored and placed around the base value.

Figure 21c shows the active and reactive power exchanged between the distribution network and the EV system. Until t = 10 s, the total power required for the EV system was from the network. Immediately after t = 10 s, the power was exchanged through the network, the EV system was zero, and it was supplied through the diesel generator.

Figure 21d–f show the batteries' charging current, voltage, and state of charge (SOC).

Figure 21e shows the voltage of all six EVs that remained below the preset value of 392 V. Finally, and partly shows the state of charge (SOC) of the batteries, all fully charged in approximately 50 min. This part was very similar to the stand-alone mode.

It should be noted that changing the mode from the network bus to the diesel generator did not cause any disturbance in the charging of the EV batteries.

The behavior of the diesel generator is shown in Figure 22. The diesel generator was only used in the interval between t = 10 s and t = 18 s, and as soon as the network bus voltage and the DC link voltage were restored, it was disconnected again. Figure 22a shows the voltage of the diesel generator, which demonstrated a decreasing trend due to the high inertia of the rotor, and it took a long time to reach a stable state.

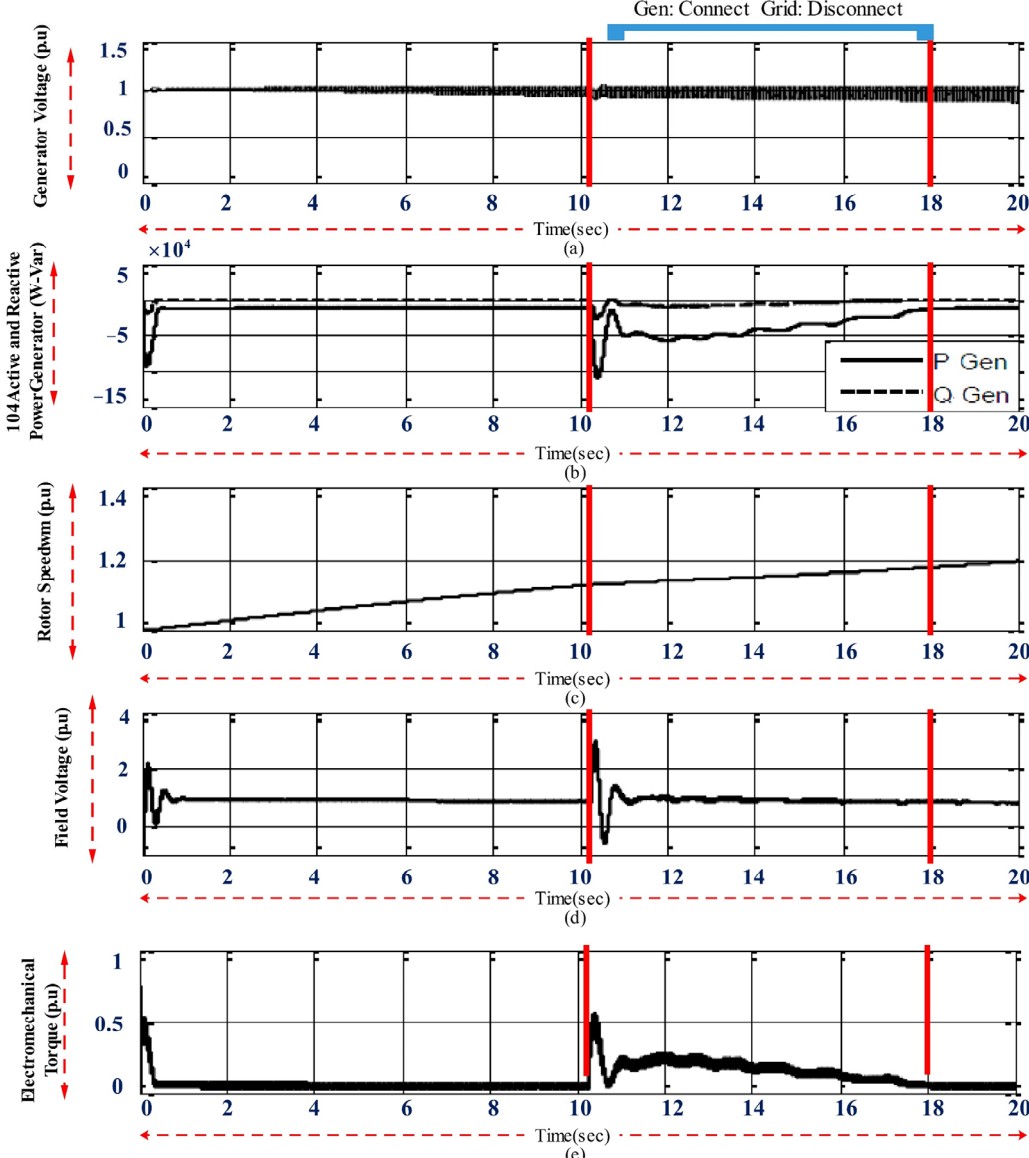

**Figure 22.** The behavior of the diesel generator in the microgrid in the state of connection to the grid and selected diesel (on-grid)—(**a**) generator bus voltage; (**b**) active and reactive power exchange of the generator with the system; (**c**) angular speed of the generator; (**d**) excitation voltage; (**e**) electromechanical torque.

Figure 20b shows the power exchanged using the diesel generator between 10 and 18 s. At the beginning of this interval, a jump between active and reactive powers was observed, which was caused by the switching. In order to maintain the DC link voltage at the reference value, approximately 50 kW of active power was injected into the EV system at maximum load.

Figure 20c shows the angular velocity of the rotor. In this mode, it can be seen that the rotor's speed increased when the generator was connected to the grid, similar to the independent mode.

Figure 20d shows the DC excitation voltage of the synchronous generator, which fluctuated during the maximum load and after the diesel entered the circuit due to the switching action, which was resolved in a few seconds.

The changes in the diesel electromechanical torque were also shown in Figure 20e. It can be seen that after the diesel generator was connected, the electromechanical torque of the diesel also increased with a few initial oscillations and reached approximately 0.3 per unit at the maximum load and t = 12 s with the gradual reduction in charging EVs and the restoration of the network bus voltage. The torque was also reduced, and the diesel was removed from the circuit.

## 5. Conclusions and Future Work

### 5.1. Conclusion

This study presented and simulated a proposed design for an intelligent control method for electric vehicle charging in microgrids (MGs). The proposed plan was studied and reviewed in three cases. In the first case, an independent diesel generator provided the power needed to fast-charge EVs in an MG. In the second case, a distribution network and diesel generator were used simultaneously to provide the necessary power to charge EVs in an MG. Finally, in the third case, a distribution network bus was used to charge EVs, and when the network bus dropped, a diesel generator was switched on, and the MG was disconnected from the distribution network. This study's results indicated that the EV charging system negatively affected the distribution network. The intensity of the effects caused due to the charging of the EVs. It depended on their connection point in the power grid. The CC/RCCC charging control method could charge with high safety without increasing the battery's maximum voltage. It could also significantly reduce the charging time compared to the common CV mode.

Additionally, with the two-way reactive power controller, the network bus voltage could be kept constant at the allowed value at the maximum loading of EVs. This was possible through the absorption of reactive power from the DC link. The presented approach had very high reliability compared to the independent method, and if one source had an outage, a different source could still provide the necessary voltage.

### 5.2. Future Work

1. The design of a power controller is suggested for the diesel generator so that with any change in the DC link voltage range, the diesel can maintain the link voltage by injecting sufficient power in the reference value.
2. To improve the microgrid's efficiency and increase its reliability, it is suggested to use other scattered products such as wind turbines, solar modules, and fuel cells.
3. In the method of the selective feeding of the diesel and network bus, the basis for changing the source from the network bus to the diesel generator and vice versa is the voltage range of the DC link (in this study, the basis for changing and switching sources to each other was the voltage drop in the distribution network bus).

**Author Contributions:** S.R.: Data curation, software, visualization, resources, Z.M.: Writing—review & editing, writing—original draft, project administration, M.A.N.: Data curation, software, resources, project administration; M.Z.: Data curation, software, supervision, methodology, S.P.: Writing—review & editing, writing—original draft, supervision. All authors have read and agreed to the published version of the manuscript.

**Funding:** This research received no external funding.

**Data Availability Statement:** Not applicable.

**Conflicts of Interest:** The authors declare no conflict of interest.

## Nomenclature and Abbreviation

**Nomenclature**

| | | | |
|---|---|---|---|
| I(t) | EV charging current | EV | Electric vehicle |
| $Q^t$ | Rated capacity of ith EV battery in Ah | BMS | Battery management system |
| $N_{EV}$ | Number of charging EVs in the time slot | NLP | Nonlinear programming problem |
| C | Charging rate of EVs | CC/RCC | Constant current/constant reverse current |
| gen | Power of generation of PV system in parking lots | PCA | Principal component analysis |
| t | Time | IOT | Internet of the things |
| I | The line connected to the bass | RFID | Radiofrequency identification |
| cost (.) | Upstream energy costs | DL | Deep learning |
| $p^{max}, p^{min}$ | Maximum and minimum active power output of the upstream network | ML | Machine learning |
| $q^{max}, q^{min}$ | Maximum and minimum reactive power produced by the upstream network | BD | Big data |
| $v^{max}, v^{min}$ | Maximum and minimum voltage | LSTM | Long short-term memory |
| N | Number of network buses | R | Resiliency |
| Connect(…..) | Connecting electric vehicles to the network | IPV | Internet protocol version |
| $C_{in}$(…..) | The initial charge level of the electric car when entering the parking lot | MG | Microgrid |
| B(….) | Suspension | EV | Electric vehicle |
| G(….) | Capacity | DG | Distributed generation |
| Lin(…) | Network line | EMS | Energy management strategy |
| qD | Reactive load | SAG | Stand-alone grid |
| | | CS | Charging station |
| $P_D$ | Active time | LIB | Lithium-ion battery |
| BC | Battery capacity | MPPT | Maximum power point tracking |
| CR | Electric car battery charge rate | PV | Photovoltaic |
| $C_{Total}$ | Total cost | PHEV | Plug-in hybrid electric vehicle |
| $PL_{Total}$ | Total wasted power | DES | Distributed energy resource |
| $V_R$(…) | The real part of the voltage | PWM | Pulse width modulation |
| $V_I$(….) | The imaginary part of the voltage | DAB | Dual active bridge |
| $I_R$(…) | The real part of the flow | STC | Standard test condition |
| $I_I$(….) | The imaginary part of the flow | BMS | Battery management system |
| $P_{loss}$(.) | Lost active power | SOC | State of charge |
| $Q_{loss}$(.) | Lost active power | MIDC | Measurement and instrumentation Data center |
| $N_{loss}$(.) | Total network losses per hour | OB | Off-board |
| $TN_{loss}$ | Total network losses | RE | Renewable energy |
| CB | Capacity of battery | PCA | Principal component analysis |

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
