# Peer review of "Using an Intelligent Control Method for Electric Vehicle Charging in Microgrids"

_wevj, doi:10.3390/wevj13120222_

Round 1

Reviewer 1 Report

Dear authors,

I appreciate your efforts involved in this research work. The proposed control method developed for electrical vehicles charging in the microgrids is an exciting approach with a promising potential for significant practical applicability in the automotive industry. Due to its practicability perspective, the topic captures the attention of a large community of readers and specialists working in the field. Increasing the overall quality of the manuscript leaves enough room for improvement. Perhaps, the following suggestions and comments could be helpful for you:

1. In the Introduction section, based on the more recent information from the literature, try to underline the performance obtained using the proposed control method compared to the results obtained by different approaches developed in the literature field.

2. For good readability of the manuscript's text, please insert a table with all the acronyms and abbreviations used in the text. 

3. An algorithmic approach (Flowcharts, algorithms' steps) for the proposed control method will be welcome for MATLAB implementers.

4. Please, offer some arguments to convince the implementers that your proposed control method is intelligent. 

5. For readers is helpful to define the SOC of the proposed Li-ion battery. Adding a separate equation representing the SOC dynamics to the battery dynamic model described in equation (6) will be very useful for implementers. 

6. Representing the Li-ion battery discharge curve graphically and showing how we can extract the parameters of the battery model given in equation (6) is also very useful for MATLAB implementers. 

7. On page 15, line 24 is mentioned that "a proposed technique is presented to investigate the fast charging of electric vehicles (EV) in a microgrid with the help of distributed generation (DG), a diesel generator with a PID controller, and automatic voltage regulation". It will also be helpful for implementers if you give them some details on the PID controller design, the optimal values of its parameters and step response performance. 

8. The robustness analysis of the proposed control method can also be challenging. 

Thanks,

Author Response

Reviewer #1

I appreciate your efforts involved in this research work. The proposed control method developed for electrical vehicles charging in the microgrids is an exciting approach with a promising potential for significant practical applicability in the automotive industry. Due to its practicability perspective, the topic captures the attention of a large community of readers and specialists working in the field. Increasing the overall quality of the manuscript leaves enough room for improvement. Perhaps, the following suggestions and comments could be helpful for you:

Please be advised that all the newly added/revised texts are marked in green in the revised version of this paper.

Comment 1:

  1. In the Introduction section, based on the more recent information from the literature, try to underline the performance obtained using the proposed control method compared to the results obtained by different approaches developed in the literature field.

Response Thank you for the helpful comment.

The clarity of the Introduction  mentioned by the Honorable reviewer has been improved.

  1. Introduction:

Road traffic is known as one of the main causes of greenhouse gas emissions. Along with rising fuel prices and the issue of high energy efficiency, electric cars will be more widespread in the next decades. To meet the needs of long trips, electric cars are more desirable devices. An electric car uses batteries and a combustion engine to minimize fuel consumption. A plug-in hybrid electric vehicle is charged when plugged into a home charger or a public charging station. However, this causes challenges in the electrical microgrid system. With the high penetration of hybrid electric vehicles, it will be added to the current peak load and will create a new peak load. This can cause voltage fluctuation and transformer overload. Voltage deviations can cause damage to electrical appliances, while persistent overload can cause transformer overheating, which will result in a blackout. Fortunately, the development of advanced measurement systems and communication systems enables us to develop better algorithms to overcome these problems; Therefore, the timing and speed of charging hybrid electric vehicles can be controlled to reduce the maximum load, which is called load demand management. In addition, numerous researches have been conducted in various fields such as micro-grids, energy storage systems, renewable energy and electric vehicles (EV) to exploit power systems in a distributed manner. Among all these, Evs have attracted the most attention because if their expansion continues, they will become the most effective solution available in the future. In addition, since Evs can be considered as a kind of mobile battery in power systems, power system operators such as micro-grids should consider the accuracy of EVs' uncertainties; Therefore, Evs are studied considering their uncertainty patterns. In particular, issues such as how Evs work and how they affect distribution systems, such as micro-grids and smart grids, have been studied.

In this section, studies have been conducted in the field of load demand management and plug-in hybrid electric vehicles. Source [3] presents a hierarchical control algorithm to understand the trade-off and cooperation between plug-in hybrid electric vehicle charging and wind power. The three-level controller proposed in this resource uses plug-in hybrid electric vehicles to compensate for wind power fluctuations and thus indirectly adjusts the grid frequency. So this type of connection between electric car and wind leads to the preservation of environmental resources and the use of clean energy sources. Reference [4] deals with the problem of bridging the gap by controlling a large population of plug-in hybrid electric vehicles. It also presents a developed decentralized algorithm, but it only proves the optimal state in the homogeneous state where all hybrid electric vehicles have the same departure time, energy requirement, and maximum charging power. In reference [5] a control signal from the supplier company is modified to drive and update profiles of plug-in hybrid electric vehicles. This algorithm converges with optimal charging profiles in homogeneous and heterogeneous cases. However, if the communication between the company and the plug-in hybrid electric vehicle is asynchronous, the performance of the algorithm can be greatly affected.

In reference [6], increasing or decreasing the maximum load is used to reach the desired waveform of the load, taking into account the preferences of the consumer and the characteristics of the load. A multi-agent system solution has been accepted in [7], which has the feature of adaptability and high scalability. The paper [8] provides a comprehensive review on the issues of plug-in hybrid electric vehicles.

In the following, we will describe the article [9], in this article, the issue of load response in intelligent systems with the presence of electric vehicles is discussed. We focus on the impact of hybrid electric vehicle charging on low voltage transformers (LVT). The goal is to smooth the load curve of each LVT and at the same time meet each of the customer's requirements for plug-in hybrid electric vehicles that must be charged at the required level at a given time. In the following, we will first introduce the dynamic model of plug-in hybrid electric vehicles. Finally, the DSM problem for plug-in hybrid electric vehicles is formulated as an optimization problem.

The latest articles in the field of the present article include the research stated in references [22, 21], in reference [21] a methodology for scheduling the charging and discharging process of Evs and PHEVs was proposed. Production, consumption, mobility of Evs and market price uncertainties are considered. The purpose of these articles is to balance the system based on market prices. Market prices are used as a reference for charging and discharging schedules of Evs to obtain load leveling. In fact, with this methodology, it is possible to obtain a good but not optimal solution. The rules used to model the constraints of the problem are only considered in one period (15 minutes) in each set of rules; Therefore, the car charging process considers the needs of travel and not the needs of the system. In addition, the technical limitations of the network are not considered.

In reference [22], the valley removal and peaking methodology considers V2G. This system considers two levels of control, i.e. the control center of smart networks and the V2G control center. The smart grid control center sends control signals to the V2G control centers based on "optimal energy distribution", and the V2G control centers are responsible for controlling the V2G charging and discharging process. The predicted load and hourly usage goals that the user is exposed to are considered. The objective function minimizes the differences between the target and the actual power demand.

The article [20] analyzes the impact of V2G on the system operation costs and on the power demand curve for the distribution network with the high influence of distributed energy sources. This effect has been analyzed from different points of view: minimizing the operation cost and minimizing the difference between the minimum and maximum demand during 24 hours a day. The multi-objective problem is presented considering the minimization of the operation cost and the optimization of the power demand curve. In the proposed methodology, the effect of the number of different Evs in the distribution network is analyzed considering the vehicle characteristics and application characteristics, which is in accordance with the report of the US Transportation Agency.

The aim of optimizing the power demand curve is to increase the minimum load consumption or decrease the maximum load consumption, which leads to a power demand curve close to a rectangular shape. The electric power demand curve is evaluated by analyzing the value of the load factor. The load factor related to the total energy consumption is obtained by the peak consumption in a certain time horizon.

The article [23] has investigated the integration of the electric vehicle connected to the PEV electricity – with the EMS energy management system – which serves as a first step in understanding its potential role in the energy resources of local semi-autonomous groups and micro-grids. In their past research, many authors have discussed new power electronics technologies that accompany various DER-CAM energy sources, especially variable frequency AC-DC sources, solar panels, batteries, and asynchronous generators such as micro-turbines. High-speed switching disconnects and reconnects to the integrated network. These power electronics will be able to form a microgrid that operates semi-autonomously from the traditional centralized power line. Also, this paper analyzes the integration of PEV in the building energy management system EMS-and the difference between vehicle-to-microgrids V2G and vehicle-to-microgrid. These relationships are modeled using the DER_CAM energy resource distribution criteria approval model, which provides optimal combinations of equipment required by the microgrid with the lowest cost and carbon footprint or other criteria. DER_CAM model is an optimization tool that minimizes annual energy costs for microgrid. For an office building using the PEV connection under the business model where the distributed threshold values ​​for the maximum payment are obtained, it is concluded that the economic effects have their own limitations. However, this paper shows that some economic benefits are created due to the avoidance of charging demand and TOU rates. The strategy adopted by the office building to reduce costs is to use PEU batteries in the afternoon hours. The results obtained will vary depending on the case. Of course, the results of CO2 removal are not presented in this article.

In the paper [24], a modified particle swarm optimization (PSO) technique is presented and demonstrated to solve the problem of managing highly influential energy resources from distributed generation and pluggable (V2G) electric devices. The purpose of reducing and minimizing operation costs, i.e. energy costs, especially the issue of managing these resources, is to get a smart network. The reforms used for PSO are aimed at improving and promoting its competence and suitability to solve the mentioned problems. In other words, this presented article is an evolution of the traditional particle swarm optimization called ASMPSO, which is used for the problem of energy resource management in smart grids. It considers and addresses realistic grids using energy sources such as distributed generation (DG) based on renewable energy sources and plug-in electric vehicles (EV) (V2G). Accurate AC load spreading and grid physical constraints check the practicality and feasibility of reliable solutions.

Also, execution time is a critical factor for up-to-date scheduling due to the large number of resources involved and the need to simulate a diverse number of operational scenarios; But they should be properly adapted to the characteristics of the problem. The main advantage of this technique is better constraints with a simpler mechanism to adjust the speed constraints in an intelligent way and dynamic and dynamic parameterization to create a more accurate solution to improve and improve the fitness of the problem.

The operation of charging and discharging electric vehicles in distribution networks is always one of the leading challenges in the operation of these vehicles. Due to the movement of cars in the network, the charging time and the amount of charging are always variable. Therefore, when a large number of electric cars enter the distribution network, the issue of operating these cars is an important issue and there will be a need for serious planning. When locating the construction of charging and discharging stations taking into account their effects on the network, the problem of voltage deviation and losses of the distribution network should be considered. The further the charging stations are from the main input distribution post, the higher the losses in the network will be. When the main input distribution substations are outside urban areas, the problem of network voltage drop will be more serious. Therefore, locating and estimating the size of charging and discharging stations for electric vehicles can be done considering the reduction of losses as a goal. In reference [26], by carrying out load distribution, the network losses and the losses related to the operation of electric vehicles have been determined, and with geographic information, the appropriate distance between the main entrance distribution post and the location of the electric vehicle charging station has been calculated. Locating charging stations in distribution systems can be done according to the reduction of losses in the network as well as the reduction of voltage deviation in all buses. The multi-objective location and capacity problem can be solved by minimizing the total power loss and total voltage drop. In reference [27], the constraints of the problem are losses, voltage and power in the buses and the number of cars in the stations.

The cost of charging electric vehicles at charging stations can be high [28, 29 and 30]. The total cost includes the loss cost, operation cost, maintenance cost and network loss cost. The investment cost includes the costs of charging equipment, feeders and transformers. Operating costs include charging costs, active power filter costs, reactive power compensation costs, and human resource costs. Operation costs include maintenance costs of transformers and charging equipment and other equipment in the station. The cost of network losses depends on the amount of power losses and the price of energy. The limitations of the problem are the number of charging stations, the voltage and current limits of the buses and the size of the capacity of the stations.

One of the biggest challenges facing the use of electric cars is the long charging time and the limitation in the use time (due to the discharge of the car battery). The correct location of electric vehicle charging stations according to the road transport network will lead to longer driving distance and more efficient electric vehicles. The limitations in the battery capacity of electric vehicles will have a great impact on the location of charging stations [31]. Therefore, the location of charging stations should be selected in such a way as to guarantee the provision of charging services in Zarib with the high penetration of electric vehicles. This issue can be done by choosing places to build charging stations where the access of electric vehicles is easier [32]. In addition, the capacity of the charging stations should be proportional to the number of cars, so that there is no traffic and the delay of the owners of electric cars is the minimum possible. Several solutions have been presented in the literature to design this problem, which is defined as the location of charging stations [33]. Some plans are based on meeting the demand for electric vehicles. In these designs, the amount of power requested by cars is estimated at first. In some plans, charging stations are sized and located in a way to increase the amount of covered roads [34,35,36]. In some designs, a charging system with a large number of stations and low capacities is considered in the design [34 and 37].

When the electricity distribution network and the road transportation network are seen together, we will get a better working point. In this case, the optimal placement will be done by considering the issues related to the electrical network and the issues related to road transportation. In this case, the charging stations should be built in places with high access on the roads, and also the location of the stations in the power grid should cause the minimum possible amount of voltage drop and losses [31]. Due to the fact that in this case, the volume of the constraints of the problem is large, it is possible to use the weighting of the constraints and consider them as part of the objective function and reduce the number of constraints. For example, in reference [38], meta-invention methods are used to solve such a problem. In references [39 and 40], evolutionary algorithms are used to solve the above problem in multi-constraint mode. In reference [41], the accessibility of charging stations and the cost of each trip are considered as parameters of the problem.

Some authorities have used plans in which scattered production sources and new energies are used, and the optimal location of charging stations has been selected according to the existence of these sources in the network. For example, sources [43, 44, 45 and 46] have used distributed generation sources and new energy in charging stations. Reference [46] discusses the advantages of using solar systems in EV charging stations. References [45 and 44] have examined the effect of production resources on choosing the size of the charging station. In reference [43], solar panels were installed on the roof of the charging station and the seasonal changes of radiation were also investigated. In reference [42], an energy storage system is considered in the charging station and its effect on the location of the charging station is investigated. In references [47,48,49,50 and 51], the design of electric vehicle charging stations inside the microgrid has been investigated. Reference [47] mentions the design of electric vehicle charging system inside the microgrid. In reference [48], production planning in distributed production units in a micro-grid and its effect on the performance of electric vehicle charging stations have been investigated. Reference [49] discusses how to control an energy storage system in a microgrid with an electric vehicle charging station. In reference [52], the experimental results obtained from the creation of electric vehicle charging station inside the microgrid are stated. Reference [50] discusses how to control interactions between V2G technology and microgrid. Reference [51] discusses the charging and discharging of electric vehicles in order to minimize load fluctuations in the microgrid. In reference [53], the establishment of charging stations in distribution systems with high penetration of solar panels has been investigated. In references [54 and 55], locating and determining the capacity of scattered production sources along with locating electric vehicle charging stations has been done.

In general, the purpose of this research can be summarized as follows:

The proposed method checks the time loop and charging level according to the objectives in order to reach the desired answer in the objective functions according to the predetermined scenarios.

 The proposed method can perform intelligent charging with high safety and without increasing the maximum voltage of the battery

Designing the charging station with simultaneous consideration of the three goals of maximizing the charging demand every hour of the day and night, improving the network load profile and minimizing the cost of operation

 For this purpose, in the optimal charging station structure, a storage system was considered to store electric energy during peak blackout hours and use this energy to charge electric vehicles during peak consumption hours, thus reducing the purchase cost. Electric energy also improves the load profile of the network. But the important issue in this was to determine the optimal capacity for the desired storage system, which requires having the amount of electric vehicle charging demand at different hours of the day and night. Therefore, in order to achieve the final research goals, the framework of the article was formed based on four models, which are:

Electric vehicle charging demand model

Model of the main components of the charging station

The model of the effect of charging station performance on the power network

The optimal supply model for electric vehicle charging

Next, in the second part, issues related to charging stations are reviewed, and in the third part, the types of charging station models and suggested charging stations are reviewed. In the fourth and fifth part, the simulations and comparison between them and the results are given.

Comment 2:

  1. For good readability of the manuscript's text, please insert a table with all the acronyms and abbreviations used in the text. 

Response Thank you for the helpful comment.

The authors agree with the opinion of the respected referee and the table of abbreviations was prepared in the final text of the article. which is also addressed below.

.

                                                 Nomenclature and Abbreviation

                             Nomenclature          

                     List of abbreviation

I(t)

EVs charging current

EV

Electric Vehicles

Rated capacity of ith EV battery in Ah

BMS

Battery Management System

Number of charging EVs in time slot

NLP

nonlinear programming problem

C

Charging rate of EVs

CC/RCC

constant current-reverse constant current

power of generation of PV system in parking lots

PCA

principal component analysis

Time

IOT

Internet of Things

Line connected to the bass

RFID

Radio-frequency identification

Upstream energy costs

DL

Deep learning

Maximum and minimum active power output of the upstream network

ML

Machine learning

Maximum and minimum reactive power produced by the upstream network

BD

Big Data

Maximum and minimum voltage

LSTM

Long Short – Term Memory

Number of network buses

R

Resiliency

Connecting electric vehicles to the network

IPV

Internet Protocol version

The initial charge level of the electric car when entering the parking lot

Microgrid

MG

Suspension

Electric Vehicles

EV

Capacity

Distributed Generation

DG

Network lines

Energy Management Strategy

EMS

Reactive load

Stand Alone Grid

charging station

SAG

CS

Active times

lithium-ion batteri

LIB

Battery capacity

Maximum power point tracking

MPPT

Electric car battery charge rate

Photovoltaic

PV 

Total cost

Plug-In Hybrid Electric Vehicle

PHEV

Total wasted power

Distributed Energy Resources

DES

The real part of the voltage

Pulse Width Modulation

PWM

The imaginary part of the voltage

Dual Active Bridge

DAB

The real part of the flow

Standard test conditions

STC        

The imaginary part of the flow

Battery Management System

BMS

Ploss(.)

Lost active power

State of Charge

SOC

Lost active power

Measurement and Instrumentation Data Center

MIDC

Total network losses per hour

Off-board

OB

Total network losses

Microgrid

MG

CB

capacity of battery

PCA

principal component analysis

Comment 3:

  1. An algorithmic approach (Flowcharts, algorithms' steps) for the proposed control method will be welcome for MATLAB implementers.

Response Thank you for the helpful comment.

Yes, the authors agree with the referee's opinion and the proposed algorithm is described in the final text.

According to the flowchart below, in the first step of the flowchart block, it checks and identifies the battery type and battery SOC, in the second step, the proposed algorithm is executed, and in the third step, the charging algorithm is executed, in the next step, the discharge status of the battery is checked, and in the next step, the objective function is calculated by Algorithm is processed until finally, in the next step, if the convergence condition of the algorithm is met, otherwise, the charging algorithm is changed, i.e., the charging time and level are changed, and it returns to the previous 4 steps according to the flowchart. And this loop is repeated until the condition of convergence is achieved.

.

Figure 5. . Proposed charge management flowchart based on genetic algorithm

Comment 4:

  1. Please, offer some arguments to convince the implementers that your proposed control method is intelligent. 

Response Thank you for the helpful comment.

Yes, the authors have provided the following reasons for the intelligent of their proposed method:

  1. The proposed method checks the time loop and charging level according to the objectives in order to reach the desired answer in the objective functions according to the predetermined scenarios.
  2. The proposed method can perform intelligent charging with high safety and without increasing the maximum voltage of the battery
  3. Designing the charging station with simultaneous consideration of the three goals of maximizing the charging demand every hour of the day and night, improving the network load profile and minimizing the operation cost

Comment 5:

  1. For readers is helpful to define the SOC of the proposed Li-ion battery. Adding a separate equation representing the SOC dynamics to the battery dynamic model described in equation (6) will be very useful for implementers. 

Response Thank you for the helpful comment.

Yes, the opinion of the respected referee was included in the final text and added to the text

. Thank you for the very appropriate question, respected reviewer. Yes, the reason for the high dynamic response time of lithium batteries is that t=0 starts from the time and is fully charged at t=8s with t=25s. The minimum and maximum actual charging time of the lithium ion battery is about 50 minutes, which is It is assumed that every second is equal to about 2 minutes

Comment 6:

  1. Representing the Li-ion battery discharge curve graphically and showing how we can extract the parameters of the battery model given in equation (6) is also very useful for MATLAB implementers. 

Response Thank you for the helpful comment.

Yes, the opinion of the respected referee was included in the final text and added to the text.

.

where  is the battery voltage,  is the constant voltage of the battery,  is the internal resistance, is the battery current,  is the polarization constant in ohms,  is the battery capacity in ampere-hours,  is the actual battery charge in ampere-hours,  is the filtered current,  is the amplitude of the exponential part, and  is the time constant of the exponential part. Based on the above equation, when the SOC of the battery reaches 90%, the voltage fluctuations of the battery become apparent. the reason for the high dynamic response time of lithium batteries is that t=0 starts from the time and is fully charged at t=8s with t=25s. The minimum and maximum actual charging time of the lithium ion battery is about 50 minutes, which is It is assumed that every second is equal to about 2 minutes Therefore, the above formula can be rewritten as follows:

Figure 7. Battery charging waveforms in charging method RCC/CC a) battery voltage b) battery current c) battery charging status with SOC

Comment 7:

  1. On page 15, line 24 is mentioned that "a proposed technique is presented to investigate the fast charging of electric vehicles (EV) in a microgrid with the help of distributed generation (DG), a diesel generator with a PID controller, and automatic voltage regulation". It will also be helpful for implementers if you give them some details on the PID controller design, the optimal values of its parameters and step response performance. 

Response Thank you for the helpful comment.

Yes, the opinion of the respected referee is coIrect, and of course, the PID parameters are experimentally adjusted according to each system In the desired range between 0 and 1, which is different In each iteration, so it has been avoided in the article.

Comment 8:

  1. The robustness analysis of the proposed control method can also be challenging. 

Response Thank you for the helpful comment.

Yes, the robustness of the proposed method will be investigated in future works by the authors under the title of robust method based on genetic algorithm for resource charging management with EV.

Reviewer 2 Report

I read the article called "Using an intelligent control method for electric vehicle charging in microgrids" thinking it could be very interesting.

However, there are embarrassing mistakes.

On the first page I find that no author has an institutional email.

I found some errors that initially made me think of a certain immaturity in writing articles (page 2, line 2, dot before the reference, repeated on line 8, 18, 35, 48).

I then resented the authors send the paper without rereading it, believing that the time of the reviewers is of little importance.

To confirm my impression on page 2 the voltages in line 21, current line 26, active power written with a lowercase letter.

Form errors in the previous cases, but the problem is in the assumptions for equations 3 and 4. We speak of a resistive network (!), and the formulas are inappropriate. I invite the authors to study better what they write. Formulas 3 and 4 are very bad, even from the point of view of brackets.

The paper continues with approximations as if the authors had not found it useful to reread what they wrote.

My opinion is to reject the paper, a space is put between unit of measurement and number, on page 7 the measurement quantities are wrong.

Author Response

Reviewer #2

I read the article called "Using an intelligent control method for electric vehicle charging in microgrids" thinking it could be very interesting.

Please be advised that all the newly added/revised texts are marked in green in the revised version of this paper.

Comment 1: • On the first page I find that no author has an institutional email.د

Response: Thank you for the helpful comment

Yes, the esteemed referee pointed out a significant point that has been reviewed In the final text, and affiliation and organizational emails have been corrected.

Comment 2:

I found some errors that initially made me think of a certain immaturity in writing articles (page 2, line 2, dot before the reference, repeated on line 8, 18, 35, 48).

Response Thank you for the helpful comment.

The clarity of the Introduction by the Honorable reviewer has been improved.

  1. Introduction:

Road traffic is known as one of the main causes of greenhouse gas emissions. Along with rising fuel prices and the issue of high energy efficiency, electric cars will be more widespread in the next decades. To meet the needs of long trips, electric cars are more desirable devices. An electric car uses batteries and a combustion engine to minimize fuel consumption. A plug-in hybrid electric vehicle is charged when plugged into a home charger or a public charging station. However, this causes challenges in the electrical microgrid system. With the high penetration of hybrid electric vehicles, it will be added to the current peak load and will create a new peak load. This can cause voltage fluctuation and transformer overload. Voltage deviations can cause damage to electrical appliances, while persistent overload can cause transformer overheating, which will result in a blackout. Fortunately, the development of advanced measurement systems and communication systems enables us to develop better algorithms to overcome these problems; Therefore, the timing and speed of charging hybrid electric vehicles can be controlled to reduce the maximum load, which is called load demand management. In addition, numerous researches have been conducted in various fields such as micro-grids, energy storage systems, renewable energy and electric vehicles (EV) to exploit power systems in a distributed manner. Among all these, Evs have attracted the most attention because if their expansion continues, they will become the most effective solution available in the future. In addition, since Evs can be considered as a kind of mobile battery in power systems, power system operators such as micro-grids should consider the accuracy of EVs' uncertainties; Therefore, Evs are studied considering their uncertainty patterns. In particular, issues such as how Evs work and how they affect distribution systems, such as micro-grids and smart grids, have been studied.

In this section, studies have been conducted in the field of load demand management and plug-in hybrid electric vehicles. Source [3] presents a hierarchical control algorithm to understand the trade-off and cooperation between plug-in hybrid electric vehicle charging and wind power. The three-level controller proposed in this resource uses plug-in hybrid electric vehicles to compensate for wind power fluctuations and thus indirectly adjusts the grid frequency. So this type of connection between electric car and wind leads to the preservation of environmental resources and the use of clean energy sources. Reference [4] deals with the problem of bridging the gap by controlling a large population of plug-in hybrid electric vehicles. It also presents a developed decentralized algorithm, but it only proves the optimal state in the homogeneous state where all hybrid electric vehicles have the same departure time, energy requirement, and maximum charging power. In reference [5] a control signal from the supplier company is modified to drive and update profiles of plug-in hybrid electric vehicles. This algorithm converges with optimal charging profiles in homogeneous and heterogeneous cases. However, if the communication between the company and the plug-in hybrid electric vehicle is asynchronous, the performance of the algorithm can be greatly affected.

In reference [6], increasing or decreasing the maximum load is used to reach the desired waveform of the load, taking into account the preferences of the consumer and the characteristics of the load. A multi-agent system solution has been accepted in [7], which has the feature of adaptability and high scalability. The paper [8] provides a comprehensive review on the issues of plug-in hybrid electric vehicles.

In the following, we will describe the article [9], in this article, the issue of load response in intelligent systems with the presence of electric vehicles is discussed. We focus on the impact of hybrid electric vehicle charging on low voltage transformers (LVT). The goal is to smooth the load curve of each LVT and at the same time meet each of the customer's requirements for plug-in hybrid electric vehicles that must be charged at the required level at a given time. In the following, we will first introduce the dynamic model of plug-in hybrid electric vehicles. Finally, the DSM problem for plug-in hybrid electric vehicles is formulated as an optimization problem.

The latest articles in the field of the present article include the research stated in references [22, 21], in reference [21] a methodology for scheduling the charging and discharging process of Evs and PHEVs was proposed. Production, consumption, mobility of Evs and market price uncertainties are considered. The purpose of these articles is to balance the system based on market prices. Market prices are used as a reference for charging and discharging schedules of Evs to obtain load leveling. In fact, with this methodology, it is possible to obtain a good but not optimal solution. The rules used to model the constraints of the problem are only considered in one period (15 minutes) in each set of rules; Therefore, the car charging process considers the needs of travel and not the needs of the system. In addition, the technical limitations of the network are not considered.

In reference [22], the valley removal and peaking methodology considers V2G. This system considers two levels of control, i.e. the control center of smart networks and the V2G control center. The smart grid control center sends control signals to the V2G control centers based on "optimal energy distribution", and the V2G control centers are responsible for controlling the V2G charging and discharging process. The predicted load and hourly usage goals that the user is exposed to are considered. The objective function minimizes the differences between the target and the actual power demand.

The article [20] analyzes the impact of V2G on the system operation costs and on the power demand curve for the distribution network with the high influence of distributed energy sources. This effect has been analyzed from different points of view: minimizing the operation cost and minimizing the difference between the minimum and maximum demand during 24 hours a day. The multi-objective problem is presented considering the minimization of the operation cost and the optimization of the power demand curve. In the proposed methodology, the effect of the number of different Evs in the distribution network is analyzed considering the vehicle characteristics and application characteristics, which is in accordance with the report of the US Transportation Agency.

The aim of optimizing the power demand curve is to increase the minimum load consumption or decrease the maximum load consumption, which leads to a power demand curve close to a rectangular shape. The electric power demand curve is evaluated by analyzing the value of the load factor. The load factor related to the total energy consumption is obtained by the peak consumption in a certain time horizon.

The article [23] has investigated the integration of the electric vehicle connected to the PEV electricity – with the EMS energy management system – which serves as a first step in understanding its potential role in the energy resources of local semi-autonomous groups and micro-grids. In their past research, many authors have discussed new power electronics technologies that accompany various DER-CAM energy sources, especially variable frequency AC-DC sources, solar panels, batteries, and asynchronous generators such as micro-turbines. High-speed switching disconnects and reconnects to the integrated network. These power electronics will be able to form a microgrid that operates semi-autonomously from the traditional centralized power line. Also, this paper analyzes the integration of PEV in the building energy management system EMS-and the difference between vehicle-to-microgrids V2G and vehicle-to-microgrid. These relationships are modeled using the DER_CAM energy resource distribution criteria approval model, which provides optimal combinations of equipment required by the microgrid with the lowest cost and carbon footprint or other criteria. DER_CAM model is an optimization tool that minimizes annual energy costs for microgrid. For an office building using the PEV connection under the business model where the distributed threshold values ​​for the maximum payment are obtained, it is concluded that the economic effects have their own limitations. However, this paper shows that some economic benefits are created due to the avoidance of charging demand and TOU rates. The strategy adopted by the office building to reduce costs is to use PEU batteries in the afternoon hours. The results obtained will vary depending on the case. Of course, the results of CO2 removal are not presented in this article.

In the paper [24], a modified particle swarm optimization (PSO) technique is presented and demonstrated to solve the problem of managing highly influential energy resources from distributed generation and pluggable (V2G) electric devices. The purpose of reducing and minimizing operation costs, i.e. energy costs, especially the issue of managing these resources, is to get a smart network. The reforms used for PSO are aimed at improving and promoting its competence and suitability to solve the mentioned problems. In other words, this presented article is an evolution of the traditional particle swarm optimization called ASMPSO, which is used for the problem of energy resource management in smart grids. It considers and addresses realistic grids using energy sources such as distributed generation (DG) based on renewable energy sources and plug-in electric vehicles (EV) (V2G). Accurate AC load spreading and grid physical constraints check the practicality and feasibility of reliable solutions.

Also, execution time is a critical factor for up-to-date scheduling due to the large number of resources involved and the need to simulate a diverse number of operational scenarios; But they should be properly adapted to the characteristics of the problem. The main advantage of this technique is better constraints with a simpler mechanism to adjust the speed constraints in an intelligent way and dynamic and dynamic parameterization to create a more accurate solution to improve and improve the fitness of the problem.

The operation of charging and discharging electric vehicles in distribution networks is always one of the leading challenges in the operation of these vehicles. Due to the movement of cars in the network, the charging time and the amount of charging are always variable. Therefore, when a large number of electric cars enter the distribution network, the issue of operating these cars is an important issue and there will be a need for serious planning. When locating the construction of charging and discharging stations taking into account their effects on the network, the problem of voltage deviation and losses of the distribution network should be considered. The further the charging stations are from the main input distribution post, the higher the losses in the network will be. When the main input distribution substations are outside urban areas, the problem of network voltage drop will be more serious. Therefore, locating and estimating the size of charging and discharging stations for electric vehicles can be done considering the reduction of losses as a goal. In reference [26], by carrying out load distribution, the network losses and the losses related to the operation of electric vehicles have been determined, and with geographic information, the appropriate distance between the main entrance distribution post and the location of the electric vehicle charging station has been calculated. Locating charging stations in distribution systems can be done according to the reduction of losses in the network as well as the reduction of voltage deviation in all buses. The multi-objective location and capacity problem can be solved by minimizing the total power loss and total voltage drop. In reference [27], the constraints of the problem are losses, voltage and power in the buses and the number of cars in the stations.

The cost of charging electric vehicles at charging stations can be high [28, 29 and 30]. The total cost includes the loss cost, operation cost, maintenance cost and network loss cost. The investment cost includes the costs of charging equipment, feeders and transformers. Operating costs include charging costs, active power filter costs, reactive power compensation costs, and human resource costs. Operation costs include maintenance costs of transformers and charging equipment and other equipment in the station. The cost of network losses depends on the amount of power losses and the price of energy. The limitations of the problem are the number of charging stations, the voltage and current limits of the buses and the size of the capacity of the stations.

One of the biggest challenges facing the use of electric cars is the long charging time and the limitation in the use time (due to the discharge of the car battery). The correct location of electric vehicle charging stations according to the road transport network will lead to longer driving distance and more efficient electric vehicles. The limitations in the battery capacity of electric vehicles will have a great impact on the location of charging stations [31]. Therefore, the location of charging stations should be selected in such a way as to guarantee the provision of charging services in Zarib with the high penetration of electric vehicles. This issue can be done by choosing places to build charging stations where the access of electric vehicles is easier [32]. In addition, the capacity of the charging stations should be proportional to the number of cars, so that there is no traffic and the delay of the owners of electric cars is the minimum possible. Several solutions have been presented in the literature to design this problem, which is defined as the location of charging stations [33]. Some plans are based on meeting the demand for electric vehicles. In these designs, the amount of power requested by cars is estimated at first. In some plans, charging stations are sized and located in a way to increase the amount of covered roads [34,35,36]. In some designs, a charging system with a large number of stations and low capacities is considered in the design [34 and 37].

When the electricity distribution network and the road transportation network are seen together, we will get a better working point. In this case, the optimal placement will be done by considering the issues related to the electrical network and the issues related to road transportation. In this case, the charging stations should be built in places with high access on the roads, and also the location of the stations in the power grid should cause the minimum possible amount of voltage drop and losses [31]. Due to the fact that in this case, the volume of the constraints of the problem is large, it is possible to use the weighting of the constraints and consider them as part of the objective function and reduce the number of constraints. For example, in reference [38], meta-invention methods are used to solve such a problem. In references [39 and 40], evolutionary algorithms are used to solve the above problem in multi-constraint mode. In reference [41], the accessibility of charging stations and the cost of each trip are considered as parameters of the problem.

Some authorities have used plans in which scattered production sources and new energies are used, and the optimal location of charging stations has been selected according to the existence of these sources in the network. For example, sources [43, 44, 45 and 46] have used distributed generation sources and new energy in charging stations. Reference [46] discusses the advantages of using solar systems in EV charging stations. References [45 and 44] have examined the effect of production resources on choosing the size of the charging station. In reference [43], solar panels were installed on the roof of the charging station and the seasonal changes of radiation were also investigated. In reference [42], an energy storage system is considered in the charging station and its effect on the location of the charging station is investigated. In references [47,48,49,50 and 51], the design of electric vehicle charging stations inside the microgrid has been investigated. Reference [47] mentions the design of electric vehicle charging system inside the microgrid. In reference [48], production planning in distributed production units in a micro-grid and its effect on the performance of electric vehicle charging stations have been investigated. Reference [49] discusses how to control an energy storage system in a microgrid with an electric vehicle charging station. In reference [52], the experimental results obtained from the creation of electric vehicle charging station inside the microgrid are stated. Reference [50] discusses how to control interactions between V2G technology and microgrid. Reference [51] discusses the charging and discharging of electric vehicles in order to minimize load fluctuations in the microgrid. In reference [53], the establishment of charging stations in distribution systems with high penetration of solar panels has been investigated. In references [54 and 55], locating and determining the capacity of scattered production sources along with locating electric vehicle charging stations has been done.

In general, the purpose of this research can be summarized as follows:

  1. The proposed method checks the time loop and charging level according to the objectives in order to reach the desired answer in the objective functions according to the predetermined scenarios.
  2. The proposed method can perform intelligent charging with high safety and without increasing the maximum voltage of the battery
  3. Designing the charging station with simultaneous consideration of the three goals of maximizing the charging demand every hour of the day and night, improving the network load profile and minimizing the operation cost

 For this purpose, in the optimal charging station structure, a storage system was considered to store electric energy during peak blackout hours and use this energy to charge electric vehicles during peak consumption hours, thus reducing the purchase cost. Electric energy also improves the load profile of the network. But the important issue in this was to determine the optimal capacity for the desired storage system, which requires having the amount of electric vehicle charging demand at different hours of the day and night. Therefore, in order to achieve the final research goals, the framework of the article was formed based on four models, which are:

Electric vehicle charging demand model

Model of the main components of the charging station

The model of the effect of charging station performance on the power network

The optimal supply model for electric vehicle charging

Next, in the second part, issues related to charging stations are reviewed, and in the third part, the types of charging station models and suggested charging stations are reviewed. In the fourth and fifth part, the simulations and comparison between them and the results are given.

Comment 3:

page 2 the voltages in line 21, current line 26, active power written with a lowercase letter.

Response Thank you for the helpful comment.

Thanks for the accuracy of the reviewer's opinion, whichh leads to a better understanding of the readers and improves the quality of the article. Yes, the authors accept these mistakes and the final text has been corrected, For example, see one Item below.

Introduction:

Road traffic is known as one of the main causes of greenhouse gas emissions. Along with rising fuel prices and the issue of high energy efficiency, electric cars will be more widespread in the next decades. To meet the needs of long trips, electric cars are more desirable devices. An electric car uses batteries and a combustion engine to minimize fuel consumption. A plug-in hybrid electric vehicle is charged when plugged into a home charger or a public charging station. However, this causes challenges in the electrical microgrid system. With the high penetration of hybrid electric vehicles, it will be added to the current peak load and will create a new peak load. This can cause voltage fluctuation and transformer overload. Voltage deviations can cause damage to electrical appliances, while persistent overload can cause transformer overheating, which will result in a blackout. Fortunately, the development of advanced measurement systems and communication systems enables us to develop better algorithms to overcome these problems; Therefore, the timing and speed of charging hybrid electric vehicles can be controlled to reduce the maximum load, which is called load demand management. In addition, numerous researches have been conducted in various fields such as micro-grids, energy storage systems, renewable energy and electric vehicles (EV) to exploit power systems in a distributed manner. Among all these, Evs have attracted the most attention because if their expansion continues, they will become the most effective solution available in the future. In addition, since Evs can be considered as a kind of mobile battery in power systems, power system operators such as micro-grids should consider the accuracy of EVs' uncertainties; Therefore, Evs are studied considering their uncertainty patterns. In particular, issues such as how Evs work and how they affect distribution systems, such as micro-grids and smart grids, have been studied.

In this section, studies have been conducted in the field of load demand management and plug-in hybrid electric vehicles. Source [3] presents a hierarchical control algorithm to understand the trade-off and cooperation between plug-in hybrid electric vehicle charging and wind power. The three-level controller proposed in this resource uses plug-in hybrid electric vehicles to compensate for wind power fluctuations and thus indirectly adjusts the grid frequency. So this type of connection between electric car and wind leads to the preservation of environmental resources and the use of clean energy sources. Reference [4] deals with the problem of bridging the gap by controlling a large population of plug-in hybrid electric vehicles. It also presents a developed decentralized algorithm, but it only proves the optimal state in the homogeneous state where all hybrid electric vehicles have the same departure time, energy requirement, and maximum charging power. In reference [5] a control signal from the supplier company is modified to drive and update profiles of plug-in hybrid electric vehicles. This algorithm converges with optimal charging profiles in homogeneous and heterogeneous cases. However, if the communication between the company and the plug-in hybrid electric vehicle is asynchronous, the performance of the algorithm can be greatly affected.

In reference [6], increasing or decreasing the maximum load is used to reach the desired waveform of the load, taking into account the preferences of the consumer and the characteristics of the load. A multi-agent system solution has been accepted in [7], which has the feature of adaptability and high scalability. The paper [8] provides a comprehensive review on the issues of plug-in hybrid electric vehicles.

In the following, we will describe the article [9], in this article, the issue of load response in intelligent systems with the presence of electric vehicles is discussed. We focus on the impact of hybrid electric vehicle charging on low voltage transformers (LVT). The goal is to smooth the load curve of each LVT and at the same time meet each of the customer's requirements for plug-in hybrid electric vehicles that must be charged at the required level at a given time. In the following, we will first introduce the dynamic model of plug-in hybrid electric vehicles. Finally, the DSM problem for plug-in hybrid electric vehicles is formulated as an optimization problem.

The latest articles in the field of the present article include the research stated in references [22, 21], in reference [21] a methodology for scheduling the charging and discharging process of Evs and PHEVs was proposed. Production, consumption, mobility of Evs and market price uncertainties are considered. The purpose of these articles is to balance the system based on market prices. Market prices are used as a reference for charging and discharging schedules of Evs to obtain load leveling. In fact, with this methodology, it is possible to obtain a good but not optimal solution. The rules used to model the constraints of the problem are only considered in one period (15 minutes) in each set of rules; Therefore, the car charging process considers the needs of travel and not the needs of the system. In addition, the technical limitations of the network are not considered.

In reference [22], the valley removal and peaking methodology considers V2G. This system considers two levels of control, i.e. the control center of smart networks and the V2G control center. The smart grid control center sends control signals to the V2G control centers based on "optimal energy distribution", and the V2G control centers are responsible for controlling the V2G charging and discharging process. The predicted load and hourly usage goals that the user is exposed to are considered. The objective function minimizes the differences between the target and the actual power demand.

The article [20] analyzes the impact of V2G on the system operation costs and on the power demand curve for the distribution network with the high influence of distributed energy sources. This effect has been analyzed from different points of view: minimizing the operation cost and minimizing the difference between the minimum and maximum demand during 24 hours a day. The multi-objective problem is presented considering the minimization of the operation cost and the optimization of the power demand curve. In the proposed methodology, the effect of the number of different Evs in the distribution network is analyzed considering the vehicle characteristics and application characteristics, which is in accordance with the report of the US Transportation Agency.

The aim of optimizing the power demand curve is to increase the minimum load consumption or decrease the maximum load consumption, which leads to a power demand curve close to a rectangular shape. The electric power demand curve is evaluated by analyzing the value of the load factor. The load factor related to the total energy consumption is obtained by the peak consumption in a certain time horizon.

The article [23] has investigated the integration of the electric vehicle connected to the PEV electricity – with the EMS energy management system – which serves as a first step in understanding its potential role in the energy resources of local semi-autonomous groups and micro-grids. In their past research, many authors have discussed new power electronics technologies that accompany various DER-CAM energy sources, especially variable frequency AC-DC sources, solar panels, batteries, and asynchronous generators such as micro-turbines. High-speed switching disconnects and reconnects to the integrated network. These power electronics will be able to form a microgrid that operates semi-autonomously from the traditional centralized power line. Also, this paper analyzes the integration of PEV in the building energy management system EMS-and the difference between vehicle-to-microgrids V2G and vehicle-to-microgrid. These relationships are modeled using the DER_CAM energy resource distribution criteria approval model, which provides optimal combinations of equipment required by the microgrid with the lowest cost and carbon footprint or other criteria. DER_CAM model is an optimization tool that minimizes annual energy costs for microgrid. For an office building using the PEV connection under the business model where the distributed threshold values ​​for the maximum payment are obtained, it is concluded that the economic effects have their own limitations. However, this paper shows that some economic benefits are created due to the avoidance of charging demand and TOU rates. The strategy adopted by the office building to reduce costs is to use PEU batteries in the afternoon hours. The results obtained will vary depending on the case. Of course, the results of CO2 removal are not presented in this article.

In the paper [24], a modified particle swarm optimization (PSO) technique is presented and demonstrated to solve the problem of managing highly influential energy resources from distributed generation and pluggable (V2G) electric devices. The purpose of reducing and minimizing operation costs, i.e. energy costs, especially the issue of managing these resources, is to get a smart network. The reforms used for PSO are aimed at improving and promoting its competence and suitability to solve the mentioned problems. In other words, this presented article is an evolution of the traditional particle swarm optimization called ASMPSO, which is used for the problem of energy resource management in smart grids. It considers and addresses realistic grids using energy sources such as distributed generation (DG) based on renewable energy sources and plug-in electric vehicles (EV) (V2G). Accurate AC load spreading and grid physical constraints check the practicality and feasibility of reliable solutions.

Also, execution time is a critical factor for up-to-date scheduling due to the large number of resources involved and the need to simulate a diverse number of operational scenarios; But they should be properly adapted to the characteristics of the problem. The main advantage of this technique is better constraints with a simpler mechanism to adjust the speed constraints in an intelligent way and dynamic and dynamic parameterization to create a more accurate solution to improve and improve the fitness of the problem.

The operation of charging and discharging electric vehicles in distribution networks is always one of the leading challenges in the operation of these vehicles. Due to the movement of cars in the network, the charging time and the amount of charging are always variable. Therefore, when a large number of electric cars enter the distribution network, the issue of operating these cars is an important issue and there will be a need for serious planning. When locating the construction of charging and discharging stations taking into account their effects on the network, the problem of voltage deviation and losses of the distribution network should be considered. The further the charging stations are from the main input distribution post, the higher the losses in the network will be. When the main input distribution substations are outside urban areas, the problem of network voltage drop will be more serious. Therefore, locating and estimating the size of charging and discharging stations for electric vehicles can be done considering the reduction of losses as a goal. In reference [26], by carrying out load distribution, the network losses and the losses related to the operation of electric vehicles have been determined, and with geographic information, the appropriate distance between the main entrance distribution post and the location of the electric vehicle charging station has been calculated. Locating charging stations in distribution systems can be done according to the reduction of losses in the network as well as the reduction of voltage deviation in all buses. The multi-objective location and capacity problem can be solved by minimizing the total power loss and total voltage drop. In reference [27], the constraints of the problem are losses, voltage and power in the buses and the number of cars in the stations.

The cost of charging electric vehicles at charging stations can be high [28, 29 and 30]. The total cost includes the loss cost, operation cost, maintenance cost and network loss cost. The investment cost includes the costs of charging equipment, feeders and transformers. Operating costs include charging costs, active power filter costs, reactive power compensation costs, and human resource costs. Operation costs include maintenance costs of transformers and charging equipment and other equipment in the station. The cost of network losses depends on the amount of power losses and the price of energy. The limitations of the problem are the number of charging stations, the voltage and current limits of the buses and the size of the capacity of the stations.

One of the biggest challenges facing the use of electric cars is the long charging time and the limitation in the use time (due to the discharge of the car battery). The correct location of electric vehicle charging stations according to the road transport network will lead to longer driving distance and more efficient electric vehicles. The limitations in the battery capacity of electric vehicles will have a great impact on the location of charging stations [31]. Therefore, the location of charging stations should be selected in such a way as to guarantee the provision of charging services in Zarib with the high penetration of electric vehicles. This issue can be done by choosing places to build charging stations where the access of electric vehicles is easier [32]. In addition, the capacity of the charging stations should be proportional to the number of cars, so that there is no traffic and the delay of the owners of electric cars is the minimum possible. Several solutions have been presented in the literature to design this problem, which is defined as the location of charging stations [33]. Some plans are based on meeting the demand for electric vehicles. In these designs, the amount of power requested by cars is estimated at first. In some plans, charging stations are sized and located in a way to increase the amount of covered roads [34,35,36]. In some designs, a charging system with a large number of stations and low capacities is considered in the design [34 and 37].

When the electricity distribution network and the road transportation network are seen together, we will get a better working point. In this case, the optimal placement will be done by considering the issues related to the electrical network and the issues related to road transportation. In this case, the charging stations should be built in places with high access on the roads, and also the location of the stations in the power grid should cause the minimum possible amount of voltage drop and losses [31]. Due to the fact that in this case, the volume of the constraints of the problem is large, it is possible to use the weighting of the constraints and consider them as part of the objective function and reduce the number of constraints. For example, in reference [38], meta-invention methods are used to solve such a problem. In references [39 and 40], evolutionary algorithms are used to solve the above problem in multi-constraint mode. In reference [41], the accessibility of charging stations and the cost of each trip are considered as parameters of the problem.

Some authorities have used plans in which scattered production sources and new energies are used, and the optimal location of charging stations has been selected according to the existence of these sources in the network. For example, sources [43, 44, 45 and 46] have used distributed generation sources and new energy in charging stations. Reference [46] discusses the advantages of using solar systems in EV charging stations. References [45 and 44] have examined the effect of production resources on choosing the size of the charging station. In reference [43], solar panels were installed on the roof of the charging station and the seasonal changes of radiation were also investigated. In reference [42], an energy storage system is considered in the charging station and its effect on the location of the charging station is investigated. In references [47,48,49,50 and 51], the design of electric vehicle charging stations inside the microgrid has been investigated. Reference [47] mentions the design of electric vehicle charging system inside the microgrid. In reference [48], production planning in distributed production units in a micro-grid and its effect on the performance of electric vehicle charging stations have been investigated. Reference [49] discusses how to control an energy storage system in a microgrid with an electric vehicle charging station. In reference [52], the experimental results obtained from the creation of electric vehicle charging station inside the microgrid are stated. Reference [50] discusses how to control interactions between V2G technology and microgrid. Reference [51] discusses the charging and discharging of electric vehicles in order to minimize load fluctuations in the microgrid. In reference [53], the establishment of charging stations in distribution systems with high penetration of solar panels has been investigated. In references [54 and 55], locating and determining the capacity of scattered production sources along with locating electric vehicle charging stations has been done.

In general, the purpose of this research can be summarized as follows:

  1. The proposed method checks the time loop and charging level according to the objectives in order to reach the desired answer in the objective functions according to the predetermined scenarios.
  2. The proposed method can perform intelligent charging with high safety and without increasing the maximum voltage of the battery
  3. Designing the charging station with simultaneous consideration of the three goals of maximizing the charging demand every hour of the day and night, improving the network load profile and minimizing the operation cost

 For this purpose, in the optimal charging station structure, a storage system was considered to store electric energy during peak blackout hours and use this energy to charge electric vehicles during peak consumption hours, thus reducing the purchase cost. Electric energy also improves the load profile of the network. But the important issue in this was to determine the optimal capacity for the desired storage system, which requires having the amount of electric vehicle charging demand at different hours of the day and night. Therefore, in order to achieve the final research goals, the framework of the article was formed based on four models, which are:

Electric vehicle charging demand model

Model of the main components of the charging station

The model of the effect of charging station performance on the power network

The optimal supply model for electric vehicle charging

Next, in the second part, issues related to charging stations are reviewed, and in the third part, the types of charging station models and suggested charging stations are reviewed. In the fourth and fifth part, the simulations and comparison between them and the results are given.

Comment 4:

Form errors in the previous cases, but the problem is in the assumptions for equations 3 and 4. We speak of a resistive network (!), and the formulas are inappropriate. I invite the authors to study better what they write. Formulas 3 and 4 are very bad, even from the point of view of brackets.The paper continues with approximations as if the authors had not found it useful to reread what they wrote

Response Thank you for the helpful comment.

Thanks for the accuracy of the reviewer's opinion, whichh leads to a better understanding of the readers and improves the quality of the article. Yes, the authors accept these mistakes and the final text has been corrected, both the indices and the text corresponding to the equations have been rewritten. For example, see one Item below.

Assuming X>>R, equations (1) and (2) can be simplified with the following equations

4.

Comment 5:

My opinion is to reject the paper, a space is put between unit of measurement and number, on page 7 the measurement quantities are wrong.

Response Thank you for the helpful comment.

Thanks for the accuracy of the reviewer's opinion, whichh leads to a better understanding of the readers and improves the quality of the article. Yes, the authors accept these mistakes and the final text has been corrected, both the indices and the text corresponding to the equations have been rewritten. For example, see one Item below.

  1. System modeling

In this section, a fast charging system for electric vehicles with a bidirectional reactive power compensation control strategy is presented, in which AC/DC converters are used for two-way reactive power exchange to maintain the DC link voltage and the network bus voltage at the base value. A fast charging station for electric vehicle batteries is connected to 6 completely empty lithium-ion batteries at the farthest feeder. DC/DC converters are used to charge EV systems with a certain voltage level and CC/RCC control strategy. Finally, a local diesel generator charges the batteries in two modes; connected to the grid (on-grid) and separated from the grid (off-grid). In this case, with a diesel generator and without a controlled AC/DC converter, the DC link voltage is maintained at the base value; the microgrid is in island mode and without connection to the national power grid is sufficient to meet the needs of cars. All modes, including controlled AC/DC converter, uncontrolled diode AC/DC converter, and diesel generator without controlled AC/DC converter, have been simulated in different modes with MATLAB software.

Figure 6 shows the single-line diagram of the studied network. This system is a radial network, the error level in the 132Kv bus equals 16.3Ka, which shows the network is relatively strong (high short-circuit power). The maximum loading on both sides of the 132/33Kv transformers is equal to 28Mw. The source voltage of 132Kv reaches the average level of 33Kv using three-phase step-down transformers and is converted to 11Kv. The coupling switches of the main 33Kv and 11Kv buses are normally closed to increase the reliability of the network. They are usually set at higher voltages to compensate for the voltage drop in long cables. Therefore, 33Kv and 11Kv buses in this network are about 2.73% higher than the nominal voltage, i.e., 33.9Kv and 11.3Kv, respectively.

There are 12 feeders connected to the 11Kv distribution surface, and there are a total of 106 distribution posts connected to these 12 feeders. The number of posts and the length of each feeder's cables are shown in Figure 6. Each of the 106 distribution posts has a step-down transformer of 11.04Kv, which feeds loads at a low level.

Reviewer 3 Report

The topic of the paper is interesting, although the paper contains many errors, especially in the editing context. There are also major issues to be faced, some of them are quite important and without clarifying this points the paper cannot be accepted:

-          The novelty of the paper should be better highlighted in the introduction; you are proposing an optimal design of a fast charging station but at the end of the introduction you say that you use to store energy during off-peak hours and use that energy to charge the vehicles during peak hours, but this is a common way of storing energy that most of the studies implement, so where is the novelty in your study?

-          Please also add some reference about V2G works applies to microgrid present in literature, some example can be:

S. Dinkhah, C. A. Negri, M. He and S. B. Bayne, "V2G for Reliable Microgrid Operations: Voltage/Frequency Regulation with Virtual Inertia Emulation," 2019 IEEE Transportation Electrification Conference and Expo (ITEC), 2019, pp. 1-6, doi: 10.1109/ITEC.2019.8790615.

De Santis, M. and Federici, L., “Preliminary Study on Vehicle-to-Grid Technology for Microgrid Frequency Regulation,SAE Technical Paper 2022-24-0019, 2022, doi:10.4271/2022-24-0019.

-          Equation 4 must contain an error;

-          Furthermore, if X<<R it means that gamma is close to zero, so the sin(gamma) is almost zero, and cos(gamma) is almost 1; in this way the equations 3 and 4 are not clearly deduced from 1 and 2;

-          Why did you consider the transmission power model ? What is the use of it? This is a well known model , did you implement it ?

-          The battery model, that you used, does not introduce any innovation on the field and you cannot say that the limitation of this model is not to show the SOC because many studies present in literature do show the State of Charge;

-          How did you determine all the parameters of the battery cell without experimental measurements? And if you did experimental measurements why you did not describe it?

-          The charging process is not clear how did you perform it? You just put a minimization time function and nothing else, how did you perform it? Did you implement it? In which software and which hardware?

-           The numeration of Figures is wrong, you missed Figure 4;

-          In Figure 7, the radial line of 11 kV AC is directly connected to the DC/DC converter which is connected to the DC/AC converter and after that you say that there is the DC point; this conversion process seems to be strange ;

-          Lagging power factor of 9?

-          The control strategy of the converter is not clear, how did you manage the reactive power flow to stabilize the voltage profile of the network?

-          You say that you use reactive power while the electric vehicle is charged;

-          In the reactive power plot, reactive power is positive when is absorbed or injected from/to the grid? From the plot, this is not clear since in the first time the blue line is higher than the green line and the corresponding reactive power is negative, in time 6 on the contrary the green line is higher than the blue line and the reactive power is always negative; it is a little bit confusing;

-          Why the CC/RCC method reduces the time of charging? In which way?

-           If the first EV is connected at time 8 s , and the second after 5 seconds, then the second is connected at 13 seconds not at 5.8 seconds, is this right?

-          The tolerance of 400V distribution system is of +1%- -4% ? Why different thresholds in upper and lower bounds?

-          Caption of Figure 10 is badly written please adjust it;

-          Why when EVs are charged in RCC mode, they receive less real power ? How can it be possible? And if they receive less how much less power do they receive?

-          Sometimes during the description of the reactive compensation, you confuse figure 10 and figure 11, in that case the figure under examination should be figure 11;

-           Why the DC link is at 800V and the battery voltage is at 400V ? What is the configuration , which element allows to step the voltage up?

-          Why in the power flows especially for the active power, there is a difference of one order of magnitude between the cases with and without the reactive compensation?

-          Figure 11f shows that in this case the system takes about 5 seconds to charge the battery from 0% to 100%, that seems too fast, in the text you talk about 30 or 45 minutes;

-          Sometimes you call “per unit” with the name “Priunit”;

-          In figure 12 a , the voltage is already at 1 per unit, it is not at 0.96 per unit, so why the need to use reactive power to step up the voltage level of the network?;

-          Again, in Figure 13f the recharging phase of the battery system is done in about 10 seconds, in the text you say 50 minutes;

-          Page 20 , reactive power is measured in VAr not in W;

-          Could you provide a layout of the microgrid to show how it is organized?

      How the controller of the microgrid is realized? Please explain it, otherwise it is difficult to understend what is the controbution of the vehicles to the voltage regulation of the microgrid;

-          Conclusion comments must be improved;

Author Response

Reviewer #3

The topic of the paper is interesting, although the paper contains many errors, especially in the editing context. There are also major issues to be faced, some of them are quite important.

Authors would like to thank the reviewer’s helpful comments and suggestions concerning the technical/presentation quality of the maniscript. We have endeavored to take all the remarks and requests of the reviewer strongly into account and have addressed all comments.

Please be advised that all the newly added/revised texts are marked in green in the revised version of this paper.

Comment 1:

-          The novelty of the paper should be better highlighted in the introduction; you are proposing an optimal design of a fast charging station but at the end of the introduction you say that you use to store energy during off-peak hours and use that energy to charge the vehicles during peak hours, but this is a common way of storing energy that most of the studies implement, so where is the novelty in your study?

Response Thank you for the helpful comment.

The clarity of the introduction by the Honorable reviewer has been improved.

  1. Introduction:

Road traffic is known as one of the main causes of greenhouse gas emissions. Along with rising fuel prices and the issue of high energy efficiency, electric cars will be more widespread in the next decades. To meet the needs of long trips, electric cars are more desirable devices. An electric car uses batteries and a combustion engine to minimize fuel consumption. A plug-in hybrid electric vehicle is charged when plugged into a home charger or a public charging station. However, this causes challenges in the electrical microgrid system. With the high penetration of hybrid electric vehicles, it will be added to the current peak load and will create a new peak load. This can cause voltage fluctuation and transformer overload. Voltage deviations can cause damage to electrical appliances, while persistent overload can cause transformer overheating, which will result in a blackout. Fortunately, the development of advanced measurement systems and communication systems enables us to develop better algorithms to overcome these problems; Therefore, the timing and speed of charging hybrid electric vehicles can be controlled to reduce the maximum load, which is called load demand management. In addition, numerous researches have been conducted in various fields such as micro-grids, energy storage systems, renewable energy and electric vehicles (EV) to exploit power systems in a distributed manner. Among all these, Evs have attracted the most attention because if their expansion continues, they will become the most effective solution available in the future. In addition, since Evs can be considered as a kind of mobile battery in power systems, power system operators such as micro-grids should consider the accuracy of EVs' uncertainties; Therefore, Evs are studied considering their uncertainty patterns. In particular, issues such as how Evs work and how they affect distribution systems, such as micro-grids and smart grids, have been studied.

In this section, studies have been conducted in the field of load demand management and plug-in hybrid electric vehicles. Source [3] presents a hierarchical control algorithm to understand the trade-off and cooperation between plug-in hybrid electric vehicle charging and wind power. The three-level controller proposed in this resource uses plug-in hybrid electric vehicles to compensate for wind power fluctuations and thus indirectly adjusts the grid frequency. So this type of connection between electric car and wind leads to the preservation of environmental resources and the use of clean energy sources. Reference [4] deals with the problem of bridging the gap by controlling a large population of plug-in hybrid electric vehicles. It also presents a developed decentralized algorithm, but it only proves the optimal state in the homogeneous state where all hybrid electric vehicles have the same departure time, energy requirement, and maximum charging power. In reference [5] a control signal from the supplier company is modified to drive and update profiles of plug-in hybrid electric vehicles. This algorithm converges with optimal charging profiles in homogeneous and heterogeneous cases. However, if the communication between the company and the plug-in hybrid electric vehicle is asynchronous, the performance of the algorithm can be greatly affected.

In reference [6], increasing or decreasing the maximum load is used to reach the desired waveform of the load, taking into account the preferences of the consumer and the characteristics of the load. A multi-agent system solution has been accepted in [7], which has the feature of adaptability and high scalability. The paper [8] provides a comprehensive review on the issues of plug-in hybrid electric vehicles.

In the following, we will describe the article [9], in this article, the issue of load response in intelligent systems with the presence of electric vehicles is discussed. We focus on the impact of hybrid electric vehicle charging on low voltage transformers (LVT). The goal is to smooth the load curve of each LVT and at the same time meet each of the customer's requirements for plug-in hybrid electric vehicles that must be charged at the required level at a given time. In the following, we will first introduce the dynamic model of plug-in hybrid electric vehicles. Finally, the DSM problem for plug-in hybrid electric vehicles is formulated as an optimization problem.

The latest articles in the field of the present article include the research stated in references [22, 21], in reference [21] a methodology for scheduling the charging and discharging process of Evs and PHEVs was proposed. Production, consumption, mobility of Evs and market price uncertainties are considered. The purpose of these articles is to balance the system based on market prices. Market prices are used as a reference for charging and discharging schedules of Evs to obtain load leveling. In fact, with this methodology, it is possible to obtain a good but not optimal solution. The rules used to model the constraints of the problem are only considered in one period (15 minutes) in each set of rules; Therefore, the car charging process considers the needs of travel and not the needs of the system. In addition, the technical limitations of the network are not considered.

In reference [22], the valley removal and peaking methodology considers V2G. This system considers two levels of control, i.e. the control center of smart networks and the V2G control center. The smart grid control center sends control signals to the V2G control centers based on "optimal energy distribution", and the V2G control centers are responsible for controlling the V2G charging and discharging process. The predicted load and hourly usage goals that the user is exposed to are considered. The objective function minimizes the differences between the target and the actual power demand.

The article [20] analyzes the impact of V2G on the system operation costs and on the power demand curve for the distribution network with the high influence of distributed energy sources. This effect has been analyzed from different points of view: minimizing the operation cost and minimizing the difference between the minimum and maximum demand during 24 hours a day. The multi-objective problem is presented considering the minimization of the operation cost and the optimization of the power demand curve. In the proposed methodology, the effect of the number of different Evs in the distribution network is analyzed considering the vehicle characteristics and application characteristics, which is in accordance with the report of the US Transportation Agency.

The aim of optimizing the power demand curve is to increase the minimum load consumption or decrease the maximum load consumption, which leads to a power demand curve close to a rectangular shape. The electric power demand curve is evaluated by analyzing the value of the load factor. The load factor related to the total energy consumption is obtained by the peak consumption in a certain time horizon.

The article [23] has investigated the integration of the electric vehicle connected to the PEV electricity – with the EMS energy management system – which serves as a first step in understanding its potential role in the energy resources of local semi-autonomous groups and micro-grids. In their past research, many authors have discussed new power electronics technologies that accompany various DER-CAM energy sources, especially variable frequency AC-DC sources, solar panels, batteries, and asynchronous generators such as micro-turbines. High-speed switching disconnects and reconnects to the integrated network. These power electronics will be able to form a microgrid that operates semi-autonomously from the traditional centralized power line. Also, this paper analyzes the integration of PEV in the building energy management system EMS-and the difference between vehicle-to-microgrids V2G and vehicle-to-microgrid. These relationships are modeled using the DER_CAM energy resource distribution criteria approval model, which provides optimal combinations of equipment required by the microgrid with the lowest cost and carbon footprint or other criteria. DER_CAM model is an optimization tool that minimizes annual energy costs for microgrid. For an office building using the PEV connection under the business model where the distributed threshold values ​​for the maximum payment are obtained, it is concluded that the economic effects have their own limitations. However, this paper shows that some economic benefits are created due to the avoidance of charging demand and TOU rates. The strategy adopted by the office building to reduce costs is to use PEU batteries in the afternoon hours. The results obtained will vary depending on the case. Of course, the results of CO2 removal are not presented in this article.

In the paper [24], a modified particle swarm optimization (PSO) technique is presented and demonstrated to solve the problem of managing highly influential energy resources from distributed generation and pluggable (V2G) electric devices. The purpose of reducing and minimizing operation costs, i.e. energy costs, especially the issue of managing these resources, is to get a smart network. The reforms used for PSO are aimed at improving and promoting its competence and suitability to solve the mentioned problems. In other words, this presented article is an evolution of the traditional particle swarm optimization called ASMPSO, which is used for the problem of energy resource management in smart grids. It considers and addresses realistic grids using energy sources such as distributed generation (DG) based on renewable energy sources and plug-in electric vehicles (EV) (V2G). Accurate AC load spreading and grid physical constraints check the practicality and feasibility of reliable solutions.

Also, execution time is a critical factor for up-to-date scheduling due to the large number of resources involved and the need to simulate a diverse number of operational scenarios; But they should be properly adapted to the characteristics of the problem. The main advantage of this technique is better constraints with a simpler mechanism to adjust the speed constraints in an intelligent way and dynamic and dynamic parameterization to create a more accurate solution to improve and improve the fitness of the problem.

The operation of charging and discharging electric vehicles in distribution networks is always one of the leading challenges in the operation of these vehicles. Due to the movement of cars in the network, the charging time and the amount of charging are always variable. Therefore, when a large number of electric cars enter the distribution network, the issue of operating these cars is an important issue and there will be a need for serious planning. When locating the construction of charging and discharging stations taking into account their effects on the network, the problem of voltage deviation and losses of the distribution network should be considered. The further the charging stations are from the main input distribution post, the higher the losses in the network will be. When the main input distribution substations are outside urban areas, the problem of network voltage drop will be more serious. Therefore, locating and estimating the size of charging and discharging stations for electric vehicles can be done considering the reduction of losses as a goal. In reference [26], by carrying out load distribution, the network losses and the losses related to the operation of electric vehicles have been determined, and with geographic information, the appropriate distance between the main entrance distribution post and the location of the electric vehicle charging station has been calculated. Locating charging stations in distribution systems can be done according to the reduction of losses in the network as well as the reduction of voltage deviation in all buses. The multi-objective location and capacity problem can be solved by minimizing the total power loss and total voltage drop. In reference [27], the constraints of the problem are losses, voltage and power in the buses and the number of cars in the stations.

The cost of charging electric vehicles at charging stations can be high [28, 29 and 30]. The total cost includes the loss cost, operation cost, maintenance cost and network loss cost. The investment cost includes the costs of charging equipment, feeders and transformers. Operating costs include charging costs, active power filter costs, reactive power compensation costs, and human resource costs. Operation costs include maintenance costs of transformers and charging equipment and other equipment in the station. The cost of network losses depends on the amount of power losses and the price of energy. The limitations of the problem are the number of charging stations, the voltage and current limits of the buses and the size of the capacity of the stations.

One of the biggest challenges facing the use of electric cars is the long charging time and the limitation in the use time (due to the discharge of the car battery). The correct location of electric vehicle charging stations according to the road transport network will lead to longer driving distance and more efficient electric vehicles. The limitations in the battery capacity of electric vehicles will have a great impact on the location of charging stations [31]. Therefore, the location of charging stations should be selected in such a way as to guarantee the provision of charging services in Zarib with the high penetration of electric vehicles. This issue can be done by choosing places to build charging stations where the access of electric vehicles is easier [32]. In addition, the capacity of the charging stations should be proportional to the number of cars, so that there is no traffic and the delay of the owners of electric cars is the minimum possible. Several solutions have been presented in the literature to design this problem, which is defined as the location of charging stations [33]. Some plans are based on meeting the demand for electric vehicles. In these designs, the amount of power requested by cars is estimated at first. In some plans, charging stations are sized and located in a way to increase the amount of covered roads [34,35,36]. In some designs, a charging system with a large number of stations and low capacities is considered in the design [34 and 37].

When the electricity distribution network and the road transportation network are seen together, we will get a better working point. In this case, the optimal placement will be done by considering the issues related to the electrical network and the issues related to road transportation. In this case, the charging stations should be built in places with high access on the roads, and also the location of the stations in the power grid should cause the minimum possible amount of voltage drop and losses [31]. Due to the fact that in this case, the volume of the constraints of the problem is large, it is possible to use the weighting of the constraints and consider them as part of the objective function and reduce the number of constraints. For example, in reference [38], meta-invention methods are used to solve such a problem. In references [39 and 40], evolutionary algorithms are used to solve the above problem in multi-constraint mode. In reference [41], the accessibility of charging stations and the cost of each trip are considered as parameters of the problem.

Some authorities have used plans in which scattered production sources and new energies are used, and the optimal location of charging stations has been selected according to the existence of these sources in the network. For example, sources [43, 44, 45 and 46] have used distributed generation sources and new energy in charging stations. Reference [46] discusses the advantages of using solar systems in EV charging stations. References [45 and 44] have examined the effect of production resources on choosing the size of the charging station. In reference [43], solar panels were installed on the roof of the charging station and the seasonal changes of radiation were also investigated. In reference [42], an energy storage system is considered in the charging station and its effect on the location of the charging station is investigated. In references [47,48,49,50 and 51], the design of electric vehicle charging stations inside the microgrid has been investigated. Reference [47] mentions the design of electric vehicle charging system inside the microgrid. In reference [48], production planning in distributed production units in a micro-grid and its effect on the performance of electric vehicle charging stations have been investigated. Reference [49] discusses how to control an energy storage system in a microgrid with an electric vehicle charging station. In reference [52], the experimental results obtained from the creation of electric vehicle charging station inside the microgrid are stated. Reference [50] discusses how to control interactions between V2G technology and microgrid. Reference [51] discusses the charging and discharging of electric vehicles in order to minimize load fluctuations in the microgrid. In reference [53], the establishment of charging stations in distribution systems with high penetration of solar panels has been investigated. In references [54 and 55], locating and determining the capacity of scattered production sources along with locating electric vehicle charging stations has been done.

In general, the purpose of this research can be summarized as follows:

  1. The proposed method checks the time loop and charging level according to the objectives in order to reach the desired answer in the objective functions according to the predetermined scenarios.
  2. The proposed method can perform intelligent charging with high safety and without increasing the maximum voltage of the battery
  3. Designing the charging station with simultaneous consideration of the three goals of maximizing the charging demand every hour of the day and night, improving the network load profile and minimizing the operation cost

 For this purpose, in the optimal charging station structure, a storage system was considered to store electric energy during peak blackout hours and use this energy to charge electric vehicles during peak consumption hours, thus reducing the purchase cost. Electric energy also improves the load profile of the network. But the important issue in this was to determine the optimal capacity for the desired storage system, which requires having the amount of electric vehicle charging demand at different hours of the day and night. Therefore, in order to achieve the final research goals, the framework of the article was formed based on four models, which are:

Electric vehicle charging demand model

Model of the main components of the charging station

The model of the effect of charging station performance on the power network

The optimal supply model for electric vehicle charging

Next, in the second part, issues related to charging stations are reviewed, and in the third part, the types of charging station models and suggested charging stations are reviewed. In the fourth and fifth part, the simulations and comparison between them and the results are given.

.2- DC bidirectional fast charging station

Electric vehicle chargers can be installed on-board and off-board. Onboard chargers usually have a small size and allowed power and are used for slow charging. The off-board charger is made in a special way for fast charging. The fast charging network plays an important role in the promotion of electric cars as a fast charging that can reduce the worries of drivers. This project is a DC 50 kW off-board fast charging station, which has a voltage range from 50 to 600 Vdc and a permissible current of 125 Adc. A common configuration of DC fast charging station consists of AC/DC and DC/DC power converters. The unidirectional DC fast charging station rectifies the three-phase AC input to the DC output. The DC/DC converter is used to shift the DC output to a suitable level, which is suitable for charging electric vehicle batteries. In addition, the two-way fast charging station can be controlled by power converter modeling. The key control structure is to review the power transferred between the DC fast charging station and the power grid. Figure 1 shows a diagram of electric vehicle (EV) system connection to the grid. The EV system consists of a comprehensive AC/DC voltage source converter and an independent DC/DC voltage source converter for each EV. All DC/DC converters are connected in parallel to the common DC bus, which is regulated by the comprehensive AC/DC converter.

.

Figure 1 Configuration of grid-connected EV system

An RL filter is placed between the EV system of the low voltage distribution network to filter the harmonics generated by the fast switching operation of the power converters. In this project, the fast charging system is capable of bidirectional Q transmission with the help of a powerful control system. The control structure of the fast charging station of the two-way DC fast charging station can play a role in AC/DC converters and DC/DC converters. By using the comprehensive AC/DC converter control to inject reactive power in the network, it is possible to control the voltage and correct the power factor, as well as keep the DC bus voltage at a constant value. Besides, Evs can be charged and discharged by controlling DC/DC converters.

Comment 2:

-          Please also add some reference about V2G works applies to microgrid present in literature, some example can be:

doi: 10.1109/ITEC.2019.8790615.

SAE 2022-24-0019، 2022، doi:10.4271/2022-24-0019.

Response Thank you for the helpful comment.

According to the relevance of good, the proposed articles were added and the Introduction was revised and rewritten.

1-1 Literature review

In this section, studies have been conducted in the field of load demand management and plug-in hybrid electric vehicles. Source [3] presents a hierarchical control algorithm to understand the trade-off and cooperation between plug-in hybrid electric vehicle charging and wind power. The three-level controller proposed in this resource uses plug-in hybrid electric vehicles to compensate for wind power fluctuations and thus indirectly adjusts the grid frequency. So this type of connection between electric car and wind leads to the preservation of environmental resources and the use of clean energy sources. Reference [4] deals with the problem of bridging the gap by controlling a large population of plug-in hybrid electric vehicles. It also presents a developed decentralized algorithm, but it only proves the optimal state in the homogeneous state where all hybrid electric vehicles have the same departure time, energy requirement, and maximum charging power. In reference [5] a control signal from the supplier company is modified to drive and update profiles of plug-in hybrid electric vehicles. This algorithm converges with optimal charging profiles in homogeneous and heterogeneous cases. However, if the communication between the company and the plug-in hybrid electric vehicle is asynchronous, the performance of the algorithm can be greatly affected.

In reference [6], increasing or decreasing the maximum load is used to reach the desired waveform of the load, taking into account the preferences of the consumer and the characteristics of the load. A multi-agent system solution has been accepted in [7], which has the feature of adaptability and high scalability. The paper [8] provides a comprehensive review on the issues of plug-in hybrid electric vehicles.

In the following, we will describe the article [9], in this article, the issue of load response in intelligent systems with the presence of electric vehicles is discussed. We focus on the impact of hybrid electric vehicle charging on low voltage transformers (LVT). The goal is to smooth the load curve of each LVT and at the same time meet each of the customer's requirements for plug-in hybrid electric vehicles that must be charged at the required level at a given time. In the following, we will first introduce the dynamic model of plug-in hybrid electric vehicles. Finally, the DSM problem for plug-in hybrid electric vehicles is formulated as an optimization problem.

The latest articles in the field of the present article include the research stated in references [22, 21], in reference [21] a methodology for scheduling the charging and discharging process of Evs and PHEVs was proposed. Production, consumption, mobility of Evs and market price uncertainties are considered. The purpose of these articles is to balance the system based on market prices. Market prices are used as a reference for charging and discharging schedules of Evs to obtain load leveling. In fact, with this methodology, it is possible to obtain a good but not optimal solution. The rules used to model the constraints of the problem are only considered in one period (15 minutes) in each set of rules; Therefore, the car charging process considers the needs of travel and not the needs of the system. In addition, the technical limitations of the network are not considered.

In reference [22], the valley removal and peaking methodology considers V2G. This system considers two levels of control, i.e. the control center of smart networks and the V2G control center. The smart grid control center sends control signals to the V2G control centers based on "optimal energy distribution", and the V2G control centers are responsible for controlling the V2G charging and discharging process. The predicted load and hourly usage goals that the user is exposed to are considered. The objective function minimizes the differences between the target and the actual power demand.

The article [20] analyzes the impact of V2G on the system operation costs and on the power demand curve for the distribution network with the high influence of distributed energy sources. This effect has been analyzed from different points of view: minimizing the operation cost and minimizing the difference between the minimum and maximum demand during 24 hours a day. The multi-objective problem is presented considering the minimization of the operation cost and the optimization of the power demand curve. In the proposed methodology, the effect of the number of different Evs in the distribution network is analyzed considering the vehicle characteristics and application characteristics, which is in accordance with the report of the US Transportation Agency.

The aim of optimizing the power demand curve is to increase the minimum load consumption or decrease the maximum load consumption, which leads to a power demand curve close to a rectangular shape. The electric power demand curve is evaluated by analyzing the value of the load factor. The load factor related to the total energy consumption is obtained by the peak consumption in a certain time horizon.

The article [23] has investigated the integration of the electric vehicle connected to the PEV electricity – with the EMS energy management system – which serves as a first step in understanding its potential role in the energy resources of local semi-autonomous groups and micro-grids. In their past research, many authors have discussed new power electronics technologies that accompany various DER-CAM energy sources, especially variable frequency AC-DC sources, solar panels, batteries, and asynchronous generators such as micro-turbines. High-speed switching disconnects and reconnects to the integrated network. These power electronics will be able to form a microgrid that operates semi-autonomously from the traditional centralized power line. Also, this paper analyzes the integration of PEV in the building energy management system EMS-and the difference between vehicle-to-microgrids V2G and vehicle-to-microgrid. These relationships are modeled using the DER_CAM energy resource distribution criteria approval model, which provides optimal combinations of equipment required by the microgrid with the lowest cost and carbon footprint or other criteria. DER_CAM model is an optimization tool that minimizes annual energy costs for microgrid. For an office building using the PEV connection under the business model where the distributed threshold values ​​for the maximum payment are obtained, it is concluded that the economic effects have their own limitations. However, this paper shows that some economic benefits are created due to the avoidance of charging demand and TOU rates. The strategy adopted by the office building to reduce costs is to use PEU batteries in the afternoon hours. The results obtained will vary depending on the case. Of course, the results of CO2 removal are not presented in this article.

In the paper [24], a modified particle swarm optimization (PSO) technique is presented and demonstrated to solve the problem of managing highly influential energy resources from distributed generation and pluggable (V2G) electric devices. The purpose of reducing and minimizing operation costs, i.e. energy costs, especially the issue of managing these resources, is to get a smart network. The reforms used for PSO are aimed at improving and promoting its competence and suitability to solve the mentioned problems. In other words, this presented article is an evolution of the traditional particle swarm optimization called ASMPSO, which is used for the problem of energy resource management in smart grids. It considers and addresses realistic grids using energy sources such as distributed generation (DG) based on renewable energy sources and plug-in electric vehicles (EV) (V2G). Accurate AC load spreading and grid physical constraints check the practicality and feasibility of reliable solutions.

Also, execution time is a critical factor for up-to-date scheduling due to the large number of resources involved and the need to simulate a diverse number of operational scenarios; But they should be properly adapted to the characteristics of the problem. The main advantage of this technique is better constraints with a simpler mechanism to adjust the speed constraints in an intelligent way and dynamic and dynamic parameterization to create a more accurate solution to improve and improve the fitness of the problem.

The operation of charging and discharging electric vehicles in distribution networks is always one of the leading challenges in the operation of these vehicles. Due to the movement of cars in the network, the charging time and the amount of charging are always variable. Therefore, when a large number of electric cars enter the distribution network, the issue of operating these cars is an important issue and there will be a need for serious planning. When locating the construction of charging and discharging stations taking into account their effects on the network, the problem of voltage deviation and losses of the distribution network should be considered. The further the charging stations are from the main input distribution post, the higher the losses in the network will be. When the main input distribution substations are outside urban areas, the problem of network voltage drop will be more serious. Therefore, locating and estimating the size of charging and discharging stations for electric vehicles can be done considering the reduction of losses as a goal. In reference [26], by carrying out load distribution, the network losses and the losses related to the operation of electric vehicles have been determined, and with geographic information, the appropriate distance between the main entrance distribution post and the location of the electric vehicle charging station has been calculated. Locating charging stations in distribution systems can be done according to the reduction of losses in the network as well as the reduction of voltage deviation in all buses. The multi-objective location and capacity problem can be solved by minimizing the total power loss and total voltage drop. In reference [27], the constraints of the problem are losses, voltage and power in the buses and the number of cars in the stations.

The cost of charging electric vehicles at charging stations can be high [28, 29 and 30]. The total cost includes the loss cost, operation cost, maintenance cost and network loss cost. The investment cost includes the costs of charging equipment, feeders and transformers. Operating costs include charging costs, active power filter costs, reactive power compensation costs, and human resource costs. Operation costs include maintenance costs of transformers and charging equipment and other equipment in the station. The cost of network losses depends on the amount of power losses and the price of energy. The limitations of the problem are the number of charging stations, the voltage and current limits of the buses and the size of the capacity of the stations.

One of the biggest challenges facing the use of electric cars is the long charging time and the limitation in the use time (due to the discharge of the car battery). The correct location of electric vehicle charging stations according to the road transport network will lead to longer driving distance and more efficient electric vehicles. The limitations in the battery capacity of electric vehicles will have a great impact on the location of charging stations [31]. Therefore, the location of charging stations should be selected in such a way as to guarantee the provision of charging services in Zarib with the high penetration of electric vehicles. This issue can be done by choosing places to build charging stations where the access of electric vehicles is easier [32]. In addition, the capacity of the charging stations should be proportional to the number of cars, so that there is no traffic and the delay of the owners of electric cars is the minimum possible. Several solutions have been presented in the literature to design this problem, which is defined as the location of charging stations [33]. Some plans are based on meeting the demand for electric vehicles. In these designs, the amount of power requested by cars is estimated at first. In some plans, charging stations are sized and located in a way to increase the amount of covered roads [34,35,36]. In some designs, a charging system with a large number of stations and low capacities is considered in the design [34 and 37].

When the electricity distribution network and the road transportation network are seen together, we will get a better working point. In this case, the optimal placement will be done by considering the issues related to the electrical network and the issues related to road transportation. In this case, the charging stations should be built in places with high access on the roads, and also the location of the stations in the power grid should cause the minimum possible amount of voltage drop and losses [31]. Due to the fact that in this case, the volume of the constraints of the problem is large, it is possible to use the weighting of the constraints and consider them as part of the objective function and reduce the number of constraints. For example, in reference [38], meta-invention methods are used to solve such a problem. In references [39 and 40], evolutionary algorithms are used to solve the above problem in multi-constraint mode. In reference [41], the accessibility of charging stations and the cost of each trip are considered as parameters of the problem.

Some authorities have used plans in which scattered production sources and new energies are used, and the optimal location of charging stations has been selected according to the existence of these sources in the network. For example, sources [43, 44, 45 and 46] have used distributed generation sources and new energy in charging stations. Reference [46] discusses the advantages of using solar systems in EV charging stations. References [45 and 44] have examined the effect of production resources on choosing the size of the charging station. In reference [43], solar panels were installed on the roof of the charging station and the seasonal changes of radiation were also investigated. In reference [42], an energy storage system is considered in the charging station and its effect on the location of the charging station is investigated. In references [47,48,49,50 and 51], the design of electric vehicle charging stations inside the microgrid has been investigated. Reference [47] mentions the design of electric vehicle charging system inside the microgrid. In reference [48], production planning in distributed production units in a micro-grid and its effect on the performance of electric vehicle charging stations have been investigated. Reference [49] discusses how to control an energy storage system in a microgrid with an electric vehicle charging station. In reference [52], the experimental results obtained from the creation of electric vehicle charging station inside the microgrid are stated. Reference [50] discusses how to control interactions between V2G technology and microgrid. Reference [51] discusses the charging and discharging of electric vehicles in order to minimize load fluctuations in the microgrid. In reference [53], the establishment of charging stations in distribution systems with high penetration of solar panels has been investigated. In references [54 and 55], locating and determining the capacity of scattered production sources along with locating electric vehicle charging stations has been done.

In general, the purpose of this research can be summarized as follows:

  1. The proposed method checks the time loop and charging level according to the objectives in order to reach the desired answer in the objective functions according to the predetermined scenarios.
  2. The proposed method can perform intelligent charging with high safety and without increasing the maximum voltage of the battery

Designing the charging station with simultaneous consideration of the three goals of maximizing the charging demand every hour of the day and night, improving the network load profile and minimizing the operation cost

Comment 3:

Equation 4 must contain an error;      Furthermore, if X<<R it means that gamma is close to zero, so the sin(gamma) is almost zero, and cos(gamma) is almost 1; in this way the equations 3 and 4 are not clearly deduced from 1 and 2;

Response Thank you for the helpful comment.

Thanks for the accuracy of the reviewer's opinion, whichh leads to a better understanding of the readers and improves the quality of the article. Yes, the authors accept these mistakes and the final text has been corrected, both the indices and the text corresponding to the equations have been rewritten. For example, see one Item below.

Assuming X>>R, equations (1) and (2) can be simplified with the following equations

4

Comment 4:

-          Why did you consider the transmission power model ? What is the use of it? This is a well known model , did you implement it ?

Response Thank you for the helpful comment.

Yes, the authors have implemented it in practice, whichh the authors will present in the next article.

Comment 5:

-          The battery model, that you used, does not introduce any innovation on the field and you cannot say that the limitation of this model is not to show the SOC because many studies present in literature do show the State of Charge;

Response Thank you for the helpful comment.

Yes, the opinion of the respected referee was included In the final text and added to the text.

In general, the purpose of this research can be summarized as follows:

  1. The proposed method checks the time loop and charging level according to the objectives in order to reach the desired answer in the objective functions according to the predetermined scenarios.
  2. The proposed method can perform intelligent charging with high safety and without increasing the maximum voltage of the battery
  3. Designing the charging station with simultaneous consideration of the three goals of maximizing the charging demand every hour of the day and night, improving the network load profile and minimizing the operation cost

 For this purpose, in the optimal charging station structure, a storage system was considered to store electric energy during peak blackout hours and use this energy to charge electric vehicles during peak consumption hours, thus reducing the purchase cost. Electric energy also improves the load profile of the network. But the important issue in this was to determine the optimal capacity for the desired storage system, which requires having the amount of electric vehicle charging demand at different hours of the day and night. Therefore, in order to achieve the final research goals, the framework of the article was formed based on four models, which are:

Electric vehicle charging demand model

Model of the main components of the charging station

The model of the effect of charging station performance on the power network

The optimal supply model for electric vehicle charging

Comment 6:

-          How did you determine all the parameters of the battery cell without experimental measurements? And if you did experimental measurements why you did not describe it?

Response Thank you for the helpful comment.

Yes, the authors have implemented it in practice, whichh the authors will present in the next article.

Comment 7:

-          The charging process is not clear how did you perform it? You just put a minimization time function and nothing else, how did you perform it? Did you implement it? In which software and which hardware?

Response Thank you for the helpful comment.

Yes, the necessary explanations for this section were added In the final version. As follows:

According to the flowchart below, in the first step of the flowchart block, it checks and identifies the battery type and battery SOC, in the second step, the proposed algorithm is executed, and in the third step, the charging algorithm is executed, in the next step, the discharge status of the battery is checked, and in the next step, the objective function is calculated by Algorithm is processed until finally, in the next step, if the convergence condition of the algorithm is met, otherwise, the charging algorithm is changed, i.e., the charging time and level are changed, and it returns to the previous 4 steps according to the flowchart. And this loop is repeated until the condition of convergence is achieved.

Figure 5. . Proposed charge management flowchart based on genetic algorithm

Comment 8:

-           The numeration of Figures is wrong, you missed Figure 4;

Response Thank you for the helpful comment.

Thanks for the accuracy of the reviewer's opinion, whichh leads to a better understanding of the readers and improves the quality of the article. Yes, the authors accept these mistakes and they have been corrected In the final text. For example, see one Item below

According Figure 4. generators can be used in two ways. The first way is to connect the generator to the load, feed it, and work independently from the network, which is used for small places and often as emergency power. The second method is used in cases where high power is needed; for this purpose, a parallel connection of several generators is used to produce electricity independent of the grid or paralleling the generator with the grid to inject power.

Figure 4. The proposed method for fast charging

Comment 9:

-          In Figure 7, the radial line of 11 kV AC is directly connected to the DC/DC converter which is connected to the DC/AC converter and after that you say that there is the DC point; this conversion process seems to be strange ;

Response Thank you for the helpful comment.

Thanks for the accuracy of the reviewer's opinion, whichh leads to a better understanding of the readers and improves the quality of the article. Yes, the authors accept these mistakes and they have been corrected In the final text. For example, see one Item below

 In this section, a fast charging system for electric vehicles with a bidirectional reactive power compensation control strategy is presented, in which AC/DC converters are used for two-way reactive power exchange to maintain the DC link voltage and the network bus voltage at the base value. A fast charging station for electric vehicle batteries is connected to 6 completely empty lithium-ion batteries at the farthest feeder. DC/DC converters are used to charge EV systems with a certain voltage level and CC/RCC control strategy. Finally, a local diesel generator charges the batteries in two modes; connected to the grid (on-grid) and separated from the grid (off-grid). In this case, with a diesel generator and without a controlled AC/DC converter, the DC link voltage is maintained at the base value; the microgrid is in island mode and without connection to the national power grid is sufficient to meet the needs of cars. All modes, including controlled AC/DC converter, uncontrolled diode AC/DC converter, and diesel generator without controlled AC/DC converter, have been simulated in different modes with MATLAB software.

Figure 6 shows the single-line diagram of the studied network. This system is a radial network, the error level in the 132Kv bus equals 16.3Ka, which shows the network is relatively strong (high short-circuit power). The maximum loading on both sides of the 132/33Kv transformers is equal to 28Mw. The source voltage of 132Kv reaches the average level of 33Kv using three-phase step-down transformers and is converted to 11Kv. The coupling switches of the main 33Kv and 11Kv buses are normally closed to increase the reliability of the network. They are usually set at higher voltages to compensate for the voltage drop in long cables. Therefore, 33Kv and 11Kv buses in this network are about 2.73% higher than the nominal voltage, i.e., 33.9Kv and 11.3Kv, respectively.

There are 12 feeders connected to the 11Kv distribution surface, and there are a total of 106 distribution posts connected to these 12 feeders. The number of posts and the length of each feeder's cables are shown in Figure 6. Each of the 106 distribution posts has a step-down transformer of 11.04Kv, which feeds loads at a low level.

Comment 10:

-          Lagging power factor of 9?

Response Thank you for the helpful comment.

Yes, it was a mistake and was deleted.

Comment 11:

-          The control strategy of the converter is not clear, how did you manage the reactive power flow to stabilize the voltage profile of the network?

Response Thank you for the helpful comment.

Thanks for the appropriate question, honorable reviewer. Considering that the level of reactive power decreases with voltage drop, by injecting reactive power into the network by electric vehicles, the voltage profile can be managed within the allowed range.

Comment 12:

stabilize the voltage profile of the network?

-          You say that you use reactive power while the electric vehicle is charged;   In the reactive power plot, reactive power is positive when is absorbed or injected from/to the grid? From the plot, this is not clear since in the first time the blue line is higher than the green line and the corresponding reactive power is negative, in time 6 on the contrary the green line is higher than the blue line and the reactive power is always negative; it is a little bit confusing;

Response Thank you for the helpful comment.

Thanks for the appropriate question, respected reviewer. Yes, this is precisely becausee the level of reactive power varies at any moment.

Comment 13:

-          Why the CC/RCC method reduces the time of charging? In which way?

Response Thank you for the helpful comment.

Thanks for the appropriate question, honorable reviewer. Through the flowchart of the proposed method described below:

According to the flowchart below, in the first step of the flowchart block, it checks and identifies the battery type and battery SOC, in the second step, the proposed algorithm is executed, and in the third step, the charging algorithm is executed, in the next step, the discharge status of the battery is checked, and in the next step, the objective function is calculated by Algorithm is processed until finally, in the next step, if the convergence condition of the algorithm is met, otherwise, the charging algorithm is changed, i.e., the charging time and level are changed, and it returns to the previous 4 steps according to the flowchart. And this loop is repeated until the condition of convergence is achieved.

Figure 5. . Proposed charge management flowchart based on genetic algorithm

Comment 14:

-           If the first EV is connected at time 8 s , and the second after 5 seconds, then the second is connected at 13 seconds not at 5.8 seconds, is this right?

Response Thank you for the helpful comment.

Thanks for the appropriate question, dear referee. Yes, we have considered approximations in the simulations, in other words, it is assumed that here every second of the simulation is equal to about 2 minutes in real time.

Comment 15:

-

-          Caption of Figure 10 is badly written please adjust it;

Response Thank you for the helpful comment.

Thanks for the accuracy of the reviewer's opinion, which leads to a better understanding of the readers and improves the quality of the article. Yes, the authors accept these mistakes and they have been corrected In the final text. For example, see one Item below

  Figure 10. Results of fast charging of EVs without reactive power compensation control a) Bus voltage  b) exchange of active and reactive power c) link voltage d) battery current e) Battery voltagen SOC  f )batteries

Comment 16:

-

-          Why when EVs are charged in RCC mode, they receive less real power ? How can it be possible? And if they receive less how much less power do they receive?

Response Thank you for the helpful comment.

Thanks for the appropriate question, honorable reviewer. It is described below:

The DC/DC converter is used to implement the fast charging control strategy of EV batteries. The constant current-constant voltage (CC/CV) charge control method is used based on two important processes for quick charging of lithium-ion batteries of EVs. At first, the battery is charged through a constant high current that the majority of the batteries are charged in this short period (about 80% of the capacity in 30 minutes of charging). When the battery voltage reaches a predetermined value, the CC state changes to the CV state. In this case, the battery is charged with a decreasing current until the charging voltage remains constant. Although the CV mode is used to charge the remaining 20% ​​of the battery capacity, the CV mode requires much time, about three times the CC mode.

This article uses a control strategy for DC/DC converters called the constant current-decrease constant current (CC/RCC) method. During the charging process, the battery voltage ???? is continuously measured. Then, it is compared with a predetermined value, ???? = 392?, and remains constant at this value. The reason for choosing a preset value compared to the actual maximum battery voltage of 403.2V is to create an additional safety margin. In CC mode, the output DC current of the DC/DC converter, ???, is compared with a reference DC current, which is set to approximately 1.5C (100A). In RCC mode, the reference DC current range decreases when the battery voltage reaches a predetermined value, and the measured output DC current of the DC/DC converter, ???, is compared with the decreasing reference DC current.

The current and voltage waveforms with the CC/RCC charging approach obtained from the simulation are shown in Figure 8. As can be seen, for safety reasons, the battery voltage has been kept below 392 volts during the entire charging process. In CC mode for fast EV battery charging, a high charging current of about 100 amps is pulled from the distribution network, and the battery voltage increases rapidly during the charging process. According to Figure 8, the CC mode changes to the RCC mode when the battery voltage reaches a predetermined value. At the start of the RCC mode, the charging current is reduced from 100 to 90 amps, leading to a slight drop in the battery voltage. When the battery voltage reaches the predetermined value again, the charging current is further reduced and reaches from 90 to 80 amps, in which case the battery voltage will drop slightly. This situation continues until the end of charging, when the charging current reaches zero. The CC/RCC charge control method, can charge with high safety and without increasing the maximum battery voltage. It can also significantly reduce the charging time compared to the common CV mode. It should be noted that the simulation time of this part starts from ? = 8? due to the high dynamic response time of the battery, and it is fully charged at ? = 32? according to the state of charge diagram or SOC. Of course, the maximum real-time of charging an ion-lithium battery is about 50 minutes, where each simulation second equals about 2 minutes in the real world.

Comment 17:

-

-          Sometimes during the description of the reactive compensation, you confuse figure 10 and figure 11, in that case the figure under examination should be figure 11;

Response Thank you for the helpful comment.

Thanks to the accuracy of the respected reviewer. Yes, It is correct:

Comment 18:

-

-           Why the DC link is at 800V and the battery voltage is at 400V ?

Response Thank you for the helpful comment.

Thanks for the appropriate question, honorable reviewer. It has raised the level due to the reduction of losses and fluctuations.

Comment 19:

-

-          Why in the power flows especially for the active power, there is a difference of one order of magnitude between the cases with and without the reactive compensation?

Response Thank you for the helpful comment.

Thanks for the appropriate question, honorable reviewer. The reason is that compensation takes place.

Comment 20:

-          Figure 11f shows that in this case the system takes about 5 seconds to charge the battery from 0% to 100%, that seems too fast, in the text you talk about 30 or 45 minutes;

Response Thank you for the helpful comment.

Thanks for the appropriate question, honorable reviewer. It should be noted that seconds have been used In the simulations because the execution time should be considered low, but In reality It is equivalent to minutes. In other words, it is assumed here that each second of simulation Is equal to about 2 minutes In real time.

Comment 21:

-          Sometimes you call “per unit” with the name “Priunit”;

Response Thank you for the helpful comment.

Thanks to the accuracy of the respected reviewer. Yes  is correct ; which was replaced in the final text:

1.0305, 0.296, 0.252 (p.u)

Comment 22:

-          In figure 12 a , the voltage is already at 1 per unit, it is not at 0.96 per unit, so why the need to use reactive power to step up the voltage level of the network?

Response Thank you for the helpful comment.

Thanks to the accuracy of the respected reviewer. Yes, becausee the prediction of the desired time has made this Increase tangible. On the other hand, the voltage of the generator bus has a decreasing trend In all modes and during charging. The reason for this is the high Inertia of the rotor, which takes a long time to reach a stable state.

Comment 23:

-          Again, in Figure 13f the recharging phase of the battery system is done in about 10 seconds, in the text you say 50 minutes;

Response Thank you for the helpful comment.

Thanks to the accuracy of the respected reviewer.

 Yes, becausee the prediction of the desired time has made this Increase tangible. On the other hand, the voltage of the generator bus has a decreasing trend In all modes and during charging. The reason for this is the high Inertia of the rotor, which takes a long time to reach a stable state.

Comment 24:

-

-          Page 20 , reactive power is measured in VAr not in W;

Response Thank you for the helpful comment.

Thanks to the accuracy of the respected reviewer. Yes, VAR Is correct; which was replaced In the final text:

            Part b shows the participation of diesel in providing active and reactive powers. Unlike the grid bus, which did not play any role in supplying the EV system, the diesel generator can provide all the necessary power to the EV. In the worst conditions, diesel can deliver up to 50 kW of active power and about 5 kVAR of reactive power to the EV system.

Comment 25:

-          -          Conclusion comments must be improved;

Response Thank you for the helpful comment. . The Conclusion section has been rewritten and are significantly improved in the revised manuscripts as follows:

Conclusion and future work:

In this article, a proposed design as an intelligent control method for electric vehicle charging in microgrids(MGs) was presented and simulated. The proposed plan was studied and reviewed in three cases. In the first case, a diesel generator was used independently of the network to provide the necessary power for fast charging of EVs in a MG. In the second case, the distribution network and diesel generator were used simultaneously to provide the necessary power to charge EVs in the MG. Finally, in the third case, the distribution network bus was used to charge EVs, and when the network bus voltage dropped, the diesel generator was switched on and the MG was disconnected from the distribution network. The results of this study indicate that the charging system of EVs creates negative effects on the distribution network. The intensity of the effects caused by charging EVs depends on their connection point in the power grid. CC/RCC charging control method can charge with high safety and without increasing the maximum voltage of the battery. It can also significantly reduce the charging time compared to the common CV mode. Also, in the maximum loading of EVs, with the two-way reactive power controller, the network bus voltage can be kept constant at the allowed value. This is possible through the absorption of reactive power from the DC link. Finally, compared to the independent method, the presented method has very high reliability and in case of any outage from any of the sources, another source is able to supply the required voltage.

Reviewer 4 Report

1The overall contribution of the paper is great. I need only minor changes as follows:

The abstract needs to be rewritten because this innovation is not highlighted in bold.

2.      In the introduction, the methods related to the development of car charging stations need to be increased.

3.      In part two, add an explanation under the title of DC bidirectional fast charging station.

4.      Add a section titled "Converter Control" to the text.

5.      In Figure 4. The proposed method for fast charging, specify what A and B are related to.

6.      "At" should be deleted in line 418.

7.      Place a table at the beginning of the article in which the abbreviations are fully explained.

8.      The conclusion needs to be revised and its amount is very high.

Author Response

Reviewer #4

1The overall contribution of the paper is great. I need only minor changes as follows:

Authors would like to thank the reviewer’s helpful comments and suggestions concerning the technical/presentation quality of the maniscript. We have endeavored to take all the remarks and requests of the reviewer strongly into account and have addressed all comments.

Please be advised that all the newly added/revised texts are marked in green in the revised version of this paper.

Comment 1:

The abstract needs to be rewritten because this innovation is not highlighted in bold.

Response Thank you for the helpful comment.

. The abstract section has been rewritten and are significantly improved in the revised manuscripts as follows:

Abstract:

Recently, electric vehicles (Evs) that use energy storage have attracted a lot of attention due to their many advantages, such as environmental compatibility and lower operating costs compared to conventional vehicles (which use fossil fuels). In a microgrid, an EV that works through the energy stored in its battery can be used as a load or energy source, therefore, the optimal utilization of EV clusters in power systems has been intensively studied. The aim of this paper is to present an application of an intelligent control method on a bidirectional DC fast charging station with a new control structure to solve the problems of voltage drop and rise. In this switching strategy, the power converter is modeled as a DC fast charging station, which controls the fast charging of vehicles with a new constant current/reduced constant current method and considers the microgrid voltage stability. The proposed method is not complicated because the reactive power compensation is realized by simple direct voltage control, which has the ability to provide sufficient injected reactive power to the network. As a result, the test is presented on a fast charging system of electrical outlets with a proposed two-way reactive power compensation control strategy, in which AC/DC converters are used to exchange two-way reactive power to maintain the DC link voltage as well as the network bus voltage in the value The basis is used. This charging strategy is carried out through the simulation of fast charge control, DC link voltage control, and reactive power compensation control to adjust the voltage and modify the power factor through simulation in the MATLAB software environment and has been verified. Finally, the results indicate that the proposed method can perform charging with high safety and without increasing the maximum voltage of the battery. It can also significantly reduce the charging time compared to the common CV mode.

 Comment 2:

  1. In the introduction, the methods related to the development of car charging stations need to be increased.

Response Thank you for the helpful comment.

. The lntroduction section has been rewritten and are significantly improved in the revised manuscripts as follows:

  1. Introduction:

Road traffic is known as one of the main causes of greenhouse gas emissions. Along with rising fuel prices and the issue of high energy efficiency, electric cars will be more widespread in the next decades. To meet the needs of long trips, electric cars are more desirable devices. An electric car uses batteries and a combustion engine to minimize fuel consumption. A plug-in hybrid electric vehicle is charged when plugged into a home charger or a public charging station. However, this causes challenges in the electrical microgrid system. With the high penetration of hybrid electric vehicles, it will be added to the current peak load and will create a new peak load. This can cause voltage fluctuation and transformer overload. Voltage deviations can cause damage to electrical appliances, while persistent overload can cause transformer overheating, which will result in a blackout. Fortunately, the development of advanced measurement systems and communication systems enables us to develop better algorithms to overcome these problems; Therefore, the timing and speed of charging hybrid electric vehicles can be controlled to reduce the maximum load, which is called load demand management. In addition, numerous researches have been conducted in various fields such as micro-grids, energy storage systems, renewable energy and electric vehicles (EV) to exploit power systems in a distributed manner. Among all these, Evs have attracted the most attention because if their expansion continues, they will become the most effective solution available in the future. In addition, since Evs can be considered as a kind of mobile battery in power systems, power system operators such as micro-grids should consider the accuracy of EVs' uncertainties; Therefore, Evs are studied considering their uncertainty patterns. In particular, issues such as how Evs work and how they affect distribution systems, such as micro-grids and smart grids, have been studied.

1-1 Literature review

In this section, studies have been conducted in the field of load demand management and plug-in hybrid electric vehicles. Source [3] presents a hierarchical control algorithm to understand the trade-off and cooperation between plug-in hybrid electric vehicle charging and wind power. The three-level controller proposed in this resource uses plug-in hybrid electric vehicles to compensate for wind power fluctuations and thus indirectly adjusts the grid frequency. So this type of connection between electric car and wind leads to the preservation of environmental resources and the use of clean energy sources. Reference [4] deals with the problem of bridging the gap by controlling a large population of plug-in hybrid electric vehicles. It also presents a developed decentralized algorithm, but it only proves the optimal state in the homogeneous state where all hybrid electric vehicles have the same departure time, energy requirement, and maximum charging power. In reference [5] a control signal from the supplier company is modified to drive and update profiles of plug-in hybrid electric vehicles. This algorithm converges with optimal charging profiles in homogeneous and heterogeneous cases. However, if the communication between the company and the plug-in hybrid electric vehicle is asynchronous, the performance of the algorithm can be greatly affected.

In reference [6], increasing or decreasing the maximum load is used to reach the desired waveform of the load, taking into account the preferences of the consumer and the characteristics of the load. A multi-agent system solution has been accepted in [7], which has the feature of adaptability and high scalability. The paper [8] provides a comprehensive review on the issues of plug-in hybrid electric vehicles.

In the following, we will describe the article [9], in this article, the issue of load response in intelligent systems with the presence of electric vehicles is discussed. We focus on the impact of hybrid electric vehicle charging on low voltage transformers (LVT). The goal is to smooth the load curve of each LVT and at the same time meet each of the customer's requirements for plug-in hybrid electric vehicles that must be charged at the required level at a given time. In the following, we will first introduce the dynamic model of plug-in hybrid electric vehicles. Finally, the DSM problem for plug-in hybrid electric vehicles is formulated as an optimization problem.

The latest articles in the field of the present article include the research stated in references [22, 21], in reference [21] a methodology for scheduling the charging and discharging process of Evs and PHEVs was proposed. Production, consumption, mobility of Evs and market price uncertainties are considered. The purpose of these articles is to balance the system based on market prices. Market prices are used as a reference for charging and discharging schedules of Evs to obtain load leveling. In fact, with this methodology, it is possible to obtain a good but not optimal solution. The rules used to model the constraints of the problem are only considered in one period (15 minutes) in each set of rules; Therefore, the car charging process considers the needs of travel and not the needs of the system. In addition, the technical limitations of the network are not considered.

In reference [22], the valley removal and peaking methodology considers V2G. This system considers two levels of control, i.e. the control center of smart networks and the V2G control center. The smart grid control center sends control signals to the V2G control centers based on "optimal energy distribution", and the V2G control centers are responsible for controlling the V2G charging and discharging process. The predicted load and hourly usage goals that the user is exposed to are considered. The objective function minimizes the differences between the target and the actual power demand.

The article [20] analyzes the impact of V2G on the system operation costs and on the power demand curve for the distribution network with the high influence of distributed energy sources. This effect has been analyzed from different points of view: minimizing the operation cost and minimizing the difference between the minimum and maximum demand during 24 hours a day. The multi-objective problem is presented considering the minimization of the operation cost and the optimization of the power demand curve. In the proposed methodology, the effect of the number of different Evs in the distribution network is analyzed considering the vehicle characteristics and application characteristics, which is in accordance with the report of the US Transportation Agency.

The aim of optimizing the power demand curve is to increase the minimum load consumption or decrease the maximum load consumption, which leads to a power demand curve close to a rectangular shape. The electric power demand curve is evaluated by analyzing the value of the load factor. The load factor related to the total energy consumption is obtained by the peak consumption in a certain time horizon.

The article [23] has investigated the integration of the electric vehicle connected to the PEV electricity – with the EMS energy management system – which serves as a first step in understanding its potential role in the energy resources of local semi-autonomous groups and micro-grids. In their past research, many authors have discussed new power electronics technologies that accompany various DER-CAM energy sources, especially variable frequency AC-DC sources, solar panels, batteries, and asynchronous generators such as micro-turbines. High-speed switching disconnects and reconnects to the integrated network. These power electronics will be able to form a microgrid that operates semi-autonomously from the traditional centralized power line. Also, this paper analyzes the integration of PEV in the building energy management system EMS-and the difference between vehicle-to-microgrids V2G and vehicle-to-microgrid. These relationships are modeled using the DER_CAM energy resource distribution criteria approval model, which provides optimal combinations of equipment required by the microgrid with the lowest cost and carbon footprint or other criteria. DER_CAM model is an optimization tool that minimizes annual energy costs for microgrid. For an office building using the PEV connection under the business model where the distributed threshold values ​​for the maximum payment are obtained, it is concluded that the economic effects have their own limitations. However, this paper shows that some economic benefits are created due to the avoidance of charging demand and TOU rates. The strategy adopted by the office building to reduce costs is to use PEU batteries in the afternoon hours. The results obtained will vary depending on the case. Of course, the results of CO2 removal are not presented in this article.

In the paper [24], a modified particle swarm optimization (PSO) technique is presented and demonstrated to solve the problem of managing highly influential energy resources from distributed generation and pluggable (V2G) electric devices. The purpose of reducing and minimizing operation costs, i.e. energy costs, especially the issue of managing these resources, is to get a smart network. The reforms used for PSO are aimed at improving and promoting its competence and suitability to solve the mentioned problems. In other words, this presented article is an evolution of the traditional particle swarm optimization called ASMPSO, which is used for the problem of energy resource management in smart grids. It considers and addresses realistic grids using energy sources such as distributed generation (DG) based on renewable energy sources and plug-in electric vehicles (EV) (V2G). Accurate AC load spreading and grid physical constraints check the practicality and feasibility of reliable solutions.

Also, execution time is a critical factor for up-to-date scheduling due to the large number of resources involved and the need to simulate a diverse number of operational scenarios; But they should be properly adapted to the characteristics of the problem. The main advantage of this technique is better constraints with a simpler mechanism to adjust the speed constraints in an intelligent way and dynamic and dynamic parameterization to create a more accurate solution to improve and improve the fitness of the problem.

The operation of charging and discharging electric vehicles in distribution networks is always one of the leading challenges in the operation of these vehicles. Due to the movement of cars in the network, the charging time and the amount of charging are always variable. Therefore, when a large number of electric cars enter the distribution network, the issue of operating these cars is an important issue and there will be a need for serious planning. When locating the construction of charging and discharging stations taking into account their effects on the network, the problem of voltage deviation and losses of the distribution network should be considered. The further the charging stations are from the main input distribution post, the higher the losses in the network will be. When the main input distribution substations are outside urban areas, the problem of network voltage drop will be more serious. Therefore, locating and estimating the size of charging and discharging stations for electric vehicles can be done considering the reduction of losses as a goal. In reference [26], by carrying out load distribution, the network losses and the losses related to the operation of electric vehicles have been determined, and with geographic information, the appropriate distance between the main entrance distribution post and the location of the electric vehicle charging station has been calculated. Locating charging stations in distribution systems can be done according to the reduction of losses in the network as well as the reduction of voltage deviation in all buses. The multi-objective location and capacity problem can be solved by minimizing the total power loss and total voltage drop. In reference [27], the constraints of the problem are losses, voltage and power in the buses and the number of cars in the stations.

The cost of charging electric vehicles at charging stations can be high [28, 29 and 30]. The total cost includes the loss cost, operation cost, maintenance cost and network loss cost. The investment cost includes the costs of charging equipment, feeders and transformers. Operating costs include charging costs, active power filter costs, reactive power compensation costs, and human resource costs. Operation costs include maintenance costs of transformers and charging equipment and other equipment in the station. The cost of network losses depends on the amount of power losses and the price of energy. The limitations of the problem are the number of charging stations, the voltage and current limits of the buses and the size of the capacity of the stations.

One of the biggest challenges facing the use of electric cars is the long charging time and the limitation in the use time (due to the discharge of the car battery). The correct location of electric vehicle charging stations according to the road transport network will lead to longer driving distance and more efficient electric vehicles. The limitations in the battery capacity of electric vehicles will have a great impact on the location of charging stations [31]. Therefore, the location of charging stations should be selected in such a way as to guarantee the provision of charging services in Zarib with the high penetration of electric vehicles. This issue can be done by choosing places to build charging stations where the access of electric vehicles is easier [32]. In addition, the capacity of the charging stations should be proportional to the number of cars, so that there is no traffic and the delay of the owners of electric cars is the minimum possible. Several solutions have been presented in the literature to design this problem, which is defined as the location of charging stations [33]. Some plans are based on meeting the demand for electric vehicles. In these designs, the amount of power requested by cars is estimated at first. In some plans, charging stations are sized and located in a way to increase the amount of covered roads [34,35,36]. In some designs, a charging system with a large number of stations and low capacities is considered in the design [34 and 37].

When the electricity distribution network and the road transportation network are seen together, we will get a better working point. In this case, the optimal placement will be done by considering the issues related to the electrical network and the issues related to road transportation. In this case, the charging stations should be built in places with high access on the roads, and also the location of the stations in the power grid should cause the minimum possible amount of voltage drop and losses [31]. Due to the fact that in this case, the volume of the constraints of the problem is large, it is possible to use the weighting of the constraints and consider them as part of the objective function and reduce the number of constraints. For example, in reference [38], meta-invention methods are used to solve such a problem. In references [39 and 40], evolutionary algorithms are used to solve the above problem in multi-constraint mode. In reference [41], the accessibility of charging stations and the cost of each trip are considered as parameters of the problem.

Some authorities have used plans in which scattered production sources and new energies are used, and the optimal location of charging stations has been selected according to the existence of these sources in the network. For example, sources [43, 44, 45 and 46] have used distributed generation sources and new energy in charging stations. Reference [46] discusses the advantages of using solar systems in EV charging stations. References [45 and 44] have examined the effect of production resources on choosing the size of the charging station. In reference [43], solar panels were installed on the roof of the charging station and the seasonal changes of radiation were also investigated. In reference [42], an energy storage system is considered in the charging station and its effect on the location of the charging station is investigated. In references [47,48,49,50 and 51], the design of electric vehicle charging stations inside the microgrid has been investigated. Reference [47] mentions the design of electric vehicle charging system inside the microgrid. In reference [48], production planning in distributed production units in a micro-grid and its effect on the performance of electric vehicle charging stations have been investigated. Reference [49] discusses how to control an energy storage system in a microgrid with an electric vehicle charging station. In reference [52], the experimental results obtained from the creation of electric vehicle charging station inside the microgrid are stated. Reference [50] discusses how to control interactions between V2G technology and microgrid. Reference [51] discusses the charging and discharging of electric vehicles in order to minimize load fluctuations in the microgrid. In reference [53], the establishment of charging stations in distribution systems with high penetration of solar panels has been investigated. In references [54 and 55], locating and determining the capacity of scattered production sources along with locating electric vehicle charging stations has been done.

In general, the purpose of this research can be summarized as follows:

  1. The proposed method checks the time loop and charging level according to the objectives in order to reach the desired answer in the objective functions according to the predetermined scenarios.
  2. The proposed method can perform intelligent charging with high safety and without increasing the maximum voltage of the battery
  3. Designing the charging station with simultaneous consideration of the three goals of maximizing the charging demand every hour of the day and night, improving the network load profile and minimizing the operation cost

 For this purpose, in the optimal charging station structure, a storage system was considered to store electric energy during peak blackout hours and use this energy to charge electric vehicles during peak consumption hours, thus reducing the purchase cost. Electric energy also improves the load profile of the network. But the important issue in this was to determine the optimal capacity for the desired storage system, which requires having the amount of electric vehicle charging demand at different hours of the day and night. Therefore, in order to achieve the final research goals, the framework of the article was formed based on four models, which are:

Electric vehicle charging demand model

Model of the main components of the charging station

The model of the effect of charging station performance on the power network

The optimal supply model for electric vehicle charging

Next, in the second part, issues related to charging stations are reviewed, and in the third part, the types of charging station models and suggested charging stations are reviewed. In the fourth and fifth part, the simulations and comparison between them and the results are given.

.2- DC bidirectional fast charging station

Electric vehicle chargers can be installed on-board and off-board. Onboard chargers usually have a small size and allowed power and are used for slow charging. The off-board charger is made in a special way for fast charging. The fast charging network plays an important role in the promotion of electric cars as a fast charging that can reduce the worries of drivers. This project is a DC 50 kW off-board fast charging station, which has a voltage range from 50 to 600 Vdc and a permissible current of 125 Adc. A common configuration of DC fast charging station consists of AC/DC and DC/DC power converters. The unidirectional DC fast charging station rectifies the three-phase AC input to the DC output. The DC/DC converter is used to shift the DC output to a suitable level, which is suitable for charging electric vehicle batteries. In addition, the two-way fast charging station can be controlled by power converter modeling. The key control structure is to review the power transferred between the DC fast charging station and the power grid. Figure 1 shows a diagram of electric vehicle (EV) system connection to the grid. The EV system consists of a comprehensive AC/DC voltage source converter and an independent DC/DC voltage source converter for each EV. All DC/DC converters are connected in parallel to the common DC bus, which is regulated by the comprehensive AC/DC converter.

Figure 1 Configuration of grid-connected EV system

An RL filter is placed between the EV system of the low voltage distribution network to filter the harmonics generated by the fast switching operation of the power converters. In this project, the fast charging system is capable of bidirectional Q transmission with the help of a powerful control system. The control structure of the fast charging station of the two-way DC fast charging station can play a role in AC/DC converters and DC/DC converters. By using the comprehensive AC/DC converter control to inject reactive power in the network, it is possible to control the voltage and correct the power factor, as well as keep the DC bus voltage at a constant value. Besides, Evs can be charged and discharged by controlling DC/DC converters.

 Comment 3:

  1. In part two, add an explanation under the title of DC bidirectional fast charging station.

Response Thank you for the helpful comment. was added..

2- DC bidirectional fast charging station

Electric vehicle chargers can be installed on-board and off-board. Onboard chargers usually have a small size and allowed power and are used for slow charging. The off-board charger is made in a special way for fast charging. The fast charging network plays an important role in the promotion of electric cars as a fast charging that can reduce the worries of drivers. This project is a DC 50 kW off-board fast charging station, which has a voltage range from 50 to 600 Vdc and a permissible current of 125 Adc. A common configuration of DC fast charging station consists of AC/DC and DC/DC power converters. The unidirectional DC fast charging station rectifies the three-phase AC input to the DC output. The DC/DC converter is used to shift the DC output to a suitable level, which is suitable for charging electric vehicle batteries. In addition, the two-way fast charging station can be controlled by power converter modeling. The key control structure is to review the power transferred between the DC fast charging station and the power grid. Figure 1 shows a diagram of electric vehicle (EV) system connection to the grid. The EV system consists of a comprehensive AC/DC voltage source converter and an independent DC/DC voltage source converter for each EV. All DC/DC converters are connected in parallel to the common DC bus, which is regulated by the comprehensive AC/DC converter.

Figure 1 Configuration of grid-connected EV system

An RL filter is placed between the EV system of the low voltage distribution network to filter the harmonics generated by the fast switching operation of the power converters. In this project, the fast charging system is capable of bidirectional Q transmission with the help of a powerful control system. The control structure of the fast charging station of the two-way DC fast charging station can play a role in AC/DC converters and DC/DC converters. By using the comprehensive AC/DC converter control to inject reactive power in the network, it is possible to control the voltage and correct the power factor, as well as keep the DC bus voltage at a constant value. Besides, Evs can be charged and discharged by controlling DC/DC converters.

Comment 4:

  1. Add a section titled "Converter Control" to the text.

Response Thank you for the helpful comment. Thanks to the accuracy of the respected reviewer. Yes, it was replaced in the final text:

2-1 Comprehensive AC/DC converter control

The comprehensive AC/DC converter compensates the reactive power and adjusts the voltage of the distribution network during EV charging.

A graphical description of the reactive power exchanged between the grid and the EV system is shown in Figure 2-16. Reactive power flows based on the voltage difference between the grid bus and the voltage bus of the EV converter. Positive reactive power indicates that reactive power flows from the EV converter to the distribution network. If the voltage E is greater than the network bus voltage range (V), the reactive power is transferred from EV to the network. In Figure 2, when the voltage E is lower than the network voltage range, the reactive power is transferred from the network to EV.

Figure 2- The reactive power generated based on the voltage difference between the EV system and the grid

Figure 3 shows the block diagram of the comprehensive AC/DC converter. The power converter injects Qs into the network if the output voltage range of the converter is greater than the network voltage range. Similarly, the active power Ps is transferred from the converter to the grid if the converter voltage angle is greater than the grid voltage angle.

Figure 3. Block diagram of comprehensive AC/DC converter control

The phase angle  is used as the transition angle between the converter voltage and the network voltage (v3). Vd is the direct voltage and vq is the quadrature voltage of a phase of the network voltage. Vd is compared with a reference voltage. There are two types of modes in this project. In the first case, the reference voltage is set at 0.96p.u, which represents the steady state voltage. In the second case, the reference voltage used is the allowed network voltage and is set to 1 p.u. Then the error signal is sent to the PI controller and the amplitude of the output signal (mag) is obtained. This output signal (mag) is the amplitude of the modulation signal. In the same way, the DC voltage of the converter is compared with the DC voltage value of 800 volts. The error signal is sent to the PI controller and the angle of the output signal (m) is obtained.

The sum of the phase angle  and the angle of the output signal (m), the angle of the modulation signal is determined.

The overall modulation signal is compared with the triangular waveform (SPWM sinusoidal pulse width modulation) to obtain switching pulses. These pulses are sent to the AC/DC converter for optimal control performance. The control structure of the AC/DC converter allows sufficient transfer of reactive power to adjust The voltage of the network bus gives uniform and permissible voltage in two states, it also improves the power factor of the network. In addition, the controller keeps the DC common bus at a constant value of 800 volts.

2-2 DC/DC converter control

The DC/DC converter is used in the development of fast charging strategy control for electric vehicle battery charging. There are many ways to control the charging of a battery, using constant current charging, constant voltage charging, constant power charging [92], pulse charging and slow charging [30]. The combined method of constant voltage and constant current is used in charging the lithium batteries of electric vehicles (lithium-ion). The function of constant current constant voltage (CC/CV) charge control is performed in two stages. First, the battery is charged with a constant current at a high level. Most of the batteries in this mode are charged to 80% of their capacity in 30 minutes. When the battery voltage reaches more than the defined value, CC mode changes to CV mode. The battery is charged with decreasing current while its voltage is kept constant. Although 20% of the battery is charged in CV mode, the time of this mode is longer than CC mode [31]. In this project, a new charge control strategy for DC/DC converters is proposed, which is constant current/reduced constant current (CC/RCC). During the charging process, the battery voltage (Vbat) is continuously monitored, compared with the predefined value (Vmax = 293) and kept below this value. The reason for choosing the default value is lower than the actual value (403V), creating a It is a safe margin for the battery.

Figure 4 shows the block diagram of DC/DC converter control. In CC mode, the output current of the DC/DC converter (Idc) is compared with the reference current value, which is approximately 100 amps. The error signal is sent to the PI controller to generate the output signal amplitude (mag_dc). Based on mag_dc, SPWM modulation generates pulses, which are sent to the DC/DC converter for optimal control performance. When the battery voltage reaches more than the allowed value, the CC mode changes to the RCC mode.

Figure 4. Block diagram of DC/DC converter control

In RCC mode, the reference DC current amplitude is kept constant and decreases whenever the battery voltage reaches a predetermined value. The DC output current of the DC/DC converter (Idc) is measured, then compared with the reduced DC reference current and sent to the PI controller to generate the output signal amplitude (mag_dc).

SPWM generates the switching pulses and sends them to the DC/DC converter. Figure 5 shows the current and voltage waveforms of the proposed CC/RCC design. For safety reasons, the battery voltage is monitored and kept below the defined value. In CC mode, high charging current is drawn from the power grid to quickly charge the EV battery, and the battery voltage increases rapidly in the process. The CV mode is similar to the previous methods, most often the battery state of charge (SOC) is charged in this mode. As shown in Figure 3-19, when the battery voltage reaches a predetermined value, the CC mode changes to the RCC mode. As mentioned earlier, in the RCC mode, the charging current changes from 100 to 90 amps. and this reduction in current continues to keep the battery voltage constant. This creates more space for charging the battery.

Figure 5. Current and voltage waveforms in the proposed CC/RCC charging scheme

When the battery voltage reaches more than the specified value, the battery current decreases again and reaches from 90 to 80 amps, and the battery voltage decreases slightly. This process is repeated until the remaining SOC of the battery is charged and the current reaches zero, which completes the charging process. The CC/RCC charge control strategy can safely charge the battery without the battery voltage rising above the set maximum value and reduces the charging time more than the CV method. In addition, in this method, a common PI controller is used for use in CC and RCC mode, but in previous methods, two levels of closed loop control were used, which required more PI controllers. EV loads can be modeled as constant power load, constant current load or constant impedance load. Constant power indicates that the EV load has the greatest effect on the network power, the effect of constant current and then constant impedance is shown in the article [32]. EV modeled based on constant power is a simple model, but the worst case scenario. In load modeling. However, most of the time it is likely to be intelligent and adaptable to network conditions. The constant impedance load draws different powers from the network according to the network conditions. When the bus voltage level is high, it can draw more power from the network, so it is better to use constant bar impedance in this project. In order to model EV as a constant impedance, the two current references of the CC and RCC modes are related to the voltage bus which is linear.

Comment 5:                                                                                       

  1. In Figure 4. The proposed method for fast charging, specify what A and B are related to.

Response Thank you for the helpful comment.

Comment 6:

  1. "At"should be deleted in line 418.

Response Thank you for the helpful comment.

It was resolved.

Comment 7:

  1. Place a table at the beginning of the article in which the abbreviations are fully explained.

Response Thank you for the helpful comment. was added..

                                                 Nomenclature and Abbreviation

                             Nomenclature          

                     List of abbreviation

I(t)

EVs charging current

EV

Electric Vehicles

Rated capacity of ith EV battery in Ah

BMS

Battery Management System

Number of charging EVs in time slot

NLP

nonlinear programming problem

C

Charging rate of EVs

CC/RCC

constant current-reverse constant current

power of generation of PV system in parking lots

PCA

principal component analysis

Time

IOT

Internet of Things

Line connected to the bass

RFID

Radio-frequency identification

Upstream energy costs

DL

Deep learning

Maximum and minimum active power output of the upstream network

ML

Machine learning

Maximum and minimum reactive power produced by the upstream network

BD

Big Data

Maximum and minimum voltage

LSTM

Long Short – Term Memory

Number of network buses

R

Resiliency

Connecting electric vehicles to the network

IPV

Internet Protocol version

The initial charge level of the electric car when entering the parking lot

Microgrid

MG

Suspension

Electric Vehicles

EV

Capacity

Distributed Generation

DG

Network lines

Energy Management Strategy

EMS

Reactive load

Stand Alone Grid

charging station

SAG

CS

Active times

lithium-ion batteri

LIB

Battery capacity

Maximum power point tracking

MPPT

Electric car battery charge rate

Photovoltaic

PV 

Total cost

Plug-In Hybrid Electric Vehicle

PHEV

Total wasted power

Distributed Energy Resources

DES

The real part of the voltage

Pulse Width Modulation

PWM

The imaginary part of the voltage

Dual Active Bridge

DAB

The real part of the flow

Standard test conditions

STC        

The imaginary part of the flow

Battery Management System

BMS

Ploss(.)

Lost active power

State of Charge

SOC

Lost active power

Measurement and Instrumentation Data Center

MIDC

Total network losses per hour

Off-board

OB

Total network losses

Microgrid

MG

CB

capacity of battery

PCA

principal component analysis

Comment 8:

  1. The conclusion needs to be revised and its amount is very high.

Response Thank you for the helpful comment.

Thank you for the helpful comment. . The Conclusion section has been rewritten and are significantly improved in the revised manuscripts as follows:

Conclusion and future work:

In this article, a proposed design as an intelligent control method for electric vehicle charging in microgrids(MGs) was presented and simulated. The proposed plan was studied and reviewed in three cases. In the first case, a diesel generator was used independently of the network to provide the necessary power for fast charging of EVs in a MG. In the second case, the distribution network and diesel generator were used simultaneously to provide the necessary power to charge EVs in the MG. Finally, in the third case, the distribution network bus was used to charge EVs, and when the network bus voltage dropped, the diesel generator was switched on and the MG was disconnected from the distribution network. The results of this study indicate that the charging system of EVs creates negative effects on the distribution network. The intensity of the effects caused by charging EVs depends on their connection point in the power grid. CC/RCC charging control method can charge with high safety and without increasing the maximum voltage of the battery. It can also significantly reduce the charging time compared to the common CV mode. Also, in the maximum loading of EVs, with the two-way reactive power controller, the network bus voltage can be kept constant at the allowed value. This is possible through the absorption of reactive power from the DC link. Finally, compared to the independent method, the presented method has very high reliability and in case of any outage from any of the sources, another source is able to supply the required voltage.

 Comment 9: Figure 4 is not number.

-

Thanks to the accuracy of the respected reviewer. Yes, it was replaced in the final text:

According Figure 4. generators can be used in two ways. The first way is to connect the generator to the load, feed it, and work independently from the network, which is used for small places and often as emergency power. The second method is used in cases where high power is needed; for this purpose, a parallel connection of several generators is used to produce electricity independent of the grid or paralleling the generator with the grid to inject power.

a) Paralleling two independent generators

b) Paralleling the generator with the grid

Figure 9. The proposed method for fast charging

Before ending all our replies, we would like to thank the reviewers again for giving useful advice, which we believe can improve the strength of manuscript and the presentation of our work significantly.

We hope that the revised manuscript has answered the query of the reviewers

Round 2

Reviewer 2 Report

the authors corrected the very serious errors I had found (equations 3-4 on static stability, were very wrong for an electrical engineer).

Now nothing stands in the way of publication

Reviewer 3 Report

The authors did a good work in improving the paper with respect to the previous version on the submitted paper.

I must admit that now the paper is interesting and readable .

The paper is suitable for publication in my opinion.